# A number-based inventory of size-resolved black carbon particle emissions by global civil aviation

Xiaole Zhang[1,2], Xi Chen[1] & Jing Wang [1,2]

With the rapidly growing global air traffic, the impacts of the black carbon (BC) in the aviation exhaust on climate, environment and public health are likely rising. The particle number and size distribution are crucial metrics for toxicological analysis and aerosol-cloud interactions. Here, a size-resolved BC particle number emission inventory was developed for the global civil aviation. The BC particle number emission is approximately $(10.9 \pm 2.1) \times 10^{25}$ per year with an average emission index of $(6.06 \pm 1.18) \times 10^{14}$ per kg of burned fuel, which is about 1.3% of the total ground anthropogenic emissions, and 3.6% of the road transport emission. The global aviation emitted BC particles follow a lognormal distribution with a geometric mean diameter (GMD) of $31.99 \pm 0.8$ nm and a geometric standard deviation (GSD) of $1.85 \pm 0.016$. The variabilities of GMDs and GSDs for all flights are about 4.8 and 0.08 nm, respectively. The inventory provides new data for assessing the aviation impacts.

[1] Institute of Environmental Engineering (IfU), ETH Zürich, 8093 Zürich, Switzerland. [2] Laboratory for Advanced Analytical Technologies, Empa, 8600 Dübendorf, Switzerland. Correspondence and requests for materials should be addressed to J.W. (email: jing.wang@ifu.baug.ethz.ch)

Air traffic is estimated to increase more than twofold (in revenue passenger kilometers) in the next 20 years[1]. The potential impacts of aviation emissions on public health, environment and climate have attracted more and more attentions[2]. On the ground, the ultrafine particles (<100 nm) from aircrafts significantly increase the particle number concentrations near airports[3,4]. Toxicological studies showed that there is strong evidence that the particle size, shape, surface and surface properties affect the particles' toxicity[5,6]. The emitted black carbon (BC) particles, which are agglomerates of nearly spherical primary particles mainly composed of graphene lamellae[7], absorb sunlight and may also increase the cirrus cloudiness[8,9] and modify the optical thickness of already-existing cirrus[10]. The aviation emissions mainly occur in the upper troposphere and lowermost stratosphere where the high-altitude clouds influence the global climate by trapping outgoing long-wave radiation and reflecting solar short-wave radiation[11]. It is shown that the formation of the contrail and the contrail cirrus is dependent on the BC number emission[12,13], hence the number and the size distribution of aviation emitted particles are required to investigate such climate effects[14]. The radiative forcing from indirect effects of the aviation-induced cloudiness still remains highly uncertain[15–17].

Aviation emission inventory is the crucial data for an appropriate assessment of these impacts[18]. However, the BC particle number emission is not available in most of the existing inventories. Due to the paucity of BC particle emission data in the ICAO (International Civil Aviation Organization) Emission Databank (EDB)[19], majority of the currently commonly utilized global aviation emission inventories[20–23] only contain gaseous emissions including unburnt hydrocarbons (HC), carbon monoxide (CO) and oxides of nitrogen (NO$_x$). Many studies tried to correlate the BC mass emission with the current regulatory metric Smoke Number (SN), including the SN dependent First Order Approximation (FOA1.0 to FOA3.0) methods[24,25], the semi-empirical SN independent Formation OXidation methods FOX[26] and imFOX[27], and the scaling methods for the cruise emissions[28]. The emergence of these methods led to the developments of BC mass emission inventories, e.g., AERO2k[29], AEIC[26], and AEDT[30]. As the only currently available BC number inventory, AERO2k estimated the number-based emission using simplified general characteristic of particle number emission from different engines.

Recently, ICAO has dedicated strong efforts to restricting particle mass and number emissions from aircraft engines and developing related regulations. In order to standardize the aircraft exhaust BC measurements, Aerospace Information Reports[31] and Aerospace Recommended Practice (ARP)[32] have been issued by the Society of Automotive Engineers E-31 Particulate Matter Committee. Many new standard (ARP6320)[32] compliant measurement campaigns have been conducted[33], e.g., Aviation-Particle Regulatory Instrumentation Demonstration Experiment (A-PRIDE)[34,35]. As more and more information becomes available for the aviation emitted BC (e.g., mass, number, and size distribution), opportunities arise for the development of a size-resolved aircraft exhaust BC number emission inventory. Several recent attempts to estimate BC number emissions from the mass inventories indicated that the geometric mean diameter (GMD) and the geometric standard deviation (GSD) of the emitted particles were the key parameters[36–38]. The development of the BC number emission inventory with size distribution information for global aviation is still in its infancy.

In this study, a size-resolved BC particle number emission inventory (details included as Supplementary Data 1) was developed for the global civil aviation based on the recent measurements. The mass-to-number conversion was conducted using the fractal aggregate theory and the particle mass parameters obtained from A-PRIDE experiments[39]. The GMD and GSD of the particles were estimated with a new fitting correlation based on the combination of both the combustor inlet temperature ($T_3$) and the BC mass emission index (EI$_m$). The inventory development followed the European Environment Agency (EEA) air pollutant emission inventory guidebook[40] by utilizing the global scheduled flight dataset in 2005[23,26,41]. The BC mass emission, the flight altitude, the total CCD (Climb/Cruise/Descent) flight duration and the fuel consumption were from EEA depending on aircraft types and route distances. The Automatic Dependent Surveillance-Broadcast (ADS-B) data was utilized to develop the correlation between the flight durations of CCD sub-phases and the total CCD duration. The flight speed, the fuel flow rate and the fuel consumption for CCD sub-phases were estimated based on the combination of EEA, Flight Data Recorder (FDR) and ADS-B data. The speeds during LTO (Landing and Take-Off) were estimated using FDR data. The jet engine data were from ICAO EDB[19]. The total BC particle number emission of global civil aviation was estimated to be $(10.9 \pm 2.1) \times 10^{25}$ per year, with a GMD of $31.99 \pm 0.8$ nm and a GSD of $1.85 \pm 0.016$. The developed emission inventory provides new input data for models assessing the environmental and climate impacts of the aviation BC emissions.

## Results

**Geometric mean diameter and geometric standard deviation.** The goodness of the fitted relations for GMD and GSD, the two important parameters for the mass-to-number conversion, is shown in Fig. 1. The measured engine emission data were from the literature[14,35,42] and described in Supplementary Table 1 and 2 and Note 1. The estimations of GMDs agree well with both the cruise (solid symbols) and ground (open symbols) measurements in Fig. 1a.

The quality of the new GMD-$T_3$&EI$_m$ correlation is compared with other commonly utilized methods including GMD-fuel flow rate, GMD-$T_3$ and GMD-EI$_m$ in Table 1 (Supplementary Figure 1). Two statistical metrics, adjusted $R^2$ and root mean square deviation (RMSE), were adopted to evaluate the goodness of these correlations. The GMD-$T_3$&EI$_m$ relation outperformed the others with an increase of adjusted $R^2$ by 37% and a decrease of RMSE by 56% on average. The details about all the fitted relations are shown in Supplementary Table 3 and Note 2–5.

The bias-variance trade-off tests (Supplementary Figure 2 and Note 6) indicate that both the training and validation errors simultaneously decrease, when the complexity of the model increases from GMD-fuel (two parameters) to GMD-$T_3$&EI$_m$ (six parameters). With further increase of the complexity from seven to ten parameters (Supplementary Table 4), the training errors nearly remain constant, but the validation errors begin to increase. The GMD-$T_3$&EI$_m$ relation is near the turning point of the validation error curve, indicating that it achieves the balance between bias and variance and is not over-fitted. The independent evaluation (Supplementary Note 7) using the APEX (Aircraft Particle Emissions eXperiment) dataset also shows the predictions have a high correlation with the measurements ($r = 0.91$) and an error RMSE of 3.33 nm (Supplementary Figure 3).

The relation between GSD and $T_3$ is highly scattered, but there is a good dependence of GSD on EI$_m$(BC) (Supplementary Figure 4 and Note 8). Figure 1b shows the goodness of the fit for GSD. The data scattering is relatively low with a small RMSE (0.07) and the adjusted $R^2$ is also acceptable. The GSD-EI$_m$ relation provides generally reasonable estimation of the GSD values except one outlier data point of CFM56-2-C1, which has high uncertainty due to only two measurements of the plume. The independent evaluation using the APEX dataset also shows

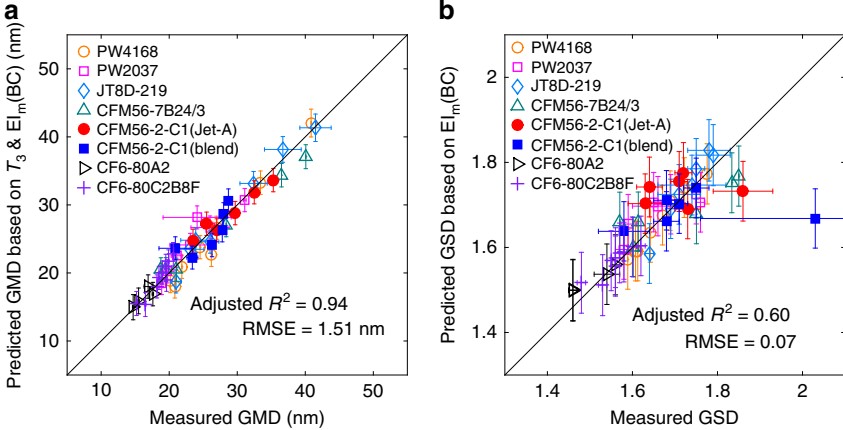

**Fig. 1** Goodness of the adopted fitting relations. **a** the geometric mean diameter (GMD) of the emitted black carbon (BC) particles estimated by the combustor inlet temperature ($T_3$) and the black carbon (BC) mass emission index ($EI_m(BC)$) (defined as the mass of the emitted BC particles due to the combustion of one kilogram of fuel (g kg$^{-1}$-fuel)); **b** the geometric standard deviation (GSD) estimated by $EI_m(BC)$. The vertical error bars represent the standard deviation of the estimations. The horizontal error bars are the uncertainties in the measurements if available

**Table 1 Comparisons among different correlations between geometric mean diameter (GMD) and engine parameters**

|  | GMD-fuel flow rate | GMD-$T_3$ | GMD-$EI_m$ | GMD-$T_3$&$EI_m$ |
|---|---|---|---|---|
| Adjusted $R^2$ | 0.63 | 0.68 | 0.81 | 0.94 |
| RMSE (nm) | 4.03 | 3.69 | 2.82 | 1.51 |

low error (RMSE = 0.06) and high correlation ($r = 0.78$) (Supplementary Figure 3).

**Validation of the black carbon mass-to-number conversion.** As the bridge between mass and number emissions, the BC mass-to-number conversion was based on the Fractal Aggregates (FA) approaches[36–38] and it is evaluated in this section. The $EI_n(BC)$ (about $10^{16}$ kg$^{-1}$-fuel) in Delta-Atlanta Hartsfield Study[42] was nearly one order of magnitude higher than the common values ($10^{14}$–$10^{15}$ kg$^{-1}$-fuel), so we excluded that dataset as an outlier. As a result, the data of CFM56-7B24/3, PW4168, and CFM56-2-C1 (Fig. 1) were adopted. In addition, we also used the BC mass and number emission indices of another five different aircrafts measured at cruise during the experiments SULFUR 1-7[43] (Supplementary Table 5). The measured $EI_m$, engine parameters and flight conditions (e.g., altitude, speed, and fuel flow rate) were utilized as the inputs. The GMD and GSD were estimated by the fitted correlations (Eqs. (6) and (7) in Methods). The estimated number emission indices $EI_n(BC)$ based on Eq. (5) were then compared with the measurements.

Three types of uncertainties were considered: the uncertainties in GMD, GSD, and the exponent ($\varepsilon$) in Eq. (2). All the uncertainties were assumed to be normal distributions. For GMD and GSD, the standard deviation (SD) was estimated based on the 95% prediction interval (1.96 × standard deviation). The SD of $\varepsilon$ was assumed to be 0.025, for which detailed justification is provided in Supplementary Figure 5 and Note 9. The final uncertainties in $EI_n(BC)$ were estimated using the Monte Carlo method with 5000 runs and the calculated uncertainties reached steady state. The estimated $EI_n(BC)$ values are in good agreement with the measurements with a high correlation coefficient (0.85), as shown in Fig. 2. Most of the estimations (74.19%) are within a factor of two of the measurements (dashed lines), and all the estimations are within a factor of three of the measurements

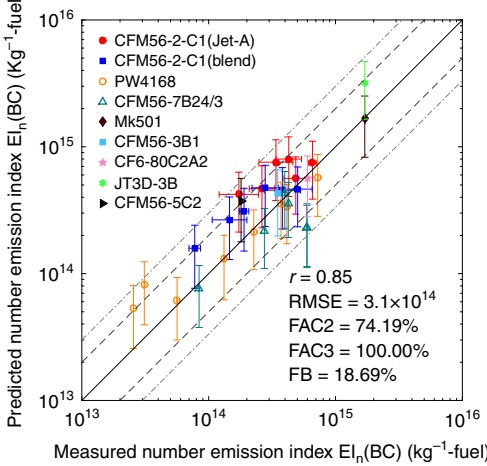

**Fig. 2** Estimations vs. measurements for black carbon particle number emission. The black carbon (BC) particle number emission index $EI_n(BC)$ is the number of the emitted BC particles due to the combustion of one kilogram of fuel (kg$^{-1}$-fuel). FAC2 and FAC3 indicate the fractions of the predictions within a factor of two and three of the measurements, respectively. Dashed lines are 1:2 (or 2:1) ratio lines. Dotted lines are 1:3 (or 3:1) ratio lines. The vertical error bars represent the standard deviation of the estimated $EI_n(BC)$. The horizontal error bars are the uncertainties in the measurements if available

(dotted lines). Generally, the model overestimates with an overall fractional bias of 18.69%. The residuals of the $EI_n(BC)$ estimations for the entire data (cruise + ground), cruise and ground data are close to normal distributions (Supplementary Figure 6 and Note 10). The means are in the order of magnitude of $10^{13}$ kg$^{-1}$-fuel. The biases of the means, in the order of magnitude of $10^{14}$ kg$^{-1}$-fuel, are within the estimated uncertainties.

The scattering of the $EI_n(BC)$ estimations are significant, which may be caused by both the experimental and modeling uncertainties, e.g., the flight parameters, atmospheric conditions and measurement technologies. The uncertainties caused by the estimation methods (vertical error bars) can be as large as a factor of two due to the uncertain particle sizes and the fractal parameters. The estimations are most sensitive to the uncertainty of $\varepsilon$ (Supplementary Figure 7). The independent evaluation using

**Table 2 Comparisons of the black carbon (BC) and gas emissions with other inventories: Aviation Emission Inventory Code (AEIC, 2005)[23, 26], Aviation Environmental Design Tool (AEDT, 2006)[30], Emissions Database for Global Atmospheric Research (EDGAR, 2010)[47] and Global Aircraft Emissions Data Project for Climate Impacts Evaluation (AERO2k, 2002)[29]**

|  | AEIC | AEDT | EDGAR | AERO2k | This study |
|---|---|---|---|---|---|
| $CO_2$ (Gg) | $5.7 \times 10^5$ | $5.9 \times 10^5$ | $7.5 \times 10^5$ | $4.9 \times 10^5$ | $5.7 \times 10^5$ |
| $NO_x$ (Gg) | $2.7 \times 10^3$ | $2.7 \times 10^3$ | $2.9 \times 10^3$ | $2.1 \times 10^3$ | $2.5 \times 10^3$ |
| $SO_x$ (Gg) | $2.2 \times 10^2$ | $2.2 \times 10^2$ | $2.5 \times 10^2$ | / | $1.5 \times 10^2$ |
| CO (Gg) | $7.9 \times 10^2$ | $6.7 \times 10^2$ | $5.3 \times 10^2$ | $5.1 \times 10^2$ | $6.2 \times 10^2$ |
| Fuel burn (Gg) | $1.8 \times 10^5$ | $1.9 \times 10^5$ | / | $1.6 \times 10^5$ | $1.8 \times 10^5$ |
| BC mass (Gg) | $16.9 \ (2.4 + 14.5)^a$ <br> $2.0 \ (0.6 + 1.4)^b$ | 6.8 | $9.0 \ (0.6 + 8.4)^c$ | 3.9 | $9.5 \ (0.7 + 8.8)$ |

$^a$Estimated by the FOX method
$^b$Estimated by the FOA3 method
$^c$The data in the parentheses are the (LTO emissions + CCD emissions)

APEX dataset also shows a similar performance with a high correlation coefficient (0.85) and low RMSE ($1.8 \times 10^{14}$) (kg$^{-1}$-fuel) (Supplementary Figure 3).

**Black carbon and gas mass emissions**. The mass emission inventories were developed by the combination of the EEA emission indices[40] and the global scheduled flight data. The BC and gas ($CO_2$, $NO_x$, $SO_x$, CO) mass emission inventories are compared with the existing ones in the literature (Table 2), to ensure the quality of the development procedures. For the gaseous emissions, the differences between our results and the previous data are within about 20%. The differences of fuel burn are within 10%, indicating the reliability of the EEA data and inventory development procedures.

The calculated total annual BC mass emission is 9.5 Gg, in which 92.5% (8.8 Gg) and 7.5% (0.72 Gg) are, respectively, attributed to the CCD and LTO phases. The estimated LTO mass emission is comparable with the data in AEIC-FOA3 (0.6 Gg) and EDGAR (0.6 Gg) inventories, but significantly smaller than that in AEIC-FOX (2.4 Gg). The recent studies[27,44] indicated that the FOX method might overestimate the BC mass emission by up to a factor of 4. The estimated CCD emission is also in line with the data in EDGAR (8.4 Gg), but the differences are relatively large compared with the AEIC estimations by FOA3 (1.4 Gg) and FOX (14.5 Gg) methods, which is assumed to be caused by the different adopted engine fuel flows at cruise[26]. The result of BC mass in this study is close to EDGAR data, because EDGAR also utilized the EEA database[45]. The current results are about 40% larger than those of AEDT inventory, which adopted a fixed BC emission index (30 mg kg$^{-1}$-fuel) for the CCD emissions[46]. The $EI_m$(BC) of AERO2k are notably smaller than other inventories. AERO2k utilized a SN-dependent estimation method for BC mass emission, so the smaller $EI_m$(BC) may be due to the missing SN in EDB. The compiled global spatial distribution of the BC mass emission is shown in Supplementary Figure 8.

**Particle number emission**. The developed BC mass emission inventory is converted into a number emission inventory. Figure 3 shows the spatial distribution of the BC number emission of global civil aviation with a grid resolution of one degree-by-one degree. The emission statistics are shown in Supplementary Table 6.

The total BC particle number emission is about $1.09 \times 10^{26}$ per year (95% confidence interval of $0.68–1.5 \times 10^{26}$ per year). The LTO cycles below 3000 ft and CCD above 3000 ft, respectively, account for 10.2% ($1.1 \times 10^{25}$) and 89.8% ($9.8 \times 10^{25}$) of the total particle number. The AERO2k inventory[29] developed by EUROCONTROL is the only currently available inventory that explicitly includes the

BC number emissions, which estimated a total number emission of $4.03 \times 10^{25}$ for the year 2002[29]. AERO2k estimated the particle number emissions using approximate ratios between particle number and mass at different altitudes: about $1.6 \times 10^{16}$ g$^{-1}$-BC for cruise and about $5 \times 10^{15}$ g$^{-1}$-BC for LTO. In contrast, our estimations are $1.1 \times 10^{16}$ g$^{-1}$-BC for CCD and $1.5 \times 10^{16}$ g$^{-1}$-BC for LTO. The ratio for LTO is much larger than that utilized by AERO2k due to the high value of taxi ($3.6 \times 10^{16}$ g$^{-1}$-BC), which is in line with the recent measurements showing that the size[7] and mass[39] of BC particles are smaller at lower thrusts. Our overall number-mass ratio ($1.15 \times 10^{16}$ g$^{-1}$-BC) is comparable with that of AERO2k ($1.03 \times 10^{16}$ g$^{-1}$-BC). Our total number emission is about 2.5 times of that in the AERO2k inventory, which may be caused by the low total BC mass emission (3.9 Gg) in AERO2k.

The uncertainty of the estimated BC particle number emissions depends on that of the BC mass emission. Sensitivity tests (Supplementary Note 11) were conducted to evaluate the influences of the BC mass emissions, and the results indicated that when the BC mass emissions were 33% (3.2Gg) and 200% (19.1Gg) of the baseline emission (9.5 Gg), the variations of the estimated particle number emissions ($6.0 \times 10^{25}$ and $15.5 \times 10^{25}$) were near the 95% confidence interval ($0.68–1.5 \times 10^{26}$ per year), shown as the dashed lines in Supplementary Figure 9. Therefore our number estimation was well constrained in the presence of the significant mass emission uncertainty.

The average $EI_n$(BC) of all flight phases is estimated to be $6.06 \times 10^{14}$ kg$^{-1}$-fuel with the 95% confidence interval between 3.74 and $8.37 \times 10^{14}$ kg$^{-1}$-fuel. The spatial distribution of $EI_n$(BC) is shown in Supplementary Figure 10. The largest $EI_n$(BC) is obtained in the climb phase above 3000 ft, with a value of $(9.2 \pm 1.95) \times 10^{14}$ kg$^{-1}$-fuel, in line with the typical emission characteristics for the widely used Rich–Quench–Lean combustors, which have the maximum $EI_n$(BC) at about 65% thrust and somewhat lower $EI_n$(BC) with higher thrust[44].

The descent phase has the lowest $EI_n$(BC), $(3.9 \pm 0.93) \times 10^{14}$ kg$^{-1}$-fuel. The average $EI_n$(BC)s for the LTO and CCD phases are, respectively, $(5.02 \pm 1.00) \times 10^{14}$ and $(6.2 \pm 1.22) \times 10^{14}$ kg$^{-1}$-fuel, which are about 2.5 and 1.5 times of the values recommended in the AEDT gridded aircraft emission inventory[30]. In AEDT, there was no BC number emission inventory, but a couple of emission indices averaged from the literature data were recommended for all the aircrafts as $2 \times 10^{14}$ ($0.1 \times 10^{14}$–$6 \times 10^{14}$) kg$^{-1}$-fuel below 3000 ft and $4 \times 10^{14}$ ($1 \times 10^{14}$–$60 \times 10^{14}$) kg$^{-1}$-fuel above 3000 ft[46]. The values in AEDT were also adopted by Aviation Climate Change Research Initiative (ACCRI)[48,49] of Federal Aviation Administration (FAA) to assess the climate impacts of aviation emissions. It was reported that the BC number emission index of Boeing 737-800 with CFM56−7B26 engine was about $6.8 \times 10^{14}$ kg$^{-1}$-

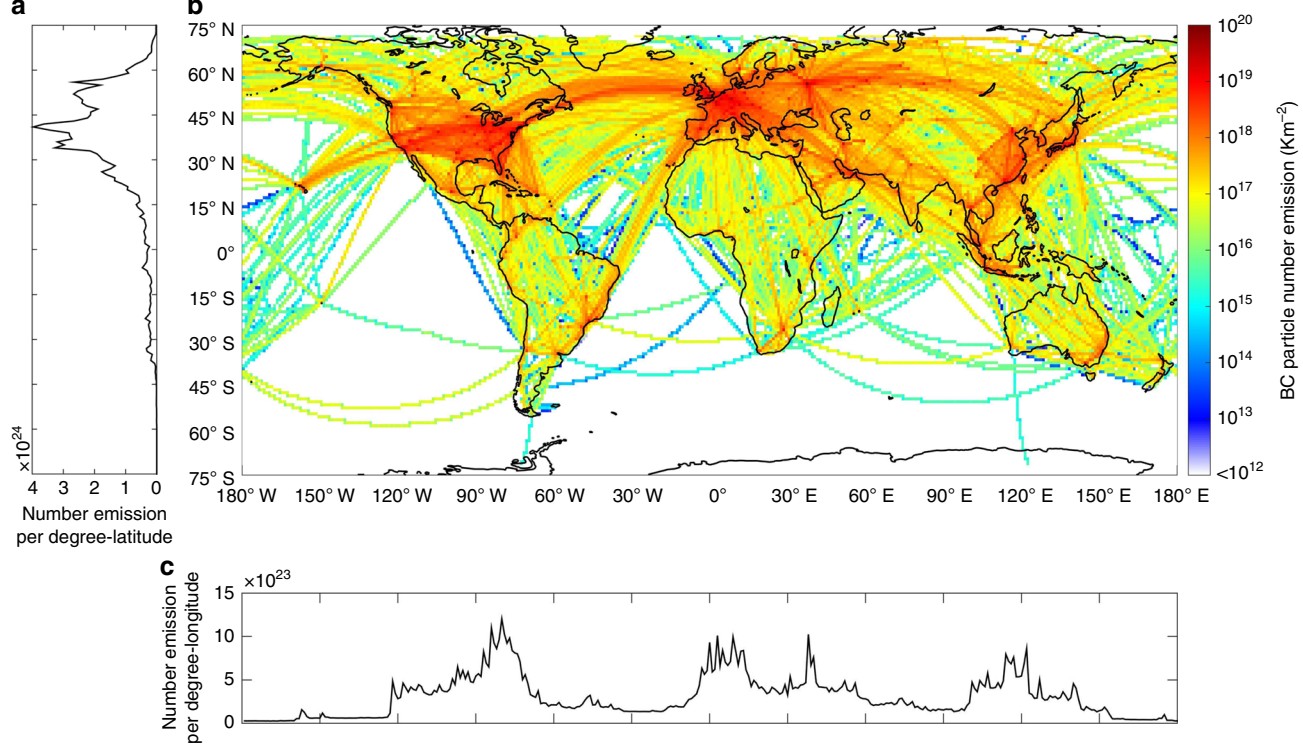

**Fig. 3** Spatial distribution of black carbon number emission. **a** Latitudinal distribution of the number emissions (integration in each latitudinal band with the resolution of 1°); **b** the spatial distribution of number emission; **c** longitudinal distribution of the number emissions (integration in each longitudinal band with the resolution of 1°)

fuel at cruise (35,000 ft and 0.8 Mach)[44], which is close to our estimations.

**Particle size distribution**. The developed global inventory has 16 size bins (Supplementary Table 7), which were calculated by integrating the size-resolved emissions from all the aircrafts. The global BC size distribution was obtained by 100 Monte Carlo (MC) simulations with the uncertainties in the parameters of mass-to-number conversion taken into account. The results reached steady state at 100 runs (Supplementary Figure 11). The inventory is the sum of the emissions from millions of different aircrafts. Each MC run for the global size distribution already covers the calculation of millions of aircrafts and routes so the results converge at a lower number of runs compared to the results of individual aircrafts.

Figure 4a and b illustrates the integrated particle size distributions during the entire LTO and CCD phases, as well as those for their individual sub-phases. The dots represent the mean of all MC runs and the shaded area indicates the standard deviation of all the MC runs in each size bin. The GMD and GSD of the global BC size distribution were obtained by fitting the particle numbers in the size bins into the lognormal distribution for each MC run. The uncertainties of GMD and GSD were estimated from these MC runs. The global fitting of all the particles gives the GMD of 31.99 ± 0.8 nm and the GSD of 1.85 ± 0.016. For LTO, the GMD is 29.51 ± 0.97 nm with a GSD of 1.88 ± 0.023. The particles emitted during CCD phases are larger with a GMD of 32.27 ± 0.81 nm and a GSD of 1.84 ± 0.016 (Supplementary Table 6).

The variabilities were also calculated as the standard deviations of the emissions from all the 27 million flights in the seven sub-phases (taxi, take-off, climb-out, approach, climb, cruise and descent). The variabilities of GMD and GSD are about 4.8 nm and 0.08, which are about five times of the estimated uncertainties

(Supplementary Table 8). More details about the uncertainties and variabilities can be found in Supplementary Figure 12 and Note 12.

The size distributions are also estimated for each flight phase (Fig. 4c). The largest particles are during take-off (GMD = 36.96 ± 0.94 nm), followed by climb-out, climb and cruise. The particles are much smaller during descent (GMD = 24.26 ± 0.84 nm), taxi (GMD = 24.57 ± 0.95 nm) and approach (GMD = 27.18 ± 1.09 nm), when the thrust is relatively low. The GSDs of different flight phases are all comparable, within a narrow interval between 1.80 and 1.85, which are much larger than those of individual aircrafts (mostly between 1.4 and 1.8, Fig. 1b) due to the integration of the BC particles from various aircrafts and flight conditions.

In the AEDT inventory, the GMD was estimated to range from 11 to 79 nm, and 38 nm for the nominal mass and number emission scenarios[46], which agrees reasonably with our estimations. IPCC (Intergovernmental Panel on Climate Change) adopted a size distribution of much smaller particles, with a GMD of 11.8 nm and GSD of 2, to evaluate the influences on climate[16], which could lead to overestimation of particle number emissions. The research of ACCRI[49] utilized a GMD of 60 nm and a *GSD* of 1.6 to investigate the climate influences. The diameter is significantly larger than our estimation and the available measurements (Fig. 1a). The usage of different particle size distributions and the corresponding number emission amounts may potentially impact the assessment of the climate influences.

**Emissions in different regions**. Most of the particles (about 91%) are emitted in the Northern Hemisphere, while only 9% in the Southern Hemisphere (Fig. 3). The emissions are divided into different regions (Supplementary Figure 13). Figure 5 shows the shares of the BC number, mass emissions and fuel consumptions

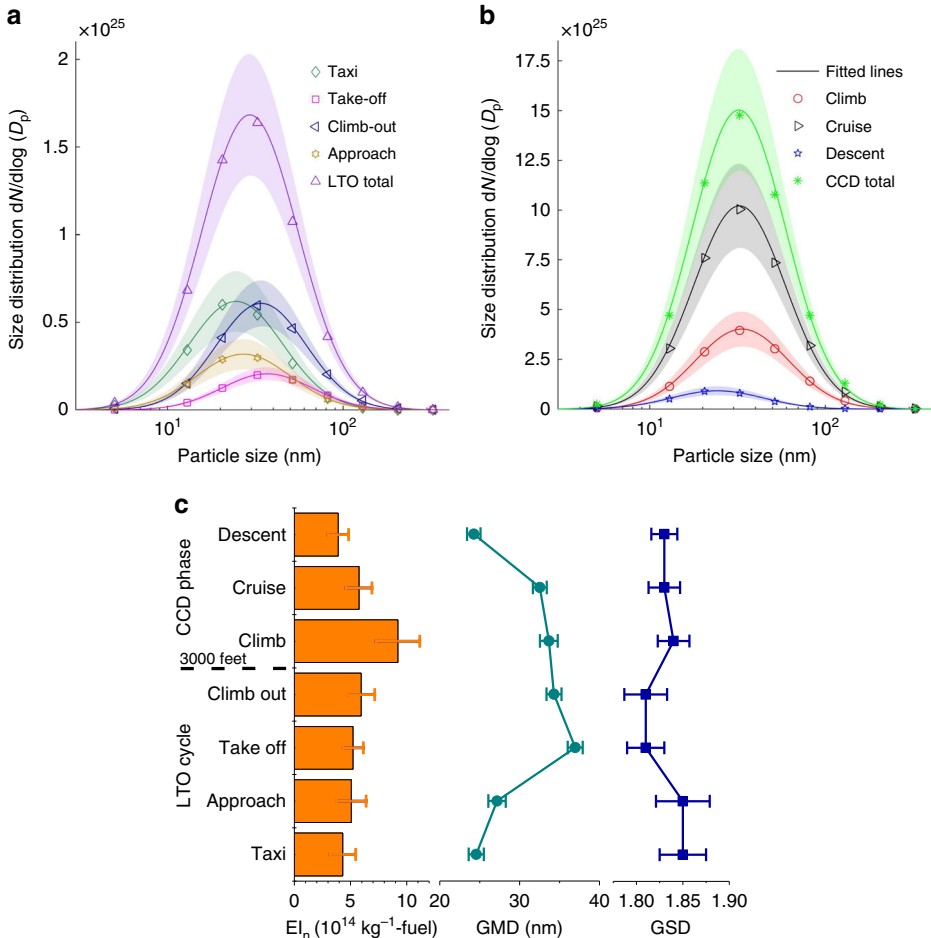

**Fig. 4** Size characteristics of global aviation black carbon emission. **a** Global black carbon (BC) particle size distributions during LTO cycles; **b** BC particle size distributions during CCD phases and **c** BC particle number emission index $EI_n$(BC) (defined as the number of the emitted BC particles due to the combustion of one kilogram of fuel ($kg^{-1}$-fuel)), geometric mean diameters GMDs and geometric standard deviations GSDs of the seven sub-phases of flights. The symbols in **a** and **b** represent the mean values of the MC estimated emissions in different size bins. The solid lines are the fitted lognormal distributions. The shaded area and error bars represent the standard deviation of the 100 fitted distributions in the Monte Carlo (MC) simulations

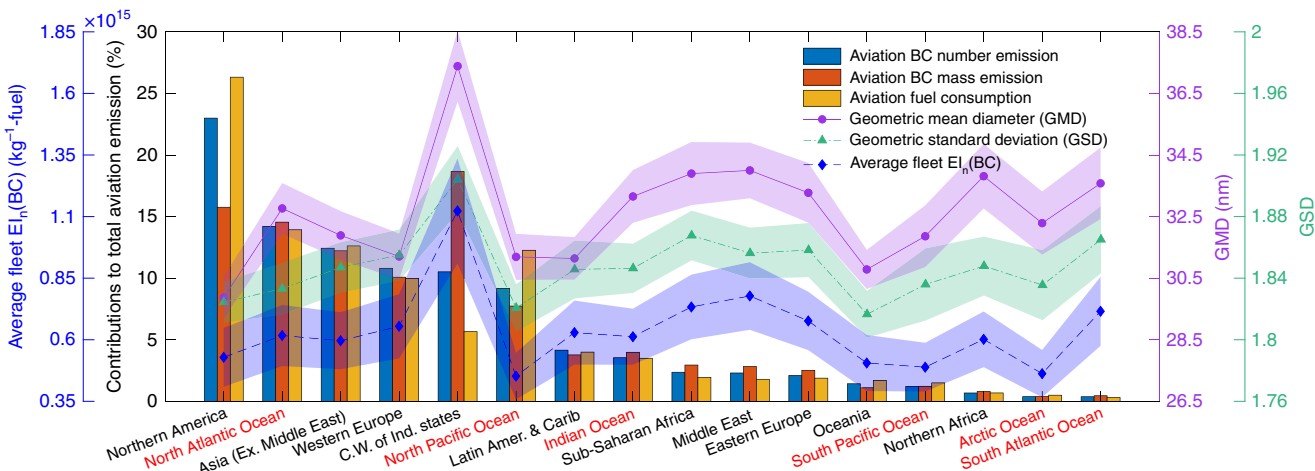

**Fig. 5** Regional characteristics of aviation black carbon emissions. (Bars) Contributions of different regions to the total aviation black carbon (BC) number and mass emissions, fuel consumption; (lines) the symbols indicate the BC particle number emission indices $EI_n$(BC), geometric mean diameter GMD and geometric standard deviation GSD of different regions, and the shaded areas are the uncertainties estimated by 100 Monte Carlo simulations

of these regions. Northern America contributes the most in terms of BC particle number emission of civil aviation, with about 23% of the global emission. The aircrafts traveling above Northern America also consume the largest proportion of aviation fuel (26.3%). However, the average $EI_n(BC)$ is only about $(5.3 \pm 1.19) \times 10^{14}$ kg$^{-1}$-fuel, which is below the global average $(6.06 \pm 1.18) \times 10^{14}$ kg$^{-1}$-fuel, due to the wide usage of modern aircrafts. The BC particles in this region have the smallest GMD ($29.9 \pm 0.68$ nm). In terms of the BC mass, Northern America contributes about 15.6%, ranked as No. 2 after the C.W. of Ind. States.

The highest $EI_n(BC)$ is found in the C.W. of Ind. States, where the $EI_n(BC)$ is $(1.1 \pm 0.21) \times 10^{15}$ kg$^{-1}$-fuel, twice of that in Northern America. This region also has the largest BC particles, with a GMD of $37.4 \pm 1.15$ nm and GSD of $1.9 \pm 0.022$. The C.W. of Ind. States contributes the most to the BC mass emissions, about 18.7%, while consuming only about 5.7% of the total aviation fuel, due to utilization of some high-emission aircrafts. Despite the highest contribution in mass, the particle number emitted above this region only contributes 10.5% to the global emission, ranked as No.5 after Northern America, North Atlantic Ocean, Asia and Western Europe, due to the large particle size.

The North Pacific Ocean has the lowest $EI_n(BC)$, $(4.5 \pm 0.95) \times 10^{14}$ kg$^{-1}$-fuel, because most of the flights are in the cruise phase. The North Atlantic Ocean contributes the most to both the BC number (14.2%) and mass (14.5%) emissions among all the ocean regions due to the busy air routes between Europe and Northern America.

**Emissions per passenger-distance**. The global aviation emission per passenger-distance was calculated using the total passengers of air traffic in 2005[50]. The lower and upper boundaries of the aviation mass emission were, respectively, calculated based on the FOA3 and FOX methods. The boundaries of the number emissions were the standard deviations based on 100 Monte Carlo simulations. The emissions of road vehicles from Durdina et al.[44] were used for comparison, which utilized the data for various engine technologies in the literature[51,52], and assumed two passengers per car and 30 passengers per bus.

The global aviation BC mass emission per passenger-distance is comparable to the road emissions from the diesel buses and gasoline cars, but notably lower than the emissions from diesel cars; the aviation mass emissions are much higher than the road emissions if diesel particulate filters (DPF) are utilized, as shown in Fig. 6. Regarding the particle number, the aviation emission

becomes significantly higher than both the diesel buses and gasoline cars, and comparable to the emissions of diesel cars without DPF. It should be noted that the standard measurements of on-road vehicle emissions are for particles larger than 23 nm[51] instead of about 10 nm used for the aircraft engines. The results are consistent with the characteristics of modern gas turbine engines which produce low mass emission but relatively high number emission due to the small GMD of the emitted particles[44].

**Comparison to surface emissions**. We assessed the magnitude of the aviation BC emissions by comparison with other global surface emissions[53] based on the ECLIPSE V5 (Evaluating the Climate and Air Quality Impacts of Short-Lived Pollutants)[54] scenario. The size-resolved particle number emission inventory contained the continental anthropogenic primary particle number emissions on the ground (excluding open burning and aviation emissions) from 11 main sectors, including surface transportation, power production, etc. The particles were divided into 8 size bins from 3 to 1000 nm[53]. There was no particle number data for 2005, so the data for 2010 were utilized instead. The gridded particle number emissions in the NetCDF files[53] were utilized, and were assigned into the regions defined in Supplementary Figure 13. The total anthropogenic BC particle number emission on the ground is about $8.6 \times 10^{27}$ per year. It should be noted that in the ECLIPSE inventory, the GMDs of BC particles were assumed to be larger than 50 nm, a reasonable estimation for diesel vehicles, which are the dominant contributor to traffic emission[55], but it may underestimate the particle number emissions from gasoline vehicles. The above anthropogenic emissions exclude the contributions from open burning (grassland, woodland and forest fires), for which the BC particle number emission was estimated to be $1.1 \times 10^{27}$ per year. The details of this estimation can be found in Supplementary Figure 14, Supplementary Table 9 and Note 13.

As shown in Fig. 7, at the global scale, the BC number emissions from air traffic are equivalent to about 3.6% of the surface transportation emissions, about 1.3% of all the anthropogenic BC containing particles emitted on the ground and about 1.1% of all the emissions including the open burning contribution. However, the contribution of aviation to the BC mass emission is much lower, only equivalent to about 0.6% of the surface transportation[56], 0.13% of all the surface anthropogenic

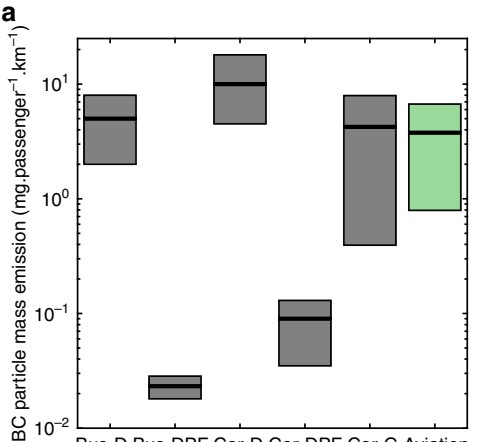
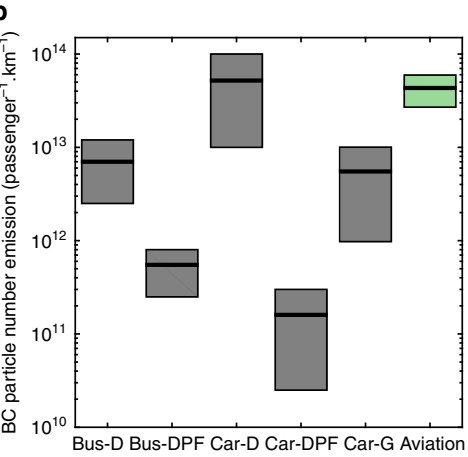

**Fig. 6** Black carbon emissions of different transportation means. **a** The black carbon (BC) mass emissions and **b** BC particle number emissions per passenger-km among different transportation means: Bus-D (with a diesel engine), Bus-DPF (with a diesel engine and a particle filter), Car-D, Car-DPF, Car-G (with a gasoline engine), and aviation

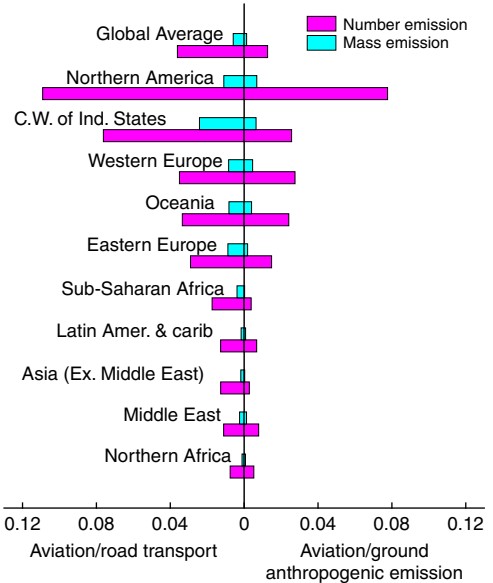

**Fig. 7** Aviation and ground black carbon emissions. (Left) The ratios between the black carbon (BC) mass and particle number emissions by aviation and the emissions by road transport; (right) the ratios between the BC mass and particle number emissions by aviation and the emissions by the total anthropogenic sources on the ground. The ground anthropogenic emission excludes aviation emission

emissions[56], and 0.1% of the total emissions including the open burning contribution.

There are significant spatial heterogeneities. In Northern America, the number of BC particles from air traffic is equivalent to nearly 11% of the surface transportation emission, making aviation a considerable contributor to the BC particle number emissions, but the aviation BC mass emission is only equivalent to 1% of the surface transportation emission, which is 10 times lower than the ratio of the particle number. The high contribution of aviation to the number emission in Northern America is due to the busy air traffic. According to World Bank[50], Northern America accounts for about one third of the total passengers carried by air traffic. On the other hand, the surface emission in Northern America is relatively low, due to the wide adoption of new emission abatement technologies and the high percentage of gasoline cars[53]. The ratio dramatically decreases to around 1% in Asia, Southern America, Middle East, and Africa, where the aviation industries are relatively underdeveloped and more particles are emitted from surface transportation due to the higher sulfur content of the fuel and less usage of new emission abatement technologies[53]. The C.W. OF IND. STATES has the highest ratio (about 2.4%) between aviation and road BC mass emissions mainly due to the relatively larger and heavier particles. In Europe and Oceania, the aviation BC number and mass emissions are, respectively, equivalent to about 3.5 and 3.3% of the road emissions, which are close to the global average. The relatively low ratio in Europe is partially due to the high road emissions[53], because of a larger proportion of diesel cars. Most of these particles from aviation are emitted in the upper troposphere and lowermost stratosphere, while the road transport emissions are emitted at the surface, so the contribution of aviation emissions to the surface BC concentration should be much smaller than that implied by these ratios.

## Discussion

We have developed a size-resolved BC particle number emission inventory by global civil aviation based on the recent measurements. The average BC particle number emission indices ($EI_n$) and size distributions (GMD and GSD) are estimated for all flight phases and different regions. The total BC number emission is about 2.5 times of the previous estimation in the AERO2k inventory[29]. The GMD and GSD are notably different from the values adopted in IPCC[16] and ACCRI[49] assessments. The usage of different particle size distributions and the corresponding number emission amounts may potentially impact the assessment of the climate influences, which needs further investigation. Compared to the total anthropogenic emissions, the aviation BC number emission constitutes a much more significant share than the mass emission due to the small ultrafine particles emitted from jet engines. The aviation BC number can be equivalent to as high as about 11% of the road emission in Northern America. The developed emission inventory provides new input data for models assessing the aviation impacts on global climate and air quality.

In the future, the BC mass ($EI_m$) and number ($EI_n$) emissions will be directly measured for the engine certifications according to the new ICAO standard, but the particle size measurement is not mandatory. There is also no plan for retroactive measurements of the currently certified engines[26]. The inventory has included most of the engines in use, which will continue to fly in the following decades. Most of the currently available relevant data and knowledge were integrated to make the estimations with our best effort for the time being, but considering the limited data, this emission inventory and the proposed methods have to be continually updated in the future when new data and knowledge are obtained.

The developed inventory contains the primary BC particles. Different from long-lived trace gases such as $CO_2$, airborne BC particle number and mass are not conserved with relatively short lifetime due to coagulation and deposition. In this study, we only utilized standardized measurements near the engine exit plane, because the aerosol dynamics in the long distance measurement depends on a lot of external factors, including meteorological conditions (e.g., humidity, temperature, and atmospheric stability), the dilution of the plume, and ambient pollution (e.g., ambient particle concentration). Further studies are needed to understand the transformation of the particles in the environment[57]. Sub-grid particle-cloud interaction models are also required to assess the climate impacts[13].

## Methods

**Conversion from mass-to-number emissions**. The BC mass-to-number conversion was based on the Fractal Aggregates (FA) approaches[36–38]. Recent studies showed that the sizes of the primary particles were relatively uniform in a single aggregate, but the primary particle sizes changed with the sizes of different aggregates[58,59]. Soot particles show self-similarity on different scales, so the fractal scaling law can be utilized to relate the fractal aggregate size and the number of primary particles ($n_{pp}$)[39,60,61].

$$n_{pp} = k \left( \frac{d_m}{d_{pp}} \right)^{D_m} \quad (1)$$

where $d_{pp}$ is the volume area equivalent primary particle diameter, $d_m$ is the mobility diameter of an aggregate, $k$ is a scaling prefactor and $D_m$ is the mass mobility exponent. Boies et al.[36] adopted a power–law relationship to fit the relation between $d_{pp}$ and $d_m$ for the BC particles from an aircraft gas turbine. As a result, the particle mass ($m_p$) for an aggregate of monodisperse, spherical primary particles can be expressed as[39]

$$m_p = C \cdot d_m^{\varepsilon} \quad (2)$$

During A-PRIDE experiments, the particle mass were directly measured by Centrifugal Particle Mass Analyzer (CPMA; Cambustion Ltd., Cambridge, UK) at fixed particle sizes and engine thrusts[39]. The particle mass increased from $10^{-21}$ to $10^{-18}$ kg with the increasing size from 20 to 150 nm, and the standard deviations were maximum 16.6% for the particles smaller than 40 nm and 11.0% for those

>40 nm. The exponent $\varepsilon$ (dimensionless unit) and prefactor $C$ (kg m$^{-\varepsilon}$) were estimated by fitting the power–law relationship between particle mass and size. The derivation of the relation and the physical meaning of the parameters are shown in Supplementary Note 14. The BC particle number-size distribution can be described using a lognormal distribution $EI_n \cdot n(d_m)$, where $EI_n$ is the particle number emission index (total particle number per kg-fuel), and $n(d_m)$ is the normalized lognormal distribution with GMD and GSD. The mass emission index $EI_m$ is expressed as follows

$$EI_m = \int_0^{\infty} m_p \cdot EI_n \cdot n(d_m) \mathrm{d}d_m \tag{3}$$

which leads to

$$EI_m = C \cdot EI_n \cdot GMD^{\varepsilon} \cdot \exp\left(\frac{\varepsilon^2 \cdot (\ln(GSD))^2}{2}\right) \tag{4}$$

The derivation can be found in Supplementary Note 15. Therefore, the BC particle number emission index $EI_n$ can be estimated based on the equation

$$EI_n = \frac{EI_m}{C \cdot GMD^{\varepsilon} \cdot \exp\left(\frac{\varepsilon^2 \cdot (\ln(GSD))^2}{2}\right)} \tag{5}$$

The GMD and GSD are the two important parameters for the conversion.

**Estimation of geometric mean diameter and standard deviation.** Recent studies attempted to estimate GMD based on its correlation with the thrust or fuel flow rate[37] (Supplementary Note 2). However the ground and the cruise measurements do not follow the same correlation and the data were highly scattered (Supplementary Figure 1(a) and (d)).

We developed a correlation to estimate the GMDs based on $T_3$ and $EI_m(BC)$. The combustion temperature controls fuel fragmentation, aromatic ring formation and generation of large polycyclic aromatic hydrocarbon (PAHs), which subsequently nucleate and carbonize to form BC particles[62]. High temperature is accompanied by high fuel–air ratio (FAR), which increases the local PAHs concentration and suppresses the oxidation of the generated BC particles. Transmission electron microscopy (TEM) measurements of the BC particles emitted by a CFM56-7B26/3 engine directly revealed that the primary BC particles were nearly spherical and their sizes became larger with increasing $T_3$[7].

The GMDs of BC particles are generally positively correlated with $T_3$, but with a slight negative correlation at low temperatures as shown in Supplementary Figure 1 (b), which was also observed in the experiments with simulated flight altitudes from sea level to 15.2 km[44,63] (Supplementary Note 3). The ANOVA tests (Supplementary Table 10) indicated that the correlation between GMD and $T_3$ has no statistically significant difference at various altitudes (Supplementary Figure 15 and Supplementary Figure 16). The correlation can be depicted by a quadratic equation. The increasing trend is mainly attributed to the growth of the primary BC particles as discussed above and also the effective agglomeration due to the violent turbulence mixing and the intensive motion of the primary particles triggered by the high temperature. In the $T_3$ range of about 300–500 °C, GMD decreases slightly with increasing $T_3$ which may be explained by the reduced agglomeration due to the decreasing residence time of the combustion product.

However, the correlation based on $T_3$ alone is insufficient to describe the GMDs measured from different engines, as it systematically overestimated (e.g. for CF6-80 engines) and underestimated (e.g., for JT8D-219 engine) the GMDs of certain engines (Supplementary Figure 1(b)). $EI_m(BC)$, as an engine specific parameter, depends on the engine design, which can either promote the BC generation by utilizing high local FAR, or suppress it by consuming the incomplete combustion product in lean fuel conditions such as in the Rich–Quench–Lean combustor. The GMD increases monotonically with $EI_m(BC)$ (Supplementary Figure 1(c) and Note 4), which is hypothetically mainly caused by coagulation. A cubic equation is utilized to depict the overall trend. The correlation of $EI_m(BC)$ is combined with that of $T_3$ to modulate the GMD, resulting in the form of Eq. (6).

$$GMD = \left(A_1 \cdot (\lg(EI_m))^3 + A_2 \cdot (\lg(EI_m))^2 + A_3 \cdot \lg(EI_m) + A_4\right) \cdot \left(\left(\frac{T_3}{1000}\right)^2 + A_5 \cdot \frac{T_3}{1000} + A_6\right) \tag{6}$$

The proposed method successfully captures the unique changes of GMD of the BC particles generated in a double annular engine (Supplementary Note 16 and Supplementary Figure 17), even though the method is developed based on the engines with a single annular combustor. The results demonstrate the necessity to include both $T_3$ and $EI_m(BC)$ as the predictors for GMDs. They also show the potential abilities of the proposed method to estimate the BC particle size distributions and the number emissions for significantly different engine designs and operating conditions.

TEM measurements[7] indicated that standard deviations of the primary particle sizes increase with $T_3$, and the distribution width of agglomerated BC particles could also change with $EI_m(BC)$ due to the abundance of primary BC particles available for the growth into a vast range of sizes. The plot between GSD and $T_3$ is highly scattered, but there is a good dependence of GSD on $EI_m(BC)$ (Supplementary Figure 4 and Note 8), which is also applicable to the double annular engine (Supplementary Figure 18 and Note 16), indicating that coagulation may be the dominant factor for the width of the BC particle distribution, therefore the relation with $EI_m(BC)$ (Eq. 7) is adopted to fit the data.

$$GSD = 0.1055 \cdot \lg(EI_m) + 1.89 \tag{7}$$

In order to estimate the size-resolved emissions, the emitted BC particles from each aircraft were divided into 16 size bins (Supplementary Table 7). For Eq. (6), the Levenberg–Marquardt nonlinear least squares algorithm was used to iteratively minimize the sum of the squares of the deviations between the observations and the predicted values. More details about the regression can be found in Supplementary Note 5 and Note 17. For Eq. (7), the least square linear regression was utilized to estimate the parameters.

**Dataset for model training.** The coefficients of the GMD-$T_3$&$EI_m$ and GSD-$EI_m$ relations were obtained by fitting the measured engine emission data from the literature[14,35,42]. The data summary is shown in Supplementary Note 1 and Supplementary Table 1 and Table 2. There were 53 available datasets, including seven types of aircraft engines from different manufacturers to cover a wide range of BC emissions. The fuel flow rate, $EI_m$, GMD, and GSD were simultaneously measured. The $EI_m$ ranged from 0.26 to 260 mg kg$^{-1}$-fuel with the thrust levels between 3 and 100%. The GMDs were from 14.7 to 40.9 nm and the GSDs located between 1.46 and 2.03. All the ground measurements were conducted using the new standard[32] compliant systems. The recent measurements of cruise emissions[14] were also included. The $T_3$ was estimated following the FOX method[26], which is briefly introduced in Supplementary Note 18.

The particles in the measurement system can deposit on the walls mainly due to diffusion and thermophoresis, leading to particle loss. The line loss depends on the line length, geometry, temperature gradients and particle size. The GMD measured by the instrument is normally larger than that at the engine exit plane due to higher losses of the smaller particles. The new measurement standard[32] attempts to minimize the losses through the system, but particle loss is still a major uncertainty. The size-dependent line loss functions were determined to correct the size distribution in many measurement campaigns, e.g., A-PRIDE[44], APEX (Aircraft Particle Emissions eXperiment)[64] and ACCESS (Alternative Fuel Effects on Contrails and Cruise Emissions Study)[14]. The corrections aim to recover the distribution at the engine exit plane for the ground measurements or at the inlet of the measurement system for the cruise measurements. In this study, we utilized the corrected GMD and GSD data, but it should be noted that the uncertainties in these correction methods may influence the results here.

The dataset from APEX[64] was utilized as independent data to validate the fitted correlation. The dataset contains 84 data points measured for 11 power settings from 4 to 100% of CFM56-2-C1 engines. More detailed information about the dataset can be found in Supplementary Note 7. Most of the size distribution and indices have already been corrected for the particle loss in the measurement system using size-dependent loss functions, except the data in A-PRIDE 4[35], which were corrected in the present study by the line loss function in Durdina et al.[65] for the same measurement system.

**Black carbon mass emission indices.** In this study, we adopted the BC mass emission indices provided by the EEA emission inventory guidebook[40], which was compiled using the Advanced Emission Model (AEM) for nearly 300 different types of aircrafts during LTO and CCD phases at various flight distances, altitudes and speeds. The dataset also contained the fuel consumptions and the gaseous emission indices. In the AEM model adopted by EEA, the BC emissions were estimated by the SN dependent FOA3 method[24], but the missing SN data in the ICAO Engine Emissions Databank[19] were estimated based on the algorithm developed by German Aerospace Center (DLR) and the cruise emissions were also corrected for the atmospheric environment at high altitudes[66]. In the 2005 inventory, there were 214 different types of aircrafts (76 types of engines). Among them there were 75 types of aircrafts (16 types of engines) without SN data in the EDB, and another 39 types of aircrafts (12 types of engines) with SN but measured before 1990.

The flight conditions (e.g., fuel flow rate, speed and altitude) influence the BC mass-to-number conversion. In order to better estimate the number emissions, the EEA mass emission indices for LTO and CCD phases were further divided into seven sub-phases, taxi/take-off/climb-out/approach for the LTO cycle and climb/cruise/descent for the CCD phase (Supplementary Note 19 to Note 23). For the LTO cycle, the duration and fuel flow rates followed the ICAO default settings. The typical flight speeds of LTO sub-phases (Supplementary Figure 19) were statistically analyzed based on 901 records of Flight Data Recorder (FDR)[67] (Supplementary Figure 20), showing the mean speeds were about 20 m s$^{-1}$ for taxi,

60 m s$^{-1}$ for take-off, 90 m s$^{-1}$ for climb-out and 80 m s$^{-1}$ for approach, with the uncertainties of 10 to 20 m s$^{-1}$.

In order to analyze the CCD sub-phase conditions, more than 13,000 records of the Automatic Dependent Surveillance-Broadcast (ADS-B), containing 1500 regular flight routes operated by American Airlines and Lufthansa on 9 different days (Supplementary Figure 21), were collected from the OpenSky network[68] [https://opensky-network.org/]. The flight speed, fuel flow rate, fuel consumption for CCD sub-phases were estimated based on the combination of EEA, FDR and ADS-B data (Supplementary Figure 22): the typical flight speed (Supplementary Figure 23) and normalized fuel flow rates (Supplementary Figure 24, taking the flow rate during take-off as the unit) for the CCD sub-phases were first, respectively, estimated using the ADS-B and the FDR data; then the values were modified for each aircraft-route pair constrained by the separated durations of CCD sub-phases (Supplementary Figure 25), the route distance (Supplementary Note 21) and the total fuel consumption (Supplementary Note 22).

The results showed that the cruise velocity (about 230 m s$^{-1}$) was the largest one, followed by climb phase (about 200 m s$^{-1}$) and descend phase (about 170 m s$^{-1}$). The uncertainties were about 10 m s$^{-1}$ for climb and descend, and larger for cruise about 20 m s$^{-1}$. The uncertainties for fuel flow ratio were much lower, only about 0.05 for cruise and descend, and about 0.1 for climb. The fuel consumptions and BC mass emissions in the EEA dataset were divided into the seven sub-phases based on the flight conditions, with the total amounts matching the original EEA data. The detailed processing method is described in Supplementary Note 19 to Note 23.

**Scheduled civil flight data**. The BC number emission inventory was compiled based on the open source Aviation Emission Inventory Code (AEIC)[23,26,41]. The code was modified to utilize the EEA dataset and the EI$_n$(BC) data converted from EI$_m$(BC), which were not in the original version. The OAG (Official Airline Guide) 2005 scheduled flight data in the original AEIC were utilized to estimate the emissions. The dataset contained the information of more than 27 million scheduled flights among 2572 airports around the world in the year 2005. The development of the inventory was compliant with the Tier 3 method for aviation emissions in the EEA guidebook[40], which required the origin and destination (OD) data of a flight. The great circle distance (shortest path between two airports) was usually adopted to estimate the flight distance. However, studies[69] showed that compared to the optimal path, the actual flight trajectories were ~4%, 12%, and 5% longer in Europe, Northern America and North Atlantic, respectively. In this study, it was assumed that the flight distance was longer than the great circle distance by 9.9% on average. The flight path was not modified due to the lack of information about the actual trajectories. The uncertainties of the flight trajectory length were not included in the current analysis.

The ambient temperature, pressure, and sound speed at the flight altitude were estimated based on the International Standard Atmosphere (ISA)[70], which were required for the calculation of $T_3$ to include the compression effects by the flight speed. We keep the symbol $T_3$ to be consistent with previous publications[26,44]. The engine data was from the ICAO Engine Emissions Databank[19]. The main input data for the number emission inventory development were summarized in Supplementary Table 11 and Note 24.

**Code availability**. The Aviation Emission Inventory Code (AEIC) was developed by the Laboratory for Aviation and the Environment in the MIT Department of Aeronautics & Astronautics and is available at [https://lae.mit.edu/codes/]. The modified code to include the black carbon particle number emissions is available from J.W. (jing.wang@ifu.baug.ethz.ch) upon reasonable request.

**Data availability**
The aviation black carbon mass emission, flight altitude, total CCD (Climb/Cruise/Descent) flight duration, and fuel consumption are available from EMEP/EEA air pollutant emission inventory guidebook 2016 "1.A.3.a Aviation – Annex 5 – Master emission calculator 2016" [https://www.eea.europa.eu/publications/emep-eea-guidebook-2016/part-b-sectoral-guidance-chapters/1-energy/1-a-combustion/1-a-3-a-aviation-1/view]. The Automatic Dependent SurveillanceBroadcast (ADS-B) data is available from the OpenSky network [https://opensky-network.org/]. The Flight Data Recorder (FDR) dataset is available from CrowdAnalytix [https://www.crowdanalytix.com/contests/predict-fuel-flow-rate-of-airplanes-during-different-phases-of-a-flight]. The aircraft engine data is available from ICAO Aircraft Engine Emissions Databank [https://www.easa.europa.eu/easa-and-you/environment/icao-aircraft-engine-emissions-databank]. The OAG 2005 scheduled flight data is available in the Aviation Emission Inventory Code (AEIC) developed by the Laboratory for Aviation and the Environment in the MIT Department of Aeronautics & Astronautics [https://lae.mit.edu/codes/]. The aircraft exhaust black carbon measurements are available in the literature as described within the paper and its supplementary information files. The developed size-resolved black carbon particle number emission inventory for global civil aviation is available in Supplementary Data. The results of Monte Carlo simulations and sensitivity tests are available from J.W. (jing.wang@ifu.baug.ethz.ch) upon reasonable request.

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

## Acknowledgements

We acknowledge the development of the open source Aviation Emission Inventory Code (AEIC) by the Laboratory for Aviation and the Environment in the MIT Department of Aeronautics & Astronautics. We acknowledge OpenSky network for the Automatic Dependent SurveillanceBroadcast (ADS-B) data and Honeywell for making the Flight Data Recorder (FDR) dataset publicly available.

## Author contributions

J.W. conceived the study. J.W. and X.Z. designed the development procedures. X.C. and X.Z. developed the calculation code. X.C. analyzed the ADS-B data. X.Z. and X.C. collected BC measurement data and carried out the calculations. X.Z., X.C., and J.W analyzed the results. X.Z. and J.W. wrote the paper.

## Additional information

**Competing interests:** The authors declare no competing interests.

