## [Peer Review File · Nature Communications]

A number-based inventory of size-resolved black carbon particle emissions by global civil aviation

Xiaole Zhang, Xi Chen and Jing Wang

Overview

Zhang et al (2018) have identified a major shortcoming in currently available aviation emissions' inventories, namely the absence of "size-resolved black carbon" number emissions. While BC mass emissions are available from numerous sources (e.g. AEIC [1]), number emissions are available from only 1 source (AERO2k), that too as an aggregated value with no size or distribution information. The authors' main goals are to develop (1) a method to predict number emissions index and (2) apply this to estimate a global emissions inventory. EI# is calculated using aggregate theory to convert the EIm to EI#. By assuming a lognormal size distribution, EIm can be converted to an EI# requiring the knowledge of only two parameters, the geometric mean diameter (GMD) and geometric standard deviation (GSD). These two parameters are calculated by correlating the GMD with EIm and T3, requiring 6 parameters and the GSD with EIm only, requiring 2 parameters. The authors find their correlations to predict GMD with an RMSE of 1.51 and 0.07 respectively. The ability to predict EI# is subsequently found to have an R^2 of 0.85 and the authors also attempt to quantify the uncertainty in these predictions.

The remainder of this paper focusses on calculating the global number emissions inventory and analyzing the results extensively. BC mass EIs were found in the "EEA emission inventory guidebook" compiled using the Advanced Emission Model. Typical flight times during the climb-cruise-descent phase were found by statistical analyzing 1500 regular routes by American Airlines and Lufthansa over 9 days. The author's total, annual mass of fuel burn and gaseous emissions is found to be in line with other published literature, and BC mass values are also within the range of variability in the literature as well. Their total BC number emissions are found to be 2.5 times larger than the single other published estimate. The paper goes on to provide a series of analyses on their AEIC simulations.

Initial assessment:

- This topic constitutes a major contribution to the field. A size-resolved BC particle emission's inventory would certainly be very useful, particularly for the climate assessment of contrails.
- The authors generate a correlation to predict the GMD and GSD, which seems to perform very well, however there is no mention of the physical basis of the algebraic form or how the regression is performed. In addition, there are concerns of overfitting (6 parameters for 53 points).
- The paper's focus seems to be more on the analysis of the AEIC simulations. While this is, of course, very interesting, the authors should spend more time discussing their correlation, particularly the underlying data on which it is based and the physical basis on which certain parameters are used and by which the algebraic form is chosen.
- The paper requires a lot of work to improve the English and some restructuring of individual sections could be useful to help improve the flow.

In its current state, I would reject this paper since the analysis raises many serious questions and the results are not communicated in a clear and structured manner. The paper needs significant work in order to be prepared for publication.

Major changes

- The correlations to predict GMD and GSD seem to have no physical basis. The text provides no background as to why EIm and T3 are chosen as dependent variables, nor is there any intuition behind the choice of the equation. The authors should spend at least 1-2 paragraphs providing this introduction. I.e. It makes physical sense that as the BC mass concentration increases, so too does the

size of the particle from a coagulation perspective. The BC mass concentration is related to the $EIm(BC)$ through knowledge of the fuel H:C content (see [2]).

- I am worried that the correlation to predict GMD has been overfit and the authors do not discuss this at all in the main paper. I would advise the authors to spend some time looking into the bias-variance trade-off as their model complexity increases. The information in the SI is currently not sufficient to provide insights into this question.
- Further discussion on the uncertainties should be included in Section 3.3. While this is a good beginning, there is no analysis on whether the residuals are normally distributed with mean zero and standard deviation to be found. There should also be some more discussion on the uncertainty of ε - the paper that the authors references [3] seems to show a much larger uncertainty on ε than the quoted value of 0.025. This value also varies with engine thrust setting, which the authors do not account for and would thus increase the uncertainty on ε further.
- The authors make a strong assumption that particle number emissions do not change from ground to cruise operations. This is based on the operation of a single engine (CFM56-2-C1) for which the authors have only shown cruise emission and no ground emissions' estimates. While the authors may be correct, the wording is far too strong to have "demonstrate that the correlation is applicable for different flight conditions". This certainly isn't a robust, statistical conclusion that the authors can justify, simply because there is not sufficient data!
- I would suggest the authors remove Sections 3.6 to 3.8 as they do not add to the discussion in this paper. Indeed, the major changes outlined above should more than compensate for the decreased length.
- It is very interesting to see the global BC size distributions, however there is no method to how these were derived and what the Monte Carlo simulations are for. It is unclear whether the authors have summed all the lognormal distributions for every aircraft over a full year or if these have simply calculated a mean GMD over all aircraft. I do not think it is a trivial task to calculate the global distribution of aircraft BC emissions.

Introduction

In general, the introduction is extremely weak given how useful this contribution is to the field. The introduction covers most of the main points, but individual paragraphs could be restructured. I suggest that the authors identify the flow of information in the introduction and primary purpose(s) of each paragraph in order to improve the message. Below are some specific comments in each case.

P1: - no mention of what BC particles constitute. Better to begin with air traffic growth, then discuss health impacts, then go into the uncertain contrail impacts and end with the highly uncertain indirect cloud impacts. There are also some more recent references that should be used (e.g. [4]).

P2: - PM needs to be defined (is this different from BC?). The inventories quoted are very old and newer inventories are now available (e.g. AEIC, AEDT), most of which include BC mass emissions. This paragraph should also be reversed with the aim of saying that number isn't available in most sources.

P3: - The AIR should be replaced with the latest ARP6320 [5] since this represents the latest updates from SAE E31. This paragraph should include a major discussion on measurement system losses that will affect all the results in the paper.

P4: - Why has volatiles been brought up? I think many points in this paragraph are incorrect. The EI-# definitely changes downstream due to coagulation etc (see e.g. Figure 2 in [6]). I'm surprised that cruise has little effect but it's hard to draw conclusions from just one measurement campaign. In general, I think this paragraph should be removed or the conclusions changed.

P5: - Again the flow of this paragraph should be inverted. The development of the method mentioned first and then the use of AEIC and finally the use of AEE. It is also a little unclear what the use of AEE is here. The

authors should be clear exactly what parameters estimated, rather than simply providing a few examples and how these mean values are used instead of the assumed values in AEIC.

Materials and methods

This section also needs substantial changes in order to improve the flow and key messages.

S2.1

P1: The opening sentence should be re-worded to simply mention that FA theory will be used. The sentences after Eq (1), beginning “Recent studies...” should be brought to the beginning of this paragraph. It is better to provide the reasoning behind the utility of a theory before introducing the formulas.

P3: The values for Eq (2), along with their confidence intervals should be quoted. Units should also be provided for the constants. What physical parameters is the prefactor C containing? It seems to hide density among other constants. What density does Abegglen et al (2015) [3] assume?

Eq 4: A citation for this derivation should be provided or included by the authors in the SI if not already published.

S2.2

P1: These papers do not use T3 or fuel flow rate to estimate GMD, rather it simple plotted on this basis since these are measurable performance characteristics. As far as I am aware, there are no published methods to estimate the GMD however this opening paragraph seems to suggest that there are. If there are, it is worth spending some time comparing them to your results later in the paper and introducing them here. This paragraph and subsequent ones in this section contain a lot of details about the measurement techniques and caveats. These should be separated into its own section. It would be nice to provide a table of the details similar to for example, Speth et al (2015) [7].

Eq 6: This equation seems to have appeared from nowhere. It is a complex form and no reasoning is provided for it. While a correlation is fine, there should be a physical basis to the form of the equation. In its current state, the equation has 6 parameters while there are 53 data points. This suggests some degree of overfitting and it would be interesting to see a bias-variance curve as certain coefficients are removed. The analysis can be left to the SI, but it should be certainly be mentioned that the authors have checked this.

P2: Is there any reason that GSD should not be independent of T3? Why is this surprising?

Eq 7: What type of regression is being conducted here? Are you using linear regression? Same with Eq 6? Presumably this is non-linear regression, however I think both equations can be re-written in the form suitable for linear regression? These are extremely important points to mention in the paper as the choice of regression cost function can alter the fitted results. In fact, conducting linear regression would allow the authors to present adjusted R^2 values for the actual fit, rather than in the parity plots shown later. (NOTE: The traditional R^2 will always increase as more parameters are introduced so this trend in the SI is not surprising at all. The adjusted R^2 takes into account the number of features used and so identifies if there is truly any performance increase with using additional parameters.)

P3: I'm not sure that there is sufficient evidence to suggest that number EIs do not change downstream. It seems coagulation is strong in the plume so I would suspect the GMD to increase and thus number EIs to reduce in order to conserve mass. Indeed, Wong et al (2014) [6] suggests that the number EI does decrease as the GMD increases. You mention that these measurements were taken at the exit plane however do the authors account for system losses? In addition, the cruise emissions could not possible be at the exit plane but some distance downstream. I don't know why sulfuric acid is being brought into this? Why are you discussing volatiles...?

S2.3

P2: The reasons for using the AEM and EEA guidebook should be more clearly stated – it is because there are missing SN values in the ICAO EDB. This is mentioned in the SI but should be brought here as well. Indeed, the authors do not mention how many engines in the 2005 inventory did not have SN entries within the EDB. There are many extremely old, outdated entries within the EDB that should not be counted. A better overview needs to be provided about the statistical analysis. E.g., what is the output of the statistics? Is it just the mean time and power in each mode? What's the confidence in these parameter estimates? Some of these details should be provided in the main paper rather than left in the SI under the results section.

S2.4

P1: You need to discuss what “re-developed version” means. What changes have been made and why? Do you have uncertainties in the error in flight trajectory length. How is the correction made? How would this affect the flight path?

P2: Is T3 referring to static or total temperature? I'm guessing total so Tt3 should be used.

Results and discussion

The sub-sections here should attempt to match the subsections in the Materials and Methods section. Thus, you should begin with the GMD and GSD correlations, then the “mass to number conversion”, before going into AEIC validation and finally the analysis of number emissions.

S3.1

Table 1: The table (either within the table or in the caption) should confirm the year for which these values are quoted. The EI(CO₂) in this paper's results seems to be higher than that for all other models. The value used by the author should be quoted. This BC number values should be removed from this table as this is not the purpose of this section.

P2: This paragraph should begin with the discussion on global values first, before talking about the spatial distribution. The method used in EDGAR should be discussed since the results in this study are most similar to them. Discrepancies with AEDT and AERO2k are also not discussed at all. Are there any other papers that suggest that the Doppelheuer and Lecht relationship underestimates cruise emissions? The reference provided [8] severely overfits the specified data and is also based on only a single engine. The final sentence of this is incorrect – using the Doppelheuer and Lecht relationship has not “offset” the FOX cruise estimate; it still seems to be twice as large as the next largest estimate.

S3.2

P1: I think it is bold to say that the “correlation is applicable for different flight conditions”. This is based on results for only a single engine so is not a robust conclusion. While it may be fine to use this in analysis for the time being, it should be mentioned that more data is required for different engines in order to be reliable. In addition, ground emissions of the CFM56-2-C1 have not been provided here and this would provide more justification on whether the GMD for this engine is predicted well from ground to cruise.

Table 2: Is there any reason that correlation coefficient is used instead of R².

P2: Why is the correlation coefficient less satisfactory? For this field, a correlation coefficient of 0.77 is extremely good, especially for a parameter where there is not great understanding on the driving forces. Why is the blended fuel likely to lead to different GSD? There is no discussion on why this could be an outlier.

S3.3

P1: The SULFUR 1-7 data seems to have some information on BC number distribution – why is this not used in the developing the correlations in the first place? Why are some of the data points used to develop the original correlation not used to compare predicted vs measured EI#? The authors only use “reliable” data, but there is no mention on what this means and how it was assessed. As a consequence, this instead sounds like the

authors have selected data to provide a higher fit. I am certain that this is not the case, but the reasons behind why other sources were ignored should be provided and it should also be mentioned which sources were ignored.

P2: The issues regarding the uncertainties analysis have already been brought up as a major focal point to improve.

P3: It is interesting that you mention that the uncertainties can be “as large as a factor of two due to the uncertainty particle sizes and the fractal parameters”. It would be interesting to see sensitivities plots to understand the contribution of each uncertain term to the overall uncertainty.

S3.4

P1: I would introduce the global number emissions estimate here rather than in Table 1.

P2: It's strange that you only bring up that AERO2k have particle number emissions' estimates as well. It would seem prudent to bring this up earlier in the methods or even introduction. You should also be more clear what you mean by “ratios”. Given that it is the only other model out there, you should spend more time discussing its method versus your own. Should “prediction interval” actually be “confidence interval” – you have confidence interval in the mean of an estimate but prediction interval if you wanted to estimate for future emissions, for example.

P3: You don't need to mention highest emissions in a particular grid square. Since you are calculating in lat/lon coordinates, this will be affected by the area of the grid cell. Stick to regions rather than grid cells.

P4: I'm a little confused by the discussion on lines 312-314. Surely larger particles at higher thrusts should decrease EI# for a fixed EIm? I think the point made here is not clear. In addition, this is a discussion that seems relevant for the method behind the correlation. You also mention number emissions from the AEDT inventory, but it's unclear why this is isn't in Table 1 then? Towards the end of this paragraph, you compare EIs for the 737-800 at cruise. Firstly, you should mention the engine as well as the aircraft and secondly, I think this should be in Section 3.3 where you validate your method. Indeed, you could have a separate section highlighting the performance at cruise.

S3.5

P1: It's unclear how the particle size distributions are integrated together. I don't think it is trivial to sum multiply different lognormal distribution together. Do you only account for the uncertainties in the mass to number conversion? It is surprising that it only takes 100 Monte Carlo simulations to converge, whereas the uncertainties calculated in Section 3.3 required 5000 runs.

S3.6 – S3.8

As discussed earlier, I believe these three sections should be removed. I do not think they add to the goals of the paper and the required changes/additions to improve this paper should compensate for the reduced length.

Reviewer #2 (Remarks to the Author):

This paper provides a novel method to estimate size-distributed particle emissions from aircraft, considering the different operating and flight conditions. The methods seem sound and the detail in the work is high. I think this is a very good piece of work that will be useful to the field. Particle number is very important, e.g. to cloud formation, and previous inventories have applied assumptions, without integrating all the knowledge available from the study of aircraft.

Authors compare aircraft emissions with those from other sectors. This is an interesting comparison but I don't think it is practically useful, because most aircraft emissions occur at altitude while most other primary emissions do not.

My one negative comment-- and this is a matter for the editor's decision-- is that despite the solid work done here, I am not sure the novelty and new insights are high enough for Nature Communications, and possibly a more specialty journal should be considered (Atmospheric Chemistry and Physics, Atmospheric Environment). I think that an investigation of size assumptions on cirrus impacts (for example) would follow on this work and would rate such novelty.

Reviewer #3 (Remarks to the Author):

When you compared the magnitude of the aviation BC emissions with other global surface emissions, is the global surface emissions evaluated (estimated) in BC number to be comparable with the aviation emissions?

Table of contents

Response to Referee #1	1
Response to Referee #2	36
Response to Referee #3	37

Response to Referee #1:

The authors appreciate the valuable comments by the reviewer. We carefully considered the comments and have revised the manuscript accordingly. The comments help us significantly to improve the manuscript. A point-by-point response is as follows.

Annotation:

- (1) The comments of the reviewer are shown in *italic font*;
- (2) The responses to the comments are shown in green color;
- (3) The explanations for the changes in the manuscript are shown in green with underlines;
- (4) The revised contents in the manuscript are shown in blue color;
- (5) The reference numbers in this response are based on the list at the end of this document, which are different from those in the manuscript.
- (6) The equation numbers in this response are different from those in the manuscript and supporting information. In this response, all the references for the equations use the numbers in the current context.

Major Changes

- 1. *The correlations to predict GMD and GSD seem to have no physical basis. The text provides no background as to why El_m and T_3 are chosen as dependent variables, nor is there any intuition behind the choice of the equation. The authors should spend at least 1-2 paragraphs providing this introduction. I.e. It makes physical sense that as the BC mass concentration increases, so too does the size of the particle from a coagulation perspective. The BC mass concentration is related to the $El_m(BC)$ through knowledge of the fuel H:C content (see [2]).*

[Response]: Thank you for pointing out this important problem. In the revised manuscript, we explained the physical basis of the correlation from the aspects of primary particle size and coagulation. Transmission electron microscopy (TEM) images showed the primary particle size increased with T_3 . The $El_m(BC)$ provided concentration information, therefore was utilized as an indicator for agglomeration. The rationales for the choice of the fitting equation were also provided. Please see the changes below.

[Changes]: In the revised manuscript, the explanations for the physical basis of the correlation were added into Section 2.2:

“2.2 Estimations of GMD and GSD

Recent studies attempted to estimate GMD based on its correlation with the thrust or fuel flow rate¹ (Section S2.1). However the ground and the cruise measurements do not follow the same correlation and the data were highly scattered (Figure S1 (a) and (d)).

We developed a correlation to estimate the GMDs based on T_3 and $El_m(BC)$. The combustion temperature controls fuel fragmentation, aromatic ring formation and generation of large polycyclic aromatic hydrocarbon (PAHs), which subsequently nucleate and carbonize to form BC particles ². High temperature is accompanied by high fuel-air ratio (FAR), which increases the local PAHs concentration and suppresses the oxidation of the generated BC particles. Transmission electron microscopy (TEM) measurements of the BC particles emitted by a CFM 56-7B26/3 engine directly revealed that the primary BC particles were nearly spherical and their sizes became larger with increasing T_3 ³.

The GMDs of BC particles are generally positively correlated with T_3 , but with a slight negative correlation at low temperatures as shown in Figure S1(b), which was also observed in the experiments with simulated flight altitudes from sea level to 15.2 km ^{4,5}. The correlation can be depicted by a quadratic equation. The increasing trend is mainly attributed to the growth of the primary BC particles as discussed above and also the effective agglomeration due to the violent turbulence mixing and the intensive motion of the primary particles triggered by the high temperature. In the T_3 range of about 300~500 °C, GMD decreases slightly with increasing T_3 which may be explained by the reduced agglomeration due to the decreasing residence time of the combustion product.

However, the correlation based on T_3 alone is insufficient to describe the GMDs measured from different engines, as it systematically overestimated (e.g. for CF6-80 engines) and underestimated (e.g. for JT8D-219 engine) the GMDs of certain engines (Figure S1(b)). $El_m(BC)$, as an engine specific parameter, depends on the engine design, which can either promote the BC generation by utilizing high local FAR, or suppress it by consuming the incomplete combustion product in lean fuel conditions such as in the Rich-Quench-Lean (RQL) combustor. The GMD increases monotonically with $El_m(BC)$ (Figure S1(c)), which is hypothetically mainly caused by coagulation. A cubic equation is utilized to depict the overall trend. The correlation of $El_m(BC)$ is combined with that of T_3 to modulate the GMD, resulting in the form of Equation (1).

$$GMD = \left(A_1 \cdot (\lg(El_m))^3 + A_2 \cdot (\lg(El_m))^2 + A_3 \cdot \lg(El_m) + A_4 \right) \cdot \left(\left(\frac{T_3}{1000} \right)^2 + A_5 \cdot \frac{T_3}{1000} + A_6 \right) \quad (1)$$

TEM measurements ³ indicated that standard deviations of the primary particle sizes increase with T_3 , and the distribution width of agglomerated BC particles could also change with $El_m(BC)$ due to the abundance of primary BC particles available for the growth into a vast range of sizes. The plot between GSD and T_3 is highly scattered, but there is a good dependence of GSD on $El_m(BC)$, indicating that coagulation may be the dominant factor for the width of the BC particle distribution, therefore the relation with $El_m(BC)$ as the parameter (Equation 2) is adopted to fit the data.

$$GSD = 0.1055 \cdot \lg(El_m) + 1.89. \quad (2)$$

In order to estimate the size-resolved emissions, the emitted BC particles from each aircraft were divided into 16 size bins (Table S7).”

- 2. I am worried that the correlation to predict GMD has been overfit and the authors do not discuss this at all in the main paper. I would advise the authors to spend some time looking into the bias-variance trade-off as their model complexity increases. The information in the SI is currently not

sufficient to provide insights into this question

[Response]: We agree with the reviewer. We conducted the bias-variance trade-off tests by randomly dividing the data of measured particle size distributions into the training (33 data points about 60%) and validation (20 data points about 40%) sets. One thousand different stochastic groupings were generated and calculated for the tests to avoid the influence of the grouping method. The results show that our GMD- T_3 & EI_m relation was near the turning point of the validation error curve, indicating that it achieves the balance between bias and variance and is not over-fitted. Please see Changes (2) and (3) below.

In addition to the bias-variance trade-off tests, the Aircraft Particle Emissions eXperiment (APEX) dataset for CFM56-2-C1 engines was also collected and utilized to evaluate the correlation as completely independent tests. The dataset contains 84 data points measured for 11 power settings from 4% to 100%. The predictions by the model had a high correlation with the measurements ($r=0.91$) and a relatively low error (RMSE = 3.33), which also indicated that the model is not over-fitted. Please see changes (1) and (4).

[Changes]:

(1) In Section 2.3, P2, the introduction about APEX dataset was added:

“The dataset from Aircraft Particle Emissions eXperiment (APEX)⁶ was utilized as independent data to validate the fitted correlation. The dataset contains 84 data points measured for 11 power settings from 4% to 100% of CFM56-2-C1 engines. More detailed information about the dataset can be found in Section S2.7 of SI. Most of the size distribution and indices have already been corrected for particle loss in the measurement system using size-dependent loss functions, except the data in A-PRIDE 4⁷, which were corrected in the present study by the line loss function in Durdina et al.⁸ for the same measurement system.”

(2) Section 3.1, P3, a new paragraph was added to discuss the bias-variance trade-off tests and the independent evaluation using APEX dataset:

“The bias-variance trade-off tests (Figure S2) indicate that both the training and validation errors simultaneously decrease, when the complexity of the model increases from GMD-fuel (two parameters) to GMD- T_3 & EI_m (six parameters). With further increase of the complexity from seven to ten parameters (Table S4), the training errors decrease very slowly and nearly remain constant, but validation errors begin to increase. The GMD- T_3 & EI_m relation is near the turning point of the validation error curve, indicating that it achieves the balance between bias and variance and is not over-fitted. The independent evaluation using the APEX dataset also shows the predictions have a high correlation with the measurements ($r=0.91$) and an error RMSE of 3.33 (Figure S3). The details of the tests are documented in Sections S2.6 and S2.7 of SI.”

(3) In the Supporting Information, the details of the bias-variance trade-off tests were added as Section S2.6 Bias-variance trade-off tests:

“S2.6 Bias-variance trade-off tests

In order to investigate the bias-variance trade-off of our models, all the data of measured particle size distributions were randomly divided into the training (33 data points about 60%) and validation (20 data points about 40%) sets. One thousand stochastic groupings were generated and calculated for the bias-variance trade-off tests to avoid the influence of the grouping method. Both the training and

validation datasets contained the data from all the engines.

Eight different models were utilized as shown in Table S4. The model complexity increased with the number of parameters ranging from 2 to 10. Models No.1 to No.4 were discussed in the main body, and model No. 4 was the one we used for calculation in our study. Models No.5 to No.8 were only used here to investigate the bias-variance trade-off. Only the complexity in the El_m term was increased by varying the polynomial order from 4 to 7. The model performance on the training dataset would substantially deteriorate if we also increased the complexity of the T_3 term, which was caused by the quadratic dependence of GMD on T_3 .

Table S4 Eight different models for fitting GMD data

No.	Name	Predictors	Number of parameters	Model form
1	GMD-Fuel flow rate	Fuel flow rate	2 (A_1, A_2)	$GMD = A_1(F_{flow}/F_{flow, max}) + A_2$
2	GMD- T_3	T_3	3 ($A_1 \sim A_3$)	$GMD = A_1(T_3/1000)^2 + A_2(T_3/1000) + A_3$
3	GMD- El_m	El_m	4 ($A_1 \sim A_4$)	$GMD = A_1lg(El_m)^3 + A_2lg(El_m)^2 + A_3lg(El_m) + A_4$
4	GMD- $T_3&El_m$	T_3, El_m	6 ($A_1 \sim A_6$)	$GMD = (A_1lg(El_m)^3 + A_2lg(El_m)^2 + A_3lg(El_m) + A_4) \times ((T_3/1000)^2 + A_5(T_3/1000) + A_6)$
5	GMD- $T_3&El_m(4)$	T_3, El_m	7 ($A_1 \sim A_7$)	$GMD = (A_1lg(El_m)^4 + A_2lg(El_m)^3 + A_3lg(El_m)^2 + A_4lg(El_m) + A_5) \times ((T_3/1000)^2 + A_6(T_3/1000) + A_7)$
6	GMD- $T_3&El_m(5)$	T_3, El_m	8 ($A_1 \sim A_8$)	$GMD = (A_1lg(El_m)^5 + A_2lg(El_m)^4 + A_3lg(El_m)^3 + A_4lg(El_m)^2 + A_5lg(El_m) + A_6) \times ((T_3/1000)^2 + A_7(T_3/1000) + A_8)$
7	GMD- $T_3&El_m(6)$	T_3, El_m	9 ($A_1 \sim A_9$)	$GMD = (A_1lg(El_m)^6 + A_2lg(El_m)^5 + A_3lg(El_m)^4 + A_4lg(El_m)^3 + A_5lg(El_m)^2 + A_6lg(El_m) + A_7) \times ((T_3/1000)^2 + A_8(T_3/1000) + A_9)$
8	GMD- $T_3&El_m(7)$	T_3, El_m	10 ($A_1 \sim A_{10}$)	$GMD = (A_1lg(El_m)^7 + A_2lg(El_m)^6 + A_3lg(El_m)^5 + A_4lg(El_m)^4 + A_5lg(El_m)^3 + A_6lg(El_m)^2 + A_7lg(El_m) + A_8) \times ((T_3/1000)^2 + A_9(T_3/1000) + A_{10})$

The models were only fitted based on the training dataset. Then the fitted relations were applied to the validation dataset to test the model quality, which was quantified by the root-mean-square errors between the model results and the data. Figure S2 shows the results of the bias-variance trade-off tests. The shaded area indicated the uncertainties obtained from 1000 different groupings. Both the training and the validation errors were high, when using the less complex models, e.g. GMD-Fuel flow rate, indicating the bias was large. When we increased the model complexity, the training and validation errors first simultaneously decreased, and then the training errors remained nearly constant with a slight drop, while the validation errors increased. The uncertainties of the validation errors became very large when more complicated models were utilized, which indicated that the model performance may be extremely poor for some situations. The variance became high for the models with high validation errors and low training

errors, which means the models were over-fitted. The GMD- T_3 & EI_m relation was near the turning point of the validation error curve. This model achieves the balance between bias and variance, so it was the optimal model based on the current dataset.

Figure S2 Results of the bias-variance trade-off tests.

(4) In the Supporting Information, the independent validation using APEX dataset was added as Section S2.7:

“S2.7 Independent validation using APEX dataset

In addition to the bias-variance trade-off tests, a new completely independent dataset is also included to investigate the model performance. The dataset is from Aircraft Particle Emissions eXperiment (APEX)⁶, containing 84 data points measured for 11 power settings from 4% to 100%. The APEX experiments measured the particle emissions from NASA’s DC-8 aircraft, equipped with four General Electric CFM56-2-C1 engines. The system utilized a heated sampling channel (300 °C) to remove volatile components and the line loss corrections were conducted. The BC particle mass, number and size distributions were simultaneously measured. Here, we only used the data for standard JP-8 fuel sampled 1 m behind the engine exit, where the non-volatile particles dominated the total emissions. The particle size distributions were measured by NASA Scanning Mobility Particle Sizing (SMPS) and Differential Mobility Spectrometer (DMS-500). The summary of the APEX data is shown in Table S1.

The estimated GMD, GSD and $EI_n(BC)$ using our model are compared with the APEX data which are not included in the model training. Figure S3 shows the comparisons. For the GMDs, the model results have a good correlation (0.91) with the measurements. The model generally overestimates the data with a RMSE of 3.33. For GSDs, the results are similar with the model training results shown in Figure 1 of the main body. The estimated particle number emission has a good agreement with the measurements, similar to those in Figure 2 of the main body.

It should be noted that the measurement uncertainties (horizontal error bars) are large. Most of the data utilized for the model training are from Aviation-Particle Regulatory Instrumentation Demonstration Experiment (A-PRIDE), which used a different system from the NASA APEX experiments. The results are reasonable considering the measurement uncertainties and the different measurement systems.

Through the bias-variance trade-off tests and comparison with the independent APEX dataset, we demonstrate that our model does not over-fit the data.

Figure S3 Comparison of the model results and APEX data.”

- 3. Further discussion on the uncertainties should be included in Section 3.3. While this is a good beginning, there is no analysis on whether the residuals are normally distributed with mean zero and standard deviation to be found. There should also be some more discussion on the uncertainty of ϵ the paper that the authors references [3] seems to show a much larger uncertainty on ϵ than the quoted value of 0.025. This values also varies with engine thrust setting, which the authors do not account for and would thus increase the uncertainty on ϵ further.

[Response]: Thanks for the suggestion. We agree with the reviewer. In the revised manuscript, we analyzed the residuals of the EI_n predictions separately for the entire data (cruise + ground), cruise data and ground data. The residuals were close to normal distributions. The means were at the magnitude of 10^{13} /kg-fuel. The biases of the means were within the estimated uncertainties, which were at the magnitude of 10^{14} /kg-fuel. Please refer to Change (1) and Change (2) below for more details.

About the uncertainty of ϵ , in the reference Abegglen et al. (2015)⁹, the particle mass were directly measured by Centrifugal Particle Mass Analyzer (CPMA; Cambustion Ltd., Cambridge, UK) at fixed particle sizes and engine thrusts. Based on the measurements, the standard deviations for the mass of the size- and thrust-selected particles were maximum 16.6% for the particles smaller than 40 nm and 11.0% for those larger than 40 nm. The uncertainties of the parameters ϵ and C in the reference significantly overestimated the uncertainties of the measured particle mass, which would lead to unrealistic results. We re-analyzed the data in the reference to determine the uncertainties of the mass data more accurately. Our adopted uncertainty of ϵ can better reconstruct the particle mass data with uncertainties of about $\pm 40\%$, which are larger than the measurements (11.0% to 16.6%). The overestimated uncertainties are acceptable, because only one type of engine was measured in Abegglen et al. (2015)⁹. Please refer to Changes (3), (4) and (5) for more details.

[Changes]:

(1) In Section 3.2, P3, discussions on the residuals were added:

“The residuals of the EI_n estimations separately for the entire data (cruise + ground), cruise and ground

data are close to normal distributions as shown in Figure S13. The means are at the magnitude of 10^{13} /kg-fuel. The biases of the means are within the estimated uncertainties, which are at the magnitude of 10^{14} /kg-fuel. More details about the residual analysis can be found in S7.2 of SI.”

(2) In SI, the detailed analysis of the residuals was added as “Section S7.2 Residuals of the EI_n estimation”:

“S7.2 Residuals of the EI_n estimation

Figure S13 shows the residuals of all the Monte Carlo runs separately for the entire data (cruise + ground), cruise and ground measurement data. The residuals are close to normal distributions. The mean of the residual for all the data is 0.84×10^{13} /kg-fuel with a 95% confidence interval between $(-0.16 \ 1.84) \times 10^{13}$. The mean residual is small compared to the BC particle number emission index which is normally at the magnitude of 10^{14} /kg-fuel. The standard deviation is 1.43×10^{14} /kg-fuel, the same magnitude with the emission index.

The EI_n data for the cruise and ground conditions are respectively over- and underestimated, evidenced by mean residuals of 7.82×10^{13} and -1.39×10^{13} /kg-fuel. The bias may be caused by both the experimental and modeling uncertainties, e.g. the flight parameters, atmospheric conditions and measurement technologies. The limited number of data points may also influence the results. The biases are within the uncertainties of the global mean $EI_n(BC)$ estimations, which have the standard deviations of 1.00×10^{14} /kg-fuel for LTO and 1.22×10^{14} /kg-fuel for CCD as shown in Table S8.

Figure S13 Residuals of all the Monte Carlo runs.”

(3) In Section 2.1, P3, the measured particle mass and uncertainties in Abegglen et al. (2015)⁹ were described:

“During A-PRIDE experiments, the particle mass were directly measured by Centrifugal Particle Mass Analyzer (CPMA; Cambustion Ltd., Cambridge, UK) at fixed particle sizes and engine thrusts⁹. The particle mass increased from 10^{-21} to 10^{-18} kg with the increasing size from 20 to 150 nm, and the standard deviations were maximum 16.6% for the particles smaller than 40 nm and 11.0% for those larger than 40 nm. The exponent ϵ (dimensionless unit) and prefactor C ($\text{kg m}^{-\epsilon}$) were estimated by fitting the power-law relationship between particle mass and size.”

(4) In Section 3.2, P2, the predicted uncertainties of particle mass were compared with the measurements:

“The standard deviation (SD) of ϵ was assumed to be 0.025, which reconstructed the particle mass data with uncertainties of about $\pm 40\%$, larger than the measurements (11.0% to 16.6%). The overestimated uncertainties are acceptable, because only one type of engine was measured⁹. The detailed justification for the adopted SD of ϵ is provided in S7.1 of SI.”

(5) SI Section S7.1, we provided the detailed justification for the adopted SD (0.025) of ϵ :

“The uncertainties of ϵ shown in Abegglen et al.⁹ were the 95% confidence interval, not the standard deviation. Besides, the uncertainties were significantly overestimated. In Abegglen et al.⁹, ϵ was estimated by fitting the power-law relationship between particle mass and size

$$m_p = C \cdot d_m^\epsilon, \quad (3)$$

The particle mass was directly measured by Centrifugal Particle Mass Analyzer (CPMA; Cambustion Ltd., Cambridge, UK) at fixed particle sizes and engine thrusts. Based on the measurements, the standard deviations for the mass of the size- and thrust-selected particles were maximum 16.6% for the particles smaller than 40 nm and 11.0% for those larger than 40 nm. We re-analyzed the data in the reference to determine the uncertainties of the mass data more accurately.

Figure S12 Comparisons between the measured uncertainties of particle mass in Abegglen et al.⁹ and the reconstructed uncertainties respectively by the parameters in Abegglen et al.⁹ and the adopted uncertainty of ϵ (0.025).”

Figure S12 shows the measured particle mass and uncertainties (error bars) for 33%, 67% and 105% thrusts from Abegglen et al.⁹. The particle sizes are between 20 to 150 nm. The mass uncertainties calculated using the ϵ data in Abegglen et al.⁹ are represented by the dashed lines, which are far from the measured data showing that the uncertainties are dramatically overestimated. The shaded area is the results estimated using an uncertainty of 0.025 for ϵ . The estimated uncertainties are close to those of the

measurements.

According to Abegglen et al.⁹, the variation of prefactor C is very small, within about 5%. As a result, for a given particle size (d_m), the change of mass caused by the parameter uncertainties is

$$\frac{m_p}{E(m_p)} = \frac{C}{E(C)} d_m^{\varepsilon - E(\varepsilon)} \approx d_m^{\Delta\varepsilon}, \quad (4)$$

where $E(m_p)$ and $E(\varepsilon)$ are the means of particle mass and ε . For the particles with an electrical mobility diameter (d_m) between 20 to 150 nm, the changes of particle mass are about $\pm 40\%$ with the ε uncertainty utilized in this study ($\Delta\varepsilon=0.025$). These estimated particle mass uncertainties are already larger than the measurements (11.0% to 16.6%). We assume that it is reasonable to use larger uncertainties for particle mass considering the measurements are only for one type of engine. However, the changes of particle mass can be as large as -90% to 2000% if the uncertainties of ε (0.04~0.3) in Abegglen et al.⁹ are utilized. The values are far away from the measurements, leading to unrealistic results. As a result, the standard deviation of ε was assumed to be 0.025.”

- 4. *The authors make a strong assumption that particle number emissions do not change from ground to cruise operations. This is based on the operation of a single engine (CFM56-2-C1) for which the authors have only shown cruise emission and no ground emissions’ estimates. While the authors may be correct, the wording is far too strong to have “demonstrate that the correlation is applicable for different flight conditions”. This certainly isn’t a robust, statistical conclusion that the authors can justify, simply because there is not sufficient data!*

[Response]: We agree with the reviewer and have deleted the sentence in the revised manuscript.

We have included the ground measurements for CFM56-2-C1 as a new completely independent dataset in the revised manuscript to investigate the model performance, as discussed above. Full validation of the model still requires more data, therefore we remove the sentence.

[Changes]:

(1) In section 3.1, delete the sentence “demonstrate that the correlation is applicable for different flight conditions”.

(2) The ground measurements of CFM56-2-C1 from APEX experiments were added, please refer to the response to the Major Changes No.2

- 5. *I would suggest the authors remove Sections 3.6 to 3.8 as they do not add to the discussion in this paper. Indeed, the major changes outlined above should more than compensate for the decreased length.*

[Response]: Most of the changes made above are related to the methods. However, according to the requirements of the journal, substantial efforts should be spent on discussion and interpretation of the results, and “Methods should be written as concisely as possible”. We would like to keep these sections, since they provide direct impression on the general characteristics of the emission inventory and may reach a broad audience. We think these new results, especially the size-resolved gridded number emission data would help raise the interests of other researchers to investigate and assess the influences

of aviation BC particle emission on climate, environment and public health, which is one of the most important goals for the development of the inventory.

- 6. *It is very interesting to see the global BC size distributions, however there is no method to how these were derived and what the Monte Carlo simulations are for. It is unclear whether the authors have summed all the lognormal distributions for every aircraft over a full year or if these have simply calculated a mean GMD over all aircraft. I do not think it is a trivial task to calculate the global distribution of aircraft BC emissions.*

[Response]: The method to obtain the global BC size distributions was mentioned in the original manuscript and further clarification was included in the revised version. The emitted particles from each aircraft were divided into 16 size bins as shown in Table S7. The global inventory with 16 size bins was calculated by integrating the size-resolved emissions from all the aircrafts. Namely, we summed all the distributions for every aircraft over a full year.

The Monte Carlo simulations were utilized to estimate the uncertainties of the global emissions. The dots in Figure 4 (a) and (b) were the mean of all MC runs and the shaded area indicated the standard deviation of all the MC runs in each size bin.

The GMD and GSD of the global BC size distribution were obtained by fitting the particle numbers in the size bins into the lognormal distribution for each MC run, and the uncertainties of GMD and GSD were estimated from these MC runs.

[Changes]:

(1) In Section 2.2, P6, a sentence was added to explain that the particle number emission was divided into 16 different size bins:

“In order to estimate the size-resolved emissions, the emitted BC particles from each aircraft were divided into 16 size bins (Table S7).”

(2) In Section 3.5, P1 & P2, Particle size distribution, add explanations for method of the global BC distribution estimation:

“The developed global inventory has 16 size bins, which were calculated by integrating the size-resolved emissions from all the aircrafts...”

... The dots are the mean of all MC runs and the shaded area indicates the standard deviation of all the MC runs in each size bin. The GMD and GSD of the global BC size distribution were obtained by fitting the particle numbers in the size bins into the lognormal distribution for each MC run. The uncertainties of GMD and GSD were estimated from these MC runs. ...”

(3) In SI, add Table S7 for the boundaries of each size bin:

No.	1	2	3	4	5	6	7	8
Size (nm)	10	15.85	25.12	39.81	63.10	100	158.49	251.19
No.	9	10	11	12	13	14	15	16
Size (nm)	251.19	398.11	630.96	1000	1584.89	2500	3981.07	6309.57

Introduction

In general, the introduction is extremely weak given how useful this contribution is to the field. The introduction covers most of the main points, but individual paragraphs could be restructured. I suggest that the authors identify the flow of information in the introduction and primary purpose(s) of each paragraph in order to improve the message. Below are some specific comments in each case.

P1: - no mention of what BC particles constitute. Better to begin with air traffic growth, then discuss health impacts, then go into the uncertain contrail impacts and end with the highly uncertain indirect cloud impacts. There are also some more recent references that should be used (e.g. [4]).

[Response]: Thank you for your suggestions. The paragraph has been rewritten, please see the change below.

[Changes]: Introduction, P1, the paragraph has been rewritten as follows:

“Air traffic is estimated to increase more than twofold (in revenue passenger kilometers) in the next 20 years ¹⁰. The potential impacts of aviation emissions on public health, environment and climate have attracted more and more attentions ¹¹. On the ground, the ultrafine particles (less than 100 nm) from aircrafts significantly increase the particle number concentrations near airports ^{12,13}. Toxicological studies showed that there is strong evidence that the particle size, shape, surface and surface properties affect the particles’ toxicity ¹⁴⁻¹⁶. The black carbon (BC) particles, agglomerates of nearly spherical primary particles mainly composed of graphene lamellae ³, absorb sunlight, may also increase the cirrus cloudiness ¹⁷ and modify the optical thickness of already-existing cirrus ¹⁸. The aviation emissions mainly occur in the upper troposphere and lowermost stratosphere where high-altitude clouds influence the global climate by trapping outgoing long-wave radiation and reflecting solar short-wave radiation ¹⁹. It is shown that the formation of the contrail and the contrail cirrus is dependent on the BC number emission ²⁰, hence the number and size distribution of the aviation emitted particles are required to investigate such climate effects ²¹. The radiative forcing from the indirect effects of the aviation-induced cloudiness still remains highly uncertain ²².”

P2: - PM needs to be defined (is this different from BC?). The inventories quoted are very old and newer inventories are now available (e.g. AEIC, AEDT), most of which include BC mass emissions. This paragraph should also be reversed with the aim of saying that number isn’t available in most sources.

[Response]: PM includes both volatile and nonvolatile particles. In this study, we only consider the nonvolatile BC particles, so we delete PM here to avoid confusion. The newer inventories suggested by the reviewer have been added.

[Changes]:

(1) Introduction P2, at the beginning, add a sentence to state that the BC number emission is not available in most inventories:

“However, the BC particle number emission is not available in most of the existing inventories.”

(2) Introduction P2, delete PM and only use BC particle to avoid confusion:

“Due to the paucity of BC particle emission data in the ICAO (International Civil Aviation Organization) Emission Databank (EDB) ¹⁸...”

(3) Introduction P2, add introduction for the newer inventories containing BC mass emissions:

“The emergence of these methods led to the developments of BC mass emission inventories, e.g. AERO2k²³, AEIC²⁴ and AEDT²⁵. As the only currently available BC number inventory, AERO2k estimated the number-based inventory using simplified general characteristic of particle number emission from different engines.”

P3: - The AIR should be replaced with the latest ARP6320 [5] since this represents the latest updates from SAE E31. This paragraph should include a major discussion on measurement system losses that will affect all the results in the paper.

[Response]: The AIR has been replaced by the latest publication, please see Change (1). A new paragraph was added to discuss the system losses, please see Change (2).

[Changes]:

(1) The AIR has been replaced by the latest ARP6320.

“In order to standardize the measurements of aircraft exhaust BC, Aerospace Information Reports (AIR)²⁶ and Aerospace Recommended Practice (ARP)²⁷ have been issued by the Society of Automotive Engineers (SAE) E-31 Particulate Matter Committee. Matter Committee. Many new standard (ARP6320)²⁷ compliant measurement campaigns have been conducted²⁸ ...”

(2) Introduction P4, add a new paragraph to discuss the system losses:

“The particles in the measurement system can deposit on the walls mainly due to diffusion and thermophoresis, leading to particle loss. The line loss depends on the line length, geometry, temperature gradients and particle size. The GMD measured by the instrument is normally larger than that at the engine exit plane due to higher losses of the smaller particles. The new measurement standard²⁷ attempts to minimize the losses through the system, but particle loss is still a major uncertainty. The size dependent line loss functions were determined to correct the size distribution in many measurement campaigns, e.g. A-PRIDE⁵, APEX (Aircraft Particle Emissions eXperiment)⁶ and ACCESS (Alternative Fuel Effects on Contrails and Cruise Emissions Study)²¹. The corrections aim to recover the distribution at the engine exit plane for the ground measurements or at the inlet of the measurement system for the cruise measurements. In this study, we utilized the corrected GMD and GSD data, but it should be noted that the uncertainties in these correction methods may influence the results here.”

P4: - Why has volatiles been brought up? I think many points in this paragraph are incorrect. The EI-# definitely changes downstream due to coagulation etc (see e.g. Figure 2 in [6]). I'm surprised that cruise has little effect but it's hard to draw conclusions from just one measurement campaign. In general, I think this paragraph should be removed or the conclusions changed.

[Response]: We agree with the reviewer that the data are insufficient to draw the conclusions, so this paragraph is removed in the revised manuscript.

We have a different opinion about the coagulation of the BC particles. The evolution of the particle size in Figure 2 of Wong et al. (2014)²⁹ was caused by the condensation of organic vapor and sulfuric acid, not because of the BC-BC coagulation. We considered the BC-BC coagulation very weak within downstream exhaust due to the rapid dilution of the exhaust with the ambient air. APEX⁶ revealed that the nonvolatile particle size as well as number and mass emission indices (EI_n , EI_m) did not depend on downstream

sampling distance (plume age) based on the measurements at 1, 10, and 30 m downstream of the engine exhaust plane. However, we agree with the reviewer that the data are still insufficient to confirm the conclusion, so we decide to remove this part.

[Changes]: This paragraph has been removed.

P5: - Again the flow of this paragraph should be inverted. The development of the method mentioned first and then the use of AEIC and finally the use of AEE. It is also a little unclear what the use of AEE is here. The authors should be clear exactly what parameters estimated, rather than simply providing a few examples and how these mean values are used instead of the assumed values in AEIC.

[Response]: We agree with the reviewer. This paragraph has been rewritten; please see Change (1). More details about the usage of different databases were added and a new simplified flowchart for the inventory development was also included in the SI to illustrate what parameters were estimated and how the data were utilized; please see Change (2) to (5)

[Changes]:

(1) Introduction P5, the paragraph has been inverted according to the suggestion:

“In this study, a size-resolved BC particle number emission inventory was developed for the global civil aviation based on the recent measurements. The mass to number conversion was conducted using the fractal aggregate theory and the particle mass parameters obtained from Aviation-Particle Regulatory Instrumentation Demonstration Experiment (A-PRIDE) experiments⁹. The GMD and GSD of the particles were estimated with a new fitting correlation based on the combination of both the combustor inlet temperature (T_3) and the BC mass emission index (EI_m) of aircraft engines. The inventory development followed the European Environment Agency (EEA) air pollutant emission inventory guidebook³⁰ by utilizing the global scheduled flight dataset in 2005^{24,31,32}.”

(2) Introduction P5, more details about the usage of different databases were added:

“The BC mass emission, flight altitude, total CCD (Climb/Cruise/Descent) flight duration and fuel consumption were from EEA depending on the aircraft type and route distance. The Automatic Dependent Surveillance-Broadcast (ADS-B) data was utilized to develop the correlation between the flight durations of CCD sub-phases and the total CCD duration. The flight speed, fuel flow rate, fuel consumption for CCD sub-phases were estimated based on the combination of EEA, Flight Data Recorder (FDR) and ADS-B data (Figure S11): the typical flight speed and normalized fuel flow rates (taking the flow rate during take-off as the unit) for the CCD sub-phases were first respectively estimated using the ADS-B and the FDR data; then the values were modified for each aircraft-route pair constrained by the separated durations of CCD sub-phases, the route distance and the total fuel consumption (S5.3 and S5.4 of SI). The speeds during LTO were estimated using the FDR data. The jet engine data were from ICAO EDB³³.”

(3) In SI, Section S6, a new simplified flowchart for the development was added:

Figure S11 Simplified flowchart for the development of the number based inventory of size-resolved BC particle emissions.”

“Table S5 The main input data required by our BC number emission estimation method.

Estimated parameters	Input data	Engine Specific Data ³⁶	BC particle parameters ⁹
EI_n	Flight data ^{30,34}	Maximum fuel flow rate	Prefactor C
GMD	Flight altitude	π_{00} (Engine pressure ratio)	Exponent ϵ
GSD	Fuel flow rate		
	Flight speed		
	Mass emission indices		

(4) In S5.3 of SI, add details about the flight speed calculation:

“S5.3 Flight speed

For different flight routes (short, medium and long), the typical speeds (mean speeds) of the sub-phases in CCD are first respectively estimated based on the ADS-B data, as shown in Figure S7. When estimating the number emission for a specific aircraft-route pair, the typical speeds (V_C^0 , V_{CR}^0 , V_D^0 with subscripts ‘C’ for Climb, ‘CR’ for Cruise and ‘D’ for Descent) are modified by a correction factor (f_V) for each flight to make the calculated flight distance equal to the route length (D_{Route}) between the departure and the arrival airports.

$$f_V = \frac{D_{Route}}{V_C^0 \cdot T_C + V_{CR}^0 \cdot T_{CR} + V_D^0 \cdot T_D} \quad (5)$$

$$(V_C^{flight}, V_{CR}^{flight}, V_D^{flight}) = f_V \cdot (V_C^0, V_{CR}^0, V_D^0) \quad (6)$$

(5) In S5.4 of SI, add details about the fuel flow rate calculations:

“S5.4 Fuel flow rate and thrust setting

The thrust setting during LTO cycles follows the ICAO default settings. The typical fuel flow rates during CCD are analyzed using the FDR dataset³⁴. One example of the fuel flows normalized by the maximum flow rate (taking-off) during CCD is shown in Figure S9. The typical normalized fuel flow rates of different sub-phases (FF^0) are estimated based on all the FDR data (Figure S10). Similar as the estimation of flight

speed, the typical fuel consumptions are also modified by a correction factor (f_{FF}) for each specific flight to make the total fuel consumption (F_{CCD}) during CCD phases the same as that defined in the EEA dataset.

$$f_{FF} = \frac{F_{CCD}}{FF_C^0 \cdot T_C + FF_{CR}^0 \cdot T_{CR} + FF_D^0 \cdot T_D} \quad (7)$$

$$(FF_C^{flight}, FF_{CR}^{flight}, FF_D^{flight}) = f_{FF} \cdot (FF_C^0, FF_{CR}^0, FF_D^0) \quad (8)$$

Materials and methods

This section also needs substantial changes in order to improve the flow and key messages.

S2.1

P1: The opening sentence should be re-worded to simply mention that FA theory will be used. The sentences after Eq (1), beginning “Recent studies...” should be brought to the beginning of this paragraph. It is better to provide the reasoning behind the utility of a theory before introducing the formulas.

[Response]: We agree with the reviewer. This paragraph has been rewritten as follows:

[Changes]: Section 2.1, P1:

“The BC mass to number conversion was based on the Fractal Aggregates (FA) approaches^{1,36,37}. Recent studies showed that the sizes of the primary particles were relatively uniform in a single aggregate, but the primary particle sizes changed with the sizes of different aggregates^{38,39}. Soot particles show self-similarity on different scales, so the fractal scaling law can be utilized to relate the fractal aggregate size and the number of primary particles (n_{pp})^{9,40,41}.”

P3: The values for Eq (2), along with their confidence intervals should be quoted. Units should also be provided for the constants. What physical parameters is the prefactor C containing? It seems to hide density among other constants. What density does Abegglen et al (2015) [3] assume?

[Response]: We added the explanations for the measurement and uncertainties in Section 2.1, as shown in Change (1). Yes, the prefactor contains the density of the primary particle (the derivations in Change (2)). Abegglen et al.⁹ did not assume any density, but they directly measured the particle mass with desired sizes selected by a Differential Mobility Analyzer (DMA) under specific thrusts. The ϵ and prefactor C were estimated by fitting the power-law relationship between particle mass and size. The analyses of these parameters are added as a new section in SI, as shown in Change (2).

[Changes]:

(1) Section 2.1, P3, add the measurement and uncertainties of particle mass:

“During A-PRIDE experiments, the particle mass were directly measured by Centrifugal Particle Mass Analyzer (CPMA; Cambustion Ltd., Cambridge, UK) at fixed particle sizes and engine thrusts⁹. The particle mass increased from 10^{-21} to 10^{-18} kg with the increasing size from 20 to 150 nm, and the standard deviations were maximum 16.6% for the particles smaller than 40 nm and 11.0% for those larger than 40 nm. The exponent ϵ (dimensionless unit) and prefactor C ($\text{kg m}^{-\epsilon}$) were estimated by fitting the power-law relationship between particle mass and size. The derivation of the relation and the physical meaning of the parameters are shown S4.1 of SI.”

(2) SI, add a new section “S4 Derivation of the fractal aggregate relation”

“S4.1 Power law between particle mass and diameter

Soot particles show self-similarity on different scales, so the fractal scaling law can be utilized to relate the fractal aggregate size and the number of primary particles (n_{pp})^{9,40,41}.

$$n_{pp} = k \left(\frac{d_m}{d_{pp}} \right)^{D_m} \quad (9)$$

where d_{pp} is the volume area equivalent primary particle diameter, d_m is the mobility diameter of an aggregate, k is a scaling prefactor and D_m is the mass mobility exponent.

According to TEM measurements³, the primary BC particles were nearly spherical. The mass of the aggregate particle can be calculated as

$$m_p = n_{pp} \cdot \left(\rho_0 \cdot \frac{\pi}{6} \cdot d_{pp}^3 \right) = k \left(\frac{d_m}{d_{pp}} \right)^{D_m} \cdot \left(\rho_0 \cdot \frac{\pi}{6} \cdot d_{pp}^3 \right) = \frac{\pi}{6} \rho_0 \cdot k \cdot d_m^{D_m} \cdot d_{pp}^{3-D_m}, \quad (10)$$

where ρ_0 is the density of primary particles. According to the FA theory⁴² and recent measurements of jet engine³⁶, the primary particle diameter and mobility diameter of an aggregate has the following relation:

$$d_{pp} = f \cdot d_m^\xi, \quad (11)$$

where f and ξ are constant parameters. The mass of aggregates is expressed as

$$m_p = \frac{\pi}{6} \rho_0 \cdot k \cdot d_m^{D_m} \cdot \left(f \cdot d_m^\xi \right)^{3-D_m} = \left(\frac{\pi}{6} \rho_0 \cdot k \cdot f^{3-D_m} \right) \cdot d_m^{((1-\xi)D_m+3\xi)} = C \cdot d_m^\varepsilon. \quad (12)$$

The prefactor contains the density of the primary particle. However, Abegglen et al.⁹ did not assume any density. They directly measured the particle mass with desired sizes selected by a Differential Mobility Analyzer (DMA) under specific thrusts. The ε and prefactor C were estimated by fitting the power-law relationship between particle mass and size. It is difficult to estimated ρ_0 from the prefactor C , because the parameters k and f were not specifically estimated.”

Eq 4: A citation for this derivation should be provided or included by the authors in the SI if not already published.

[Response]: We added the derivation in Section 4.2 of SI, please see Change (2). In the main manuscript, a sentence was added to indicate the derivation in SI, Change (1).

[Changes]:

(1) After Eq 4, add a sentence

“The derivation can be found in S4.2 of SI.”

(2) In SI, Section 4.2, add the derivation of the relation:

“S4.2 Particle mass-number conversion

The particle mass emission index can be expressed as the integration of all the mass of the single particles:

$$EI_m = \int_0^\infty m_p \cdot EI_n \cdot n(d_m) dd_m \quad (13)$$

$$EI_m = \int_0^{\infty} C \cdot d_m^\varepsilon \cdot EI_n \cdot n(d_m) dd_m = C \cdot EI_n \cdot \int_0^{\infty} d_m^\varepsilon \cdot n(d_m) dd_m \quad (14)$$

We first introduce a theorem: according to the properties of lognormal distribution⁴³, if a random variable X follows the lognormal distribution with parameters μ and σ , for any real number t we have:

$$E(X^t) = \exp\left(\mu t + \frac{1}{2} \sigma^2 t^2\right), \quad (15)$$

Since d_m follows the lognormal distribution $n(d_m)$, we have

$$EI_m = C \cdot EI_n \cdot \int_0^{\infty} d_m^\varepsilon \cdot n(d_m) dd_m = C \cdot EI_n \cdot E(d^\varepsilon) = C \cdot EI_n \cdot \exp\left(\mu\varepsilon + \frac{1}{2} \sigma^2 \varepsilon^2\right). \quad (16)$$

According to the definitions of GMD and GSD, there are

$$GMD = \exp(\mu), \quad (17)$$

$$GSD = \exp(\sigma), \quad (18)$$

Finally, the relation between EI_m and EI_n is expressed as

$$EI_m = C \cdot EI_n \cdot (\exp(\mu))^\varepsilon \cdot \exp\left(\frac{1}{2} \sigma^2 \varepsilon^2\right) = C \cdot EI_n \cdot GMD^\varepsilon \cdot \exp\left(\frac{\varepsilon^2 \cdot (\ln(GSD))^2}{2}\right). \quad (19)''$$

S2.2

P1 (Part 1/3): These papers do not use T3 or fuel flow rate to estimate GMD, rather it simple plotted on this basis since these are measurable performance characteristics. As far as I am aware, there are no published methods to estimate the GMD however this opening paragraph seems to suggest that there are. If there are, it is worth spending some time comparing them to your results later in the paper and introducing them here.

[Response]: In a recent conference poster, Teoh et al. (2017)¹ attempted to estimate the GMD based on thrust or fuel flow rate. Yes, there are not fully established methods based T₃ or EI_m, only some suggestions that these kinds of correlation could be used to estimate GMD, e.g. Durdina et al. (2017)⁵ and Brem et al. (2016)⁴⁴. In the revised manuscript, we modified this paragraph, added discussion on the performance of the correlations based on fuel flow rate, T₃ and EI_m. At the same time, we also tried to explain the physical basis and the reason for the utilization of the final complex form of the fitting correlation.

[Changes]: Section 2.2, P1, P3 and P4 compare the results calculated by different models:

“Recent studies attempted to estimate GMD based on its correlation with the thrust or fuel flow rate¹, which is discussed in S2.1. However the ground and the cruise measurements do not follow the same correlation and the data are highly scattered (Figure S1 (a) and (d)).”

“The GMDs of BC particles are generally positively correlated with T₃, but with a slight negative correlation at low temperatures as shown in Figure S1(b), which was also observed in the experiments with simulated flight altitudes from sea level to 15.2 km^{4,5}. The correlation can be depicted by a quadratic equation. The increasing trend is because of the growth of the primary BC particles as discussed above

and also the effective agglomeration due to the violent turbulence mixing and the intensive motion of the primary particles triggered by the high temperature. In the T_3 range of about 300~500 °C, GMD decreases slightly with increasing T_3 which may be explained by the reduced agglomeration due to the decreasing residence time of the combustion product.

...The GMD increases monotonically with $EI_m(BC)$ (Figure S1(c)), which is hypothetically mainly caused by coagulation. A cubic equation is utilized to depict the overall trend. The correlation of $EI_m(BC)$ is combined with that of T_3 to modulate the GMD, resulting in the form of Equation (1).”

P1 (Part 2/3): This paragraph and subsequent ones in this section contain a lot of details about the measurement techniques and caveats. These should be separated into its own section.

[Response]: Thank you for your suggestion. In the revised manuscript, these contents were separated into 2.3 Dataset for model training.

[Changes]: In Section 2.3, add the separated section for measurement details

“2.3 Dataset for model training

The coefficients of the GMD- T_3 & EI_m and GSD- EI_m relations were obtained by fitting the measured engine emission data from the literature ^{7,21,45}. The data summary is shown in Section S1 and Table S1 & S2 of the Supporting Information (SI). There were 53 available datasets, including 7 types of aircraft engines from different manufacturers to cover a wide range of BC emissions. The fuel flow rate, EI_m , GMD and GSD were simultaneously measured. The EI_m ranged from 0.26 to 260 mg per kg-fuel with the thrust levels between 3% and 100%. The GMDs were from 14.7 to 40.9 nm and the GSDs located between 1.46 and 2.03. All the ground measurements were conducted using the new standard ³² compliant systems. The recent measurements of cruise emissions ²¹ were also included. The T_3 was estimated following the FOX method ²⁴, which is briefly introduced in Section S3 of SI.

The dataset from APEX ⁶ was utilized as independent data to validate the fitted correlation. The dataset contains 84 data points measured for 11 power settings from 4% to 100% of CFM56-2-C1 engines. More detailed information about the dataset can be found in Section S2.7 of SI.”

P1(Part 3/3): It would be nice to provide a table of the details similar to for example, Speth et al (2015) [7].

[Response]: The details about the dataset were summarized in Table S1 and S2 of the SI.

[Changes]:

(1) Section 2.3, P1, add sentence to show the summary of the utilized dataset was provided in SI:

“The data summary is shown in Section S1 and Table S1 & S2 of the Supporting Information (SI).”

(2) In SI, Section S1, add the tables to provide the information about the utilized dataset:

“Table S1 Database for GMD and GSD fitting

No.	Engine type	Altitude	Fuel Type	Fuel flow rate (%)	EI_m Range (mg/kg)	GMD (nm)	GSD	Data Num	Ref.
1	JT8D-219	ground	Jet A	4~100	1.3~260	21~41	1.64~1.79	7	⁴⁵
2	CF6-80A2	ground	Jet A	4~30	0.73~0.45	14.7~17.6	1.46~1.56	4	⁴⁵

3	CF6-80C2B8F	ground	Jet A	4~50	0.26~1.8	15.2~20.6	1.48~1.62	9	45
4	PW 2037	ground	Jet A	4~70	1.6~27.4	18.9~31.1	1.58~1.76	7	45
5	CFM56-7B24/3	ground	Jet A-1	3~96	1.81~69.3	18.8~40.1*	1.57~1.85*	6	7
6	PW4168	ground	Jet A-1	4~99	0.97~77.8	20.2~40.9*	1.59~1.78*	8	7
7	CFM56-2-C1	cruise	Jet A	23~38	12.7~82.4	23.5~35.3	1.63~1.86	6	21
8	CFM56-2-C1	cruise	HEFA -Jet A	23~38	4.1~37.6	20.9~28.7	1.58~2.03	6	21
9 ^Δ	CFM56-2-C1	ground	JP8	4~100	2.4~198.6	15.9~36.9	1.51~1.98	84	6

* The GMD and GSD have been corrected in this study by the line loss function in Durdina et al. ⁸.

^Δ The dataset was only utilized to validate the fitted correlation, not used for the model training.

Table S2 Experimental conditions for the dataset

	Engine type	Size distribution	Number instrument	Mass instrument	Probe distance	Line loss correction	Ref.
1	JT8D-219	DMS500 & TSI3071	TSI3022	LII300	Exhaust exit plane	Yes	45
2	CF6-80A2	DMS500	TSI3022	LII300	Exhaust exit plane	Yes	45
3	CF6-80C2B8F	DMS500	TSI3022	LII300	Exhaust exit plane	Yes	45
4	PW 2037	DMS500	TSI3022	LII300	Exhaust exit plane	Yes	45
5	CFM56-7B24/3	DMS500	TSI3790E	LII300 & MSS	Exhaust exit plane	No*	7
6	PW4168	DMS500	TSI3790E	LII300 & MSS	Exhaust exit plane	No*	7
7	CFM56-2-C1	TSI3936	TSI7610 TSI3010	PSAP	30~150 m	Yes	21
8	CFM56-2-C1	TSI3936	TSI7610 TSI3010	PSAP	30~150 m	Yes	21
9	CFM56-2-C1	SMPS DMS500	TSI3022 TSI3760 TSI3022	PSAP MAAP	1 m	Yes	6

* The GMD and GSD were not corrected in the original article, but have been corrected in this study by the line loss function in Durdina et al. ⁸.

Eq 6 (Part 1/2): This equation seems to have appeared from nowhere. It is a complex form and no reasoning is provided for it. While a correlation is fine, there should be a physical basis to the form of the equation.

[Response]: We have added explanations in the revised manuscript. Please refer to the response to Major Changes No.1.

[Changes]: Please refer to the changes in Major Changes No. 1.

Eq 6 (Part 2/2): In its current state, the equation has 6 parameters while there are 53 data points. This suggests some degree of overfitting and it would be interesting to see a bias-variance curve as certain coefficients are removed. The analysis can be left to the SI, but it should be certainly be mentioned that the authors have checked this.

[Response]: Bias-variance trade-off of our model has been tested. We have added explanations in the revised manuscript. Please refer to the response to Major Changes No.2.

[Changes]: Please refer to the changes in Major Changes No. 2.

P2: Is there any reason that GSD should not be independent of T3? Why is this surprising?

[Response]: We added a new paragraph to explain why the form was utilized.

[Changes]: Section 2.2, P5:

“TEM measurements³ indicated that standard deviations of the primary particle sizes increase with T_3 , and the distribution width of agglomerated BC particles could also change with $EI_m(BC)$ due to the abundance of primary BC particles available for the growth into a vast range of sizes. The plot between GSD and T_3 is highly scattered, but there is a good dependence of GSD on $EI_m(BC)$, indicating that coagulation may be the dominant factor for the width of the BC particle distribution, therefore the relation with $EI_m(BC)$ as the parameter (Equation 2) is adopted to fit the data.”

Eq 7 (Part 1/2): What type of regression is being conducted here? Are you using linear regression? Same with Eq 6? Presumably this is non-linear regression, however I think both equations can be re-written in the form suitable for linear regression? These are extremely important points to mention in the paper as the choice of regression cost function can alter the fitted results.

[Response]: The Levenberg-Marquardt nonlinear least squares algorithm, which can be viewed as a combined algorithm of Gauss-Newton and gradient descent methods, was used to iteratively minimize the sum of the squares of the deviations between the observations and the predicted values in Equation (1). We utilized the function “nlinfit” in the Statistics and Machine Learning Toolbox of MATLAB (Version 2015a) to estimate the parameters. The initial value for the iterative algorithm was [1 1 1 1 1 1], and the final estimated coefficients were [0.7812 7.289 30.34 85.85 -0.7605 0.5859]. Termination tolerance on the estimated coefficients was 10^{-8} . For Eq 7, the least square linear regression was utilized to estimate the parameters.

[Changes]:

(1) Section 2.2 P6, add information for the utilized regression method and cost function:

“For Equation (1), the Levenberg-Marquardt nonlinear least squares algorithm was used to iteratively minimize the sum of the squares of the deviations between the observations and the predicted values. For Equation (2), the least square linear regression was utilized to estimate the parameters.”

(2) In SI Section S2.4, add detailed information for the utilized regression method:

“The Levenberg-Marquardt nonlinear least squares algorithm, which can be viewed as a combined algorithm of Gauss-Newton and gradient descent methods, was used to iteratively minimize the sum of

the squares of the deviations between the observations and the predicted values. We utilized the function “nlinfit” in the Statistics and Machine Learning Toolbox of MATLAB (Version 2015a) to estimate the parameters. The initial value for the iterative algorithm was [1 1 1 1 1 1], and the final estimated coefficients were [0.7812 7.289 30.34 85.85 -0.7605 0.5859]. Termination tolerance on the estimated coefficients was 10^{-8} .”

Eq 7(Part 2/2) In fact, conducting linear regression would allow the authors to present adjusted R2 values for the actual fit, rather than in the parity plots shown later. (NOTE: The traditional R2 will always increase as more parameters are introduced so this trend in the SI is not surprising at all. The adjusted R2 takes into account the number of features used and so identifies if there is truly any performance increase with using additional parameters.)

[Response]: Thank you for your suggestions! Equation (1) is nonlinear. It is not easy to transform it into linear form. We thought that R^2 is not valid for nonlinear regression. But after further analysis as follows, we found that R^2 is still applicable for Equation (1) due to the cost function we utilized and the form of the model, please see Change (3). In the revised manuscript, we added the adjusted R^2 , please see Change (1) and (2)

[Changes]:

(1) Section 3.1 Quality of GMD and GSD estimations, Figure 1 and Table 2 have been modified to include adjusted R^2

Figure 1. The goodness of the adopted fitting relations: (a) GMD estimated by both T_3 and EI_m ; (b) GSD estimated by EI_m .

Table 2 Comparisons among different correlations between GMD and engine parameters

	GMD-fuel flow rate	GMD-T ₃	GMD-EI _m	GMD-T ₃ &EI _m
Adjusted R^2	0.63	0.68	0.81	0.94
RMSE	4.03	3.69	2.82	1.51

(2) In SI Section S2.1, Figure S1 has been modified to include adjusted R^2 .

Figure S2 The correlations between GMD and the engine parameters: a) ratio of maximum fuel flow rate, b) T₃ temperature and c) EI_m. The shaded area represents the 95% prediction interval. The comparisons among the qualities of the different fitting approaches, namely the goodness of fittings: d) GMD-fuel flow rate; e) GMD-T₃ and f) GMD-EI_m.

(3) In SI Section S2.5, add the proof to show R^2 is still valid for the nonlinear GMD-T₃&EI_m regression:

“Normally, the adjusted R^2 is applicable to the linear fittings. Here we show that it is also valid for the current GMD-T₃&EI_m correlation. In order to calculate an effective R^2

$$R^2 = 1 - \frac{Err_{res}}{Err_{tot}} = \frac{Err_{reg}}{Err_{tot}}, \quad (20)$$

we should have

$$Err_{tot} = Err_{reg} + Err_{res}, \quad (21)$$

where Err_{tot} is variance of the data y_i ; Err_{reg} is the explainable variance by the regression f_i ; Err_{res} is the residual or unexplainable variance:

$$Err_{tot} = \sum_{i=1}^n (y_i - \bar{y})^2; Err_{reg} = \sum_{i=1}^n (f_i - \bar{y})^2; Err_{res} = \sum_{i=1}^n (y_i - f_i)^2. \quad (22)$$

Based on the definition of Err_{tot} , we can have

$$\begin{aligned} Err_{tot} &= \sum_{i=1}^n (y_i - \bar{y})^2 = \sum_{i=1}^n (y_i - f_i)^2 + \sum_{i=1}^n (f_i - \bar{y})^2 + 2 \sum_{i=1}^n ((y_i - f_i)(f_i - \bar{y})) \\ &= Err_{res} + Err_{reg} + 2 \sum_{i=1}^n ((y_i - f_i)(f_i - \bar{y})) \end{aligned} \quad (23)$$

Equation (21) can be proved, if the third term is zero.

$$\sum_{i=1}^n ((y_i - f_i)(f_i - \bar{y})) = \sum_{i=1}^n (y_i f_i) - \sum_{i=1}^n (f_i f_i) - \sum_{i=1}^n (y_i \bar{y}) + \sum_{i=1}^n (f_i \bar{y}) \quad (24)$$

Here, we suppose a new function $g(a, b)$:

$$g(a, b) = \sum_{i=1}^n (a \cdot f_i + b - y_i)^2 \quad (25)$$

It is necessary to introduce two assumptions: (1) the cost function is in the form of least square; (2) scaling (multiplied by a) or shifting (added by b) of the model f would not improve the fitting. If the two

assumptions are satisfied, then $\sum_{i=1}^n (f_i - y_i)^2$ is the optimized fitting, namely the function $g(a, b)$ can

achieve the minimum value when $a=1$ and $b=0$. Then we have

$$\left. \frac{\partial g(a, b)}{\partial a} \right|_{a=1, b=0} = \left. \frac{\partial g(a, b)}{\partial b} \right|_{a=1, b=0} = 0 \quad (26)$$

$$\left. \frac{\partial g(a, b)}{\partial a} \right|_{a=1, b=0} = 2 \sum_{i=1}^n ((f_i - y_i) f_i) = 0 \quad (27)$$

$$\left. \frac{\partial g(a, b)}{\partial b} \right|_{a=1, b=0} = 2 \sum_{i=1}^n (f_i - y_i) = 0$$

As a result, we have

$$\sum_{i=1}^n ((y_i - f_i)(f_i - \bar{y})) = 0 \quad (28)$$

Then Equation (20) is proved.

For the linear regressions with a constant term, the two assumptions can be satisfied. Equation (1) meets the first assumption, because the Levenberg-Marquardt nonlinear least squares algorithm was utilized. For the second assumption, scaling (multiplied by a) of the model f cannot improve the fitting, and shifting (added by b) of the model f also barely changes the results, because there is a constant term $A_4 \cdot A_6$. As a result, R^2 is still applicable for the nonlinear regression of GMD-T₃&EI_m correlation."

P3 (Part 1/3): I'm not sure that there is sufficient evidence to suggest that number EIs do not change downstream. It seems coagulation is strong in the plume so I would suspect the GMD to increase and thus number EIs to reduce in order to conserve mass. Indeed, Wong et al (2014) [6] suggests that the number EI does decrease as the GMD increases.

[Response]: In the revised manuscript, this part is removed. Please also refer to the response to “Introduction P4”.

P3 (Part 2/3): You mention that these measurements were taken at the exit plane however do the authors account for system losses? In addition, the cruise emissions could not possible be at the exit plane but some distance downstream.

[Response]: It is more accurate to say that the data have been corrected for the particles loss in the measurement system. Most of the data have already been corrected using size-dependent loss functions. In the revised manuscript, we revised this part as follows:

[Changes]:

(1) Section 2.3 P2, add explanations for the system loss correction:

“Most of the size distribution and indices have already been corrected for the particle loss in the measurement system using size-dependent loss function, except the data in A-PRIDE 4⁷, which were corrected in the present study by the line loss function in Durdina et al.⁸ for the same measurement system.”

(2) For the discussion on system losses, please also refer to the response to “Introduction P3”.

P3 (Part 3/3): I don't know why sulfuric acid is being brought into this? Why are you discussing volatiles...?

[Response]: We removed these sentences in the revised manuscript. Please also refer to the response to “Introduction P4”.

S2.3

P2 (Part 1/2): The reasons for using the AEM and EEA guidebook should be more clearly stated – it is because there are missing SN values in the ICAO EDB. This is mentioned in the SI but should be brought here as well. Indeed, the authors do not mention how many engines in the 2005 inventory did not have SN entries within the EDB. There are many extremely old, outdated entries within the EDB that should not be counted.

[Response]: We agree with the reviewer. The reasons for utilization of EEA data were added in the revised manuscript. In the 2005 inventory, there were 214 different aircrafts (76 engine types). Among them there were 75 aircrafts (16 engine types) without SN data in the EDB, and another 39 aircrafts (12 engine types) with SN but measured before 1990.

[Changes]:

(1) Section 2.4 P1, add reasons for the utilization of EEA data:

“In the AEM model adopted by EEA, the BC emissions were estimated by the SN dependent FOA3 method⁴⁶, but the missing SN data in the ICAO Engine Emissions Databank³⁵ were estimated based on the algorithm developed by German Aerospace Center (DLR) and the cruise emissions are also corrected for the atmospheric environment at high altitudes⁴⁷.”

(2) Section 2.4 P1, add the statistics data of the engines in the inventory

“In the 2005 inventory, there were 214 different types of aircrafts (76 types of engines). Among them there

were 75 types of aircrafts (16 types of engines) without SN data in the EDB, and another 39 types of aircrafts (12 types of engines) with SN but measured before 1990.”

P2 (Part 2/2): A better overview needs to be provided about the statistical analysis. E.g., what is the output of the statistics? Is it just the mean time and power in each mode? What's the confidence in these parameter estimates? Some of these details should be provided in the main paper rather than left in the SI under the results section.

[Response]: Discussion on the statistics was added in the revised manuscript, please see Changes below. For the usage of these data, please refer to the response to “Introduction P5”.

[Changes]: Section 2.4 P2, add discussion on the statistics

“The flight conditions (e.g. fuel flow rate, speed and altitude) influence the BC mass to number conversion. In order to better estimate the number emissions, the EEA mass emission indices for LTO and CCD phases were further divided into seven sub-phases, taxi/take-off/climb-out/approach for the LTO cycle and climb/cruise/descent for the CCD phase. For the LTO cycle, the duration and fuel flow rates followed the ICAO default settings. The typical flight speeds of LTO sub-phases were statistically analyzed based on 901 records of Flight Data Recorder (FDR)³⁴, showing the mean speeds were about 20 m/s for taxi, 60 m/s for take-off, 90 m/s for climb-out and 80 m/s for approach, with the uncertainties of 10 to 20 m/s. In order to analyze the CCD sub-phase conditions, more than 13000 records of the Automatic Dependent Surveillance-Broadcast (ADS-B), containing 1500 regular flight routes operated by American Airlines and Lufthansa on 9 different days, were collected from the OpenSky network (<https://opensky-network.org/>). The results showed that the cruise velocity (about 230 m/s) was the largest one, followed by climb phase (about 200 m/s) and descend phase (about 170 m/s). The uncertainties were about 10 m/s for climb and descend, and larger for cruise about 20 m/s. The uncertainties for fuel flow ratio were much lower, only about 0.05 for cruise and descend, and about 0.1 for climb. The fuel consumptions and BC mass emissions in the EEA dataset were divided into the seven sub-phases based on the flight conditions, with the total amounts matching the original EEA data. The detailed processing method is described in Section S5 of SI.”

S2.4

P1 (Part 1/2): You need to discuss what “re-developed version” means. What changes have been made and why?

[Response]: The code was modified to utilize the EEA dataset and the $EI_n(BC)$ data converted from $EI_m(BC)$, which were not in the original version. In the revised manuscript, we added explanations for the “re-developed version”:

[Changes]: Section 2.5, P1:

“The BC number emission inventory was compiled based on the open source Aviation Emission Inventory Code (AEIC)^{24,31,32}. The code was modified to utilize the EEA dataset and the $EI_n(BC)$ data converted from $EI_m(BC)$, which were not in the original version.”

P1 (Part 2/2): Do you have uncertainties in the error in flight trajectory length. How is the correction made? How would this affect the flight path?

[Response]: In the current analysis, the uncertainties of the flight length error were not included, and the

flight path was not modified due to the lack of information about the actual trajectories. In the revised manuscript, we added explanation for this:

[Changes]: Section 2.5, P1:

“The flight path was not modified due to the lack of information about the actual trajectories. The uncertainties of the flight trajectory length were not included in the current analysis.”

P2: Is T3 referring to static or total temperature? I'm guessing total so Tt3 should be used.

[Response]: Yes, it is the total temperature, since the compression caused by the flight speed is considered in the calculation. In the revised manuscript, the sentence was modified as follows:

[Changes]: Section 2.5, P2:

“... which were required for the calculation of T_3 to include the compression effects by the flight speed. We keep the symbol T_3 to be consistent with previous publications^{5,24}.”

Results and discussion

The sub-sections here should attempt to match the subsections in the Materials and Methods section. Thus, you should begin with the GMD and GSD correlations, then the “mass to number conversion”, before going into AEIC validation and finally the analysis of number emissions.

[Response]: In the revised manuscript, we changed the sequence of the sections as suggested:

[Changes]: New sequence of this section:

“3.1 Quality of GMD and GSD estimations; 3.2 Validation of the BC mass to number conversion
3.3 BC and gas mass emissions; 3.4 Particle number emission ...”

S3.1

Table 1: The table (either within the table or in the caption) should confirm the year for which these values are quoted. The EI(CO₂) in this paper's results seems to be higher than that for all other models. The value used by the author should be quoted. This BC number values should be removed from this table as this is not the purpose of this section.

[Response]: The years of the emission inventories are included in the caption in the revised manuscript.

In the EEA database, the ratio between CO₂ emission and fuel consumption is 3.15 as a constant. The gas emissions were calculated separately from the fuel consumption and the BC emissions. We found that there was a bug for the gas emissions in our code for the short distance flights (less than half an hour), which caused about 5% difference. In the revised manuscript, we corrected the bug and updated the gas emissions.

[Changes]: Table 2:

“Table 1 Comparisons of the BC and gas emissions with other inventories: Aviation Emission Inventory Code (AEIC, 2005)^{24,27}, Aviation Environmental Design Tool (AEDT, 2006)⁵⁹, Emissions Database for Global Atmospheric Research (EDGAR, 2010)⁶⁰ and Global Aircraft Emissions Data Project for Climate Impacts Evaluation (AERO2k, 2002)³⁰.”

	AEIC ^{24,27}	AEDT ⁵⁹	EDGAR ⁶⁰	AERO2k ³⁰	This study
CO ₂ (Gg)	5.7×10 ⁵	5.9×10 ⁵	7.5×10 ⁵	4.9×10 ⁵	5.7×10 ⁵
NO _x (Gg)	2.7×10 ³	2.7×10 ³	2.9×10 ³	2.1×10 ³	2.5×10 ³
SO _x (Gg)	2.2×10 ²	2.2×10 ²	2.5×10 ²	/	1.5×10 ²
CO (Gg)	7.9×10 ²	6.7×10 ²	5.3×10 ²	5.1×10 ²	6.2×10 ²
Fuel burn (Gg)	1.8×10 ⁵	1.9×10 ⁵	/	1.6×10 ⁵	1.8×10 ⁵
BC mass (Gg)	16.9(2.4+14.5) ^a 2.0(0.6+1.4) ^b	6.8	9.0(0.6+8.4) ^c	3.9	9.5(0.7+8.8)

^a Estimated by the FOX method;

^b Estimated by the FOA3 method;

^c The data in the parentheses are the (LTO emissions + CCD emissions).”

P2: This paragraph should begin with the discussion on global values first, before talking about the spatial distribution. The method used in EDGAR should be discussed since the results in this study are most similar to them. Discrepancies with AEDT and AERO2k are also not discussed at all. Are there any other papers that suggest that the Doppelheuer and Lecht relationship underestimates cruise emissions? The reference provided [8] severely overfits the specified data and is also based on only a single engine. The final sentence of this is incorrect – using the Doppelheuer and Lecht relationship has not “offset” the FOX cruise estimate; it still seems to be twice as large as the next largest estimate.

[Response]: We agree with the reviewer. We revised this paragraph to begin with the global emission. More discussions were added for the comparison between different inventories, please see Changes (1) and (2). We deleted the discussion about Doppelheuer and Lecht relationship, because currently there is not enough evidence to show it underestimate the cruise emission.

[Changes]:

(1) Section 3.3, P2

“The result of BC mass in this study is close to EDGAR data, because EDGAR also utilized the EEA database⁴⁸.”

(2) Section 3.3, P2

“The current results are about 40% larger than those of AEDT inventory, which adopted a fixed BC emission index (30 mg/kg-fuel) for the CCD emissions⁴⁹. The EI_m(BC) of AERO2k are notably smaller than other inventories. AERO2k utilized a SN dependent estimation method for BC mass emission, so the smaller EI_m(BC) may be due to the missing SN in EDB.”

S3.2

P1: I think it is bold to say that the “correlation is applicable for different flight conditions”. This is based on results for only a single engine so is not a robust conclusion. While it may be fine to use this in analysis for the time being, it should be mentioned that more data is required for different engines in order to be reliable. In addition, ground emissions of the CFM56-2-C1 have not been provided here and this would provide more justification on whether the GMD for this engine is predicted well from ground to cruise.

[Response]: Thank you for pointing out this important issue. We have deleted the sentence in the revised manuscript. Please refer to the response to the Major Changes No. 4.

Table 2: Is there any reason that correlation coefficient is used instead of R2.

[Response]: Adjusted R² has been added in the table. Please refer to the response to S2.2 Eq7.

P2: Why is the correlation coefficient less satisfactory? For this field, a correlation coefficient of 0.77 is extremely good, especially for a parameter where there is not great understanding on the driving forces. Why is the blended fuel likely to lead to different GSD? There is no discussion on why this could be an outlier.

[Response]: We removed the expression of “less satisfactory”, please see Change (1). The outlier is not due to the burning of biofuel, but because only two plume samples were captured, which led to high measurement uncertainty. Please see Change (2). The results of the independent evaluation using APEX data were provided (Change 3).

[Changes]:

(1) Section 3.1, P4

“Figure 1(b) shows the goodness of fit for GSD. The data scattering is relatively low with a small RMSE (0.07) and the adjusted R^2 is also acceptable.”

(2) Section 3.1, P4

“The GSD- EI_m relation provides generally reasonable estimation of the GSD values except one outlier data point of CFM56-2-C1 which has high uncertainty due to only two measurements of the plume.”

(3) Section 3.1, P4

“The independent evaluation using the APEX dataset also shows low error (RMSE = 0.06) and high correlation ($r = 0.78$) (Figure S3).”

S3.3

P1: The SULFUR 1-7 data seems to have some information on BC number distribution – why is this not used in the developing the correlations in the first place? Why are some of the data points used to develop the original correlation not used to compare predicted vs measured $EI\#$? The authors only use “reliable” data, but there is no mention on what this means and how it was assessed. As a consequence, this instead sounds like the authors have selected data to provide a higher fit. I am certain that this is not the case, but the reasons behind why other sources were ignored should be provided and it should also be mentioned which sources were ignored.

[Response]: We tried our best to utilize all the data including the SULFUR 1-7 data to validate the model, but the particle size distribution data was incomplete in SULFUR 1-7. Only the particle size distribution for ATTAS and B737-300 aircrafts were measured during SULFUR 6, so we did not use the data to fit the correlation, but in Table S5, we compared the measured and calculated GMD and GSD for these two aircrafts, please see Change (2) and (3). The $EI_n(BC)$ (about 10^{16} /kg-fuel) in Delta-Atlanta Hartsfield Study⁴⁵ were nearly one magnitude higher than the common values ($10^{14}\sim 10^{15}$ /kg-fuel), which may be due to a problem of the condensation particle counter (CPC), so the dataset was not included in the validation. The evaluation results of the independent evaluation using APEX dataset were also provided in the revised manuscript, Change (4).

[Changes]:

(1) Section 3.2 P1, add explanations for the reason why the dataset in⁴⁵ was not utilized.

“The $EI_n(BC)$ (about 10^{16} /kg-fuel) in⁴⁵ were nearly one magnitude higher than the common values

(10^{14} ~ 10^{15} /kg-fuel), so we excluded the dataset as outliers.”

(2) Section 3.2 P1

“The estimated GMDs and GSDs of SULFUR experiments are also provided in Table S6 of SI.”

(3) In SI, Table S6

Table S6 Cruise data for the validation of particle number emission estimation

Aircraft	B707-307C	A340-300	ATTAS ⁺	B737-300 ⁺	A310-300
Engine Type	PW JT3D-3B	CFM56-5C4	Mk501	CFM56-3B1	CF6-80C2A2
Flight height (100 ft)	328.67 (310, 334 342)*	325.4 (336.6, 314)	260 (260, 260)	260	350
Flight Speed (m s ⁻¹)	193.33 (187, 190, 203)	194.5 (195, 194)	156.5 (160, 153)	167	180
Fuel flow (kg s ⁻¹)	0.37	0.26	0.151	0.213	0.4
El _m (g kg ⁻¹ -fuel)	0.5	0.01	0.1	0.011	0.019
El _n (kg ⁻¹ -fuel)	1.7×10^{15}	1.8×10^{14}	1.7×10^{15}	3.5×10^{14}	6×10^{14}
Measured GMD	/	/	(34, 35)	25	/
Estimated GMD	39.64	26.41	32.38	25.96	28.03
Measured GSD	/	/	1.55	1.55	/
Estimated GSD	1.86	1.68	1.78	1.68	1.71
Reference	50	50	51	51	51

* The values in the brackets are the measurements during the experiment. Only the means of these data are utilized as the flight condition.

⁺The particle size distribution for ATTAS and B737-300 aircrafts were measured during SULFUR 6.

(4) Section 3.2 P4, add the discussion on the independent evaluation using APEX dataset:

“The independent evaluation using APEX dataset also shows a similar performance with a high correlation coefficient (0.85) and low RMSE (1.8×10^{14}) (Figure S3).”

P2: The issues regarding the uncertainties analysis have already been brought up as a major focal point to improve.

[Response]: Please refer to the response to the Major Changes No.3.

P3: It is interesting that you mention that the uncertainties can be “as large as a factor of tw due to the uncertainty particle sizes and the fractal parameters”. It would be interesting to see sensitivities plots to understand the contribution of each uncertain term to the overall uncertainty.

[Response]: We added a new plot in SI for relative uncertainties caused by individual parameters as follows:

[Changes]:

(1) Section 3.2, P4

“The estimations are most sensitive to the uncertainty of ϵ (Figure S16).”

(2) Figure S16

Figure S16 Uncertainties caused by the individual uncertain parameters. “Relative” means the predicted value divided by the measured value.

S3.4

P1: I would introduce the global number emissions estimate here rather than in Table 1.

[Response]: In the revised manuscript, the global number emissions in Table 1 are removed. Introductions are added in this paragraph.

[Changes]: Section 3.4 P2:

“The total BC particle number emission is about 1.09×10^{26} per year (95% prediction interval of $0.68 \sim 1.5 \times 10^{26}$ per year). The LTO cycles below 3000 ft and CCD above 3000 ft respectively account for 10.2% (1.1×10^{25}) and 89.8% (9.8×10^{25}) of the total particle number. The AERO2k inventory³⁰ developed by EUROCONTROL is the only currently available inventory that explicitly includes the BC number emissions, which estimated a total number emission of 4.03×10^{25} for the year 2002³⁰.”

P2: It’s strange that you only bring up that AERO2k have particle number emissions’ estimates as well. It would seem prudent to bring this up earlier in the methods or even introduction. You should also be more clear what you mean by “ratios”. Given that it is the only other model out there, you should spend more time discussing its method versus your own. Should “prediction interval” actually be “confidence interval” – you have confidence interval in the mean of an estimate but prediction interval if you wanted to estimate for future emissions, for example.

[Response]: In the revised manuscript, we added a description of AERO2k inventory in the introduction section. Please see Change (1).

We added explanations on how the number emissions were estimated in AERO2k, and also added more discussion to compare our results with those of AERO2k. Please see Change (2).

Yes, it should be “confidence interval”, and we have corrected them in the revised manuscript. Actually, we used “prediction interval” to represent the uncertainties in the GMDs and GSDs estimated by the developed correlations, so it should be “confidence interval” for the final inventory, which was generated by MC runs.

[Changes]:

(1) Introduction, P2:

“As the only currently available BC number inventory, AERO2k estimated the number-based emission using simplified general characteristic of particle number emission from different engines.”

(2) Section 3.4 P2:

“AERO2k estimated the particle number emissions using approximate ratios between particle number and mass at different altitudes: about 1.6×10^{16} /g-BC for cruise and about 5×10^{15} /g-BC for LTO. In contrast, our estimations are 1.1×10^{16} /g-BC for CCD and 1.5×10^{16} /g-BC for LTO. The ratio for LTO is much larger than that utilized by AERO2k due to the high value of taxi (3.6×10^{16} /g-BC), which is in line with the recent measurements showing that the size³ and mass⁹ of BC particles are smaller at lower thrusts. Our overall number-mass ratio (1.15×10^{16} /g-BC) is comparable with that of AERO2k (1.03×10^{16} /g-BC). Our total number emission is about 2.5 times of that in the AERO2k inventory, which may be caused by the low total BC mass emission (3.9 Gg) in AERO2k.”

P3: You don't need to mention highest emissions in a particular grid square. Since you are calculating in lat/lon coordinates, this will be affected by the area of the grid cell. Stick to regions rather than grid cells.

[Response]: We agree and deleted the sentences in the revised manuscript.

P4: I'm a little confused by the discussion on lines 312-314. Surely larger particles at higher thrusts should decrease EI# for a fixed Elm? I think the point made here is not clear. In addition, this is a discussion that seems relevant for the method behind the correlation. You also mention number emissions from the AEDT inventory, but it's unclear why this isn't in Table 1 then? Towards the end of this paragraph, you compare EIs for the 737-800 at cruise. Firstly, you should mention the engine as well as the aircraft and secondly, I think this should be in Section 3.3 where you validate your method. Indeed, you could have a separate section highlighting the performance at cruise.

[Response]: For the discussion between Lines 312-314, the expressions were not very accurate. The emission characteristics were valid for the widely used Rich-Quench-Lean (RQL) combustors, but they may be not applicable for other engines. In the revised manuscript, we rewrote this part to mention that it is similar with the characteristics of the RQL combustors. Please see Change (1).

In AEDT inventory, there was no BC particle number emission estimations, but only a couple of BC number emission indices were provided for all the aircrafts. The recommended values were averaged from the literature data. The clarifications have been added to explain the information available in AEDT inventory. Please see Change (2).

(3) The aircraft was equipped with CFM56-7B26 engine. We added the information in the revised manuscript. Section 3.3 aimed to validate the correlation using the data for specific aircrafts. In this section, we assess the quality of the developed inventory, so the global averaged emission indices were compared with the literature data. We think it is better to keep the discussion here.

[Changes]:

(1) Section 3.4 P4

“The largest EI_n is obtained in the climb phase above 3000 ft, with a value of $(9.2 \pm 1.95) \times 10^{14}$ /kg-fuel, in line with the typical emission characteristics for the widely used RQL combustors, which have the maximum EI_n at about 65% thrust and somewhat lower EI_n with higher thrust⁵.”

(2) Section 3.4 P5

“In AEDT, there was no BC number emission inventory, but a couple of emission indices averaged from the literature data were recommended for all the aircrafts as 2×10^{14} ($0.1 \times 10^{14} \sim 6 \times 10^{14}$)/kg-fuel below 3000 ft and 4×10^{14} ($1 \times 10^{14} \sim 60 \times 10^{14}$)/kg-fuel above 3000 ft⁴⁹”

S3.5

P1: It's unclear how the particle size distributions are integrated together. I don't think it is trivial to sum multiply different lognormal distribution together. Do you only account for the uncertainties in the mass to number conversion? It is surprising that it only takes 100 Monte Carlo simulations to converge, whereas the uncertainties calculated in Section 3.3 required 5000 runs.

[Response]: For the size distribution, please refer to the response to the Major Changes No. 6. For the uncertainties, only the uncertainties in the mass to number conversion were considered. The inventory is the sum of the emissions from millions of different aircrafts, so the calculation converges at a lower number of runs compared to the results of individual aircrafts in Section 3.2.

[Changes]: Section 3.5, P1

“The global BC size distribution was obtained by 100 Monte Carlo (MC) simulations with the uncertainties in the parameters of mass to number conversion taken into account. The results reached steady state at 100 runs (Figure S18). The inventory is the sum of the emissions from millions of different aircrafts. Each MC run for the global size distribution already covers the calculation of millions of aircrafts and routes so the results converge at a lower number of runs compared to the results of individual aircrafts in Section 3.2.”

S3.6 – S3.8

As discussed earlier, I believe these three sections should be removed. I do not think they add to the goals of the paper and the required changes/additions to improve this paper should compensate for the reduced length.

[Response]: Thank you very much for your suggestion. Please refer to the response to Major Changes No.5.

References

- 1 Teoh, R., Stettler, M. E. J., Majumdar, A. & Schumann, U. Aircraft Black Carbon Particle Number Emissions - A New Predictive Method and Uncertainty Analysis. *European Aerosol Conference* (2017).

- 2 Richter, H. & Howard, J. B. Formation of polycyclic aromatic hydrocarbons and their growth to soot—a review of chemical reaction pathways. *Progress in Energy and Combustion Science* **26**, 565-608, doi:[https://doi.org/10.1016/S0360-1285\(00\)00009-5](https://doi.org/10.1016/S0360-1285(00)00009-5) (2000).
- 3 Liati, A. *et al.* Electron Microscopic Study of Soot Particulate Matter Emissions from Aircraft Turbine Engines. *Environmental Science & Technology* **48**, 10975-10983, doi:10.1021/es501809b (2014).
- 4 Howard, R. *et al.* Experimental Characterization of Gas Turbine Emissions at Simulated Flight Altitude Conditions. (Sverdrup Technology, Inc., Arnold Engineering Development Center, Arnold AFS, TN United States, 1996).
- 5 Durdina, L. *et al.* Assessment of Particle Pollution from Jetliners: from Smoke Visibility to Nanoparticle Counting. *Environmental Science & Technology*, doi:10.1021/acs.est.6b05801 (2017).
- 6 Wey, C. *et al.* Aircraft Particle Emissions eXperiment (APEX). (NASA Center for Aerospace Information, Hanover, MD, 2006).
- 7 Lobo, P. *et al.* Measurement of Aircraft Engine Non-Volatile PM Emissions: Results of the Aviation-Particle Regulatory Instrumentation Demonstration Experiment (A-PRIDE) 4 Campaign. *Aerosol Science and Technology* **49**, 472-484, doi:10.1080/02786826.2015.1047012 (2015).
- 8 Durdina, L. *et al.* Assessment of Particle Pollution from Jetliners: from Smoke Visibility to Nanoparticle Counting. *Environmental Science & Technology* **51**, 3534-3541, doi:10.1021/acs.est.6b05801 (2017).
- 9 Abegglen, M. *et al.* Effective density and mass–mobility exponents of particulate matter in aircraft turbine exhaust: Dependence on engine thrust and particle size. *Journal of Aerosol Science* **88**, 135-147, doi:<http://dx.doi.org/10.1016/j.jaerosci.2015.06.003> (2015).
- 10 Boeing. Current market outlook 2016-2035 Boeing (Market Analysis, Boeing Commercial Airplanes, Seattle, 2016).
- 11 Yim, S. H. L. *et al.* Global, regional and local health impacts of civil aviation emissions. *Environmental Research Letters* **10**, 034001 (2015).
- 12 Keuken, M. P., Moerman, M., Zandveld, P., Henzing, J. S. & Hoek, G. Total and size-resolved particle number and black carbon concentrations in urban areas near Schiphol airport (the Netherlands). *Atmospheric Environment* **104**, 132-142, doi:<http://dx.doi.org/10.1016/j.atmosenv.2015.01.015> (2015).
- 13 Hudda, N., Gould, T., Hartin, K., Larson, T. V. & Fruin, S. A. Emissions from an International Airport Increase Particle Number Concentrations 4-fold at 10 km Downwind. *Environmental Science & Technology* **48**, 6628-6635, doi:10.1021/es5001566 (2014).
- 14 Bruinink, A., Wang, J. & Wick, P. Effect of particle agglomeration in nanotoxicology. *Archives of Toxicology* **89**, 659-675, doi:10.1007/s00204-015-1460-6 (2015).
- 15 Kendall, M. & Holgate, S. Health impact and toxicological effects of nanomaterials in the lung. *Respirology* **17**, 743-758, doi:10.1111/j.1440-1843.2012.02171.x (2012).
- 16 Nel, A. E. *et al.* Understanding biophysicochemical interactions at the nano–bio interface. *Nature Materials* **8**, 543, doi:10.1038/nmat2442 (2009).
- 17 Seinfeld, J. H. Clouds, contrails and climate. *Nature* **391**, 837, doi:10.1038/35974 (1998).
- 18 Tesche, M., Achtert, P., Glantz, P. & Noone, K. J. Aviation effects on already-existing cirrus clouds. *Nature Communications* **7**, 12016, doi:10.1038/ncomms12016
<https://www.nature.com/articles/ncomms12016#supplementary-information> (2016).
- 19 Stuber, N., Forster, P., Rädcl, G. & Shine, K. The importance of the diurnal and annual cycle of air traffic for contrail radiative forcing. *Nature* **441**, 864, doi:10.1038/nature04877 (2006).
- 20 Kärcher, B. & Voigt, C. Susceptibility of contrail ice crystal numbers to aircraft soot particle emissions. *Geophysical Research Letters* **44**, 8037-8046, doi:10.1002/2017gl074949 (2017).

- 21 Moore, R. H. *et al.* Biofuel blending reduces particle emissions from aircraft engines at cruise conditions. *Nature* **543**, 411-415, doi:10.1038/nature21420 (2017).
- 22 Kärcher, B. Formation and radiative forcing of contrail cirrus. *Nature Communications* **9**, 1824, doi:10.1038/s41467-018-04068-0 (2018).
- 23 Eyers, C. J. *et al.* AERO2k Global Aviation Emissions Inventories for 2002 and 2025. (QinetiQ Ltd, Farnborough, Hampshire, 2005).
- 24 Stettler, M. E. J., Boies, A. M., Petzold, A. & Barrett, S. R. H. Global Civil Aviation Black Carbon Emissions. *Environmental Science & Technology* **47**, 10397-10404, doi:10.1021/es401356v (2013).
- 25 Wilkerson, J. T. *et al.* Analysis of emission data from global commercial aviation: 2004 and 2006. *Atmos. Chem. Phys.* **10**, 6391-6408, doi:10.5194/acp-10-6391-2010 (2010).
- 26 SAE. Aircraft Exhaust Nonvolatile Particle Matter Measurement Method Development. (2010).
- 27 SAE. Procedure for the Continuous Sampling and Measurement of Non-Volatile Particulate Matter Emissions from Aircraft Turbine Engines. (2018).
- 28 Smallwood, G. in *2016 CREATE-AAP Symposium on Atmospheric PM Research* (Vancouver, BC, 2016).
- 29 Wong, H.-W., Jun, M., Peck, J., Waitz, I. A. & Miake-Lye, R. C. Detailed Microphysical Modeling of the Formation of Organic and Sulfuric Acid Coatings on Aircraft Emitted Soot Particles in the Near Field. *Aerosol Science and Technology* **48**, 981-995, doi:10.1080/02786826.2014.953243 (2014).
- 30 Winther, M. & Rypdal, K. *EMEP/EEA air pollutant emission inventory guidebook 2016 - Update July 2017* (European Environment Agency, 2017).
- 31 Simone, N. W., Stettler, M. E. J. & Barrett, S. R. H. Rapid estimation of global civil aviation emissions with uncertainty quantification. *Transportation Research Part D: Transport and Environment* **25**, 33-41, doi:<http://dx.doi.org/10.1016/j.trd.2013.07.001> (2013).
- 32 Stettler, M. E. J., Eastham, S. & Barrett, S. R. H. Air quality and public health impacts of UK airports. Part I: Emissions. *Atmospheric Environment* **45**, 5415-5424, doi:<https://doi.org/10.1016/j.atmosenv.2011.07.012> (2011).
- 33 ICAO. *ICAO Aircraft Engine Emissions Databank*, <<https://www.easa.europa.eu/document-library/icao-aircraft-engine-emissions-databank>> (2016).
- 34 CrowdAnalytix. url: <https://www.crowdanalytix.com/> (visited on 11/06/2017). 2017).
- 35 ICAO. Annex 16 to the Convention on International Civil Aviation, Environmental Protection, Volume II Aircraft Engine Emissions. (2008).
- 36 Boies, A. M. *et al.* Particle Emission Characteristics of a Gas Turbine with a Double Annular Combustor. *Aerosol Science and Technology* **49**, 842-855, doi:10.1080/02786826.2015.1078452 (2015).
- 37 Stettler, M. E. J. & Boies, A. M. Aircraft non-volatile particle emissions: estimating number from mass. *18th ETH Conference on Combustion Generated Nanoparticles* (2014).
- 38 Dastanpour, R. *et al.* Improved sizing of soot primary particles using mass-mobility measurements. *Aerosol Science and Technology* **50**, 101-109, doi:10.1080/02786826.2015.1130796 (2016).
- 39 Dastanpour, R. & Rogak, S. N. Observations of a Correlation Between Primary Particle and Aggregate Size for Soot Particles. *Aerosol Science and Technology* **48**, 1043-1049, doi:10.1080/02786826.2014.955565 (2014).
- 40 Schmidt-Ott, A. New approaches to in situ characterization of ultrafine agglomerates. *Journal of Aerosol Science* **19**, 553-563, doi:[https://doi.org/10.1016/0021-8502\(88\)90207-8](https://doi.org/10.1016/0021-8502(88)90207-8) (1988).
- 41 Eggersdorfer, M. L., Kadau, D., Herrmann, H. J. & Pratsinis, S. E. Aggregate morphology evolution by

- sintering: Number and diameter of primary particles. *Journal of Aerosol Science* **46**, 7-19, doi:<https://doi.org/10.1016/j.jaerosci.2011.11.005> (2012).
- 42 Sorensen, C. M. The Mobility of Fractal Aggregates: A Review. *Aerosol Science and Technology* **45**, 765-779, doi:[10.1080/02786826.2011.560909](https://doi.org/10.1080/02786826.2011.560909) (2011).
- 43 Heintzenberg, J. Properties of the Log-Normal Particle Size Distribution. *Aerosol Science and Technology* **21**, 46-48, doi:[10.1080/02786829408959695](https://doi.org/10.1080/02786829408959695) (1994).
- 44 Brem, B. *et al.* Particulate Matter and Gas Phase Emission Measurement of Aircraft Engine Exhaust Final Report (04/2012 - 11/2015). (Empa, Advanced Analytical Technologies/ETH Zurich, Institute of Environmental Engineering, Zurich, Switzerland., 2016).
- 45 Lobo, P., Hagen, D. E., Whitefield, P. D. & Raper, D. PM emissions measurements of in-service commercial aircraft engines during the Delta-Atlanta Hartsfield Study. *Atmospheric Environment* **104**, 237-245, doi:<https://doi.org/10.1016/j.atmosenv.2015.01.020> (2015).
- 46 Wayson, R. L., Fleming, G. G. & Iovinelli, R. Methodology to Estimate Particulate Matter Emissions from Certified Commercial Aircraft Engines. *Journal of the Air & Waste Management Association* **59**, 91-100, doi:[10.3155/1047-3289.59.1.91](https://doi.org/10.3155/1047-3289.59.1.91) (2009).
- 47 Kugele, A., Jelinek, F. & Gaffal, R. Aircraft Particulate Matter Emission Estimation through all Phases of Flight. (EUROCONTROL Experimental Centre, Brétigny-sur-Orge, France, 2005).
- 48 Saikawa, E. *et al.* Comparison of emissions inventories of anthropogenic air pollutants and greenhouse gases in China. *Atmospheric Chemistry and Physics* **17**, 6393-6421, doi:[10.5194/acp-17-6393-2017](https://doi.org/10.5194/acp-17-6393-2017) (2017).
- 49 Barrett, S. *et al.* Guidance on the use of AEDT gridded aircraft emissions in atmospheric models. (Federal Aviation Administration, Washington, DC, 2010).
- 50 Schumann, U. *et al.* Influence of fuel sulfur on the composition of aircraft exhaust plumes: The experiments SULFUR 1-7. *Journal of Geophysical Research: Atmospheres* **107**, AAC 2-1-AAC 2-27, doi:[10.1029/2001jd000813](https://doi.org/10.1029/2001jd000813) (2002).
- 51 Petzold, A., Döpelheuer, A., Brock, C. A. & Schröder, F. In situ observations and model calculations of black carbon emission by aircraft at cruise altitude. *Journal of Geophysical Research: Atmospheres* **104**, 22171-22181, doi:[10.1029/1999jd900460](https://doi.org/10.1029/1999jd900460) (1999).

Response to Referee #2:

The authors appreciate the valuable comments by the reviewer. The comments help us significantly to improve the manuscript. A point-by-point response is as follows.

Annotation:

- (1) The comments of the reviewer are shown in *italic font*;
- (2) The responses to the comments are shown in green color;

This paper provides a novel method to estimate size-distributed particle emissions from aircraft, considering the different operating and flight conditions. The methods seem sound and the detail in the work is high. I think this is a very good piece of work that will be useful to the field. Particle number is very important, e.g. to cloud formation, and previous inventories have applied assumptions, without integrating all the knowledge available from the study of aircraft.

Authors compare aircraft emissions with those from other sectors. This is an interesting comparison but I don't think it is practically useful, because most aircraft emissions occur at altitude while most other primary emissions do not.

[Response]: Yes, the altitudes of the emissions are quite different. The main goal of the present study is to develop a number-based inventory of BC particles emitted from aviation. The comparison with road or other ground emissions help the readers to grasp the emission scale and relative magnitude. The impacts of these emissions surely are affected by the altitudes, which is out of the scope of the present work.

My one negative comment-- and this is a matter for the editor's decision-- is that despite the solid work done here, I am not sure the novelty and new insights are high enough for Nature Communications, and possibly a more specialty journal should be considered (Atmospheric Chemistry and Physics, Atmospheric Environment). I think that an investigation of size assumptions on cirrus impacts (for example) would follow on this work and would rate such novelty.

[Response]: Currently, the formation of contrail cirrus induced by aviation is still highly uncertain. This uncertainty is mainly due to the complexity of microphysics, e.g. different nucleation mechanisms for ice crystals involving ice nuclei. It is still a major and very difficult problem to be solved. As a first step, in this study we aimed to provide the size-resolved BC particle number emission as the fundamental data for further investigations. The data could better represent the aviation emissions and allow for more realistic simulations of the contrail formation, the radiative forcing of aviation induced clouds and effects of the emitted particles on natural clouds.

Response to Referee #3:

The authors appreciate the valuable comments by the reviewer. We carefully considered the comments and have revised the manuscript accordingly. The comments help us significantly to improve the manuscript. A point-by-point response is as follows.

Annotation:

- (1) The comments of the reviewer are shown in *italic font*;
- (2) The responses to the comments are shown in green color;
- (3) The explanations for the changes in the manuscript are shown in green with underlines;
- (4) The revised contents in the manuscript are shown in blue color;
- (5) The reference numbers in this response are based on the list at the end of this document, which are different from those in the manuscript.

Reviewer #3 (Remarks to the Author):

When you compared the magnitude of the aviation BC emissions with other global surface emissions, is the global surface emissions evaluated (estimated) in BC number to be comparable with the aviation emissions?

[Response]: The global surface emissions were estimated in a similar way as our estimation. The emissions were estimated based on the particle number emission factor (EF_{PN}) and size distribution (PSD)¹, which were determined from the literature. A lognormal distribution was adopted to represent the PSD. For the combustion-related sources, only the modes with GMD larger than 50 nm were assumed to be BC modes. The assumption was reasonable for diesel vehicles but may overestimate the size for gasoline vehicles, which have smaller particles. However, the BC emission of road transport was dominated by diesel vehicles². In the revised article, we added the discussion on the uncertain factor that may influence the results.

It should be noted that the standard measurements of on-road vehicle emissions are for particles larger than 23 nm³ instead of about 10 nm used for the aircraft engines, which may influence the comparison. This information is introduced in Section 3.7.

[Changes]: Section 3.8 P1

It should be noted that in the ECLIPSE inventory, the BC modes were assumed to have GMDs larger than 50 nm, a reasonable estimation for diesel vehicles, which are the dominant contributor to traffic emission², but it may underestimate the emissions from gasoline vehicles.

Section 3.7 P2

It should be noted that the standard measurements of on-road vehicle emissions are for particles larger than 23 nm³ instead of about 10 nm used for the aircraft engines.

References

- 1 Paasonen, P. *et al.* Continental anthropogenic primary particle number emissions. *Atmos. Chem. Phys.*

- 16**, 6823-6840, doi:10.5194/acp-16-6823-2016 (2016).
- 2 Milando, C., Huang, L. & Batterman, S. Trends in PM_{2.5} emissions, concentrations and apportionments in Detroit and Chicago. *Atmospheric Environment* **129**, 197-209, doi:<https://doi.org/10.1016/j.atmosenv.2016.01.012> (2016).
- 3 Giechaskiel, B. *et al.* Measurement of Automotive Nonvolatile Particle Number Emissions within the European Legislative Framework: A Review. *Aerosol Science and Technology* **46**, 719-749, doi:10.1080/02786826.2012.661103 (2012).

Reviewers' comments:

Reviewer #1 (Remarks to the Author):

The authors have appropriately addressed the concerns raised and I now think it is suitable for publication in a leading journal. There are no other issues of significance at this point.

Reviewer #4 (Remarks to the Author):

This manuscript uses emissions measurements from past ground tests and a limited number of measurements at cruise conditions to parameterize soot particle sizes as a function of measured mass emissions indices. This empirical model is then used to predict number emission indices as a function of a known mass emission index, engine fuel flow rate, and environmental conditions (assuming roughly that $\text{number} \sim \text{mass}/(\text{mode size})^{\text{(an assumed mass size exponent)}}$). The model is trained using largely ground-based data and verified against largely ground based data, and there are 10 empirical parameters that are used to fit the data (two coefficients in Eqn. 5; two coefficients in Eqn. 6; and two coefficients in Eqn. 7). A few data points for a CFM56-2C engine at cruise conditions ($\sim 30,000$ ft. altitude) are included in the comparison as well, but not enough to be meaningful. The model validation plot (Figure 2) shows that number emissions index for mostly ground-based measurements can be estimated by the empirical model to within a factor of 2-3 of the mostly ground-based test measurements. Having "validated" the parameterized model, the authors then use the parameterization to translate an existing global aviation mass emissions inventory into a number emissions inventory as well as to draw conclusions about the global aviation number emissions budget across all phases of flight operation (taxi, approach, take off, climb out, climb, cruise, and descent). The paper also compares the aviation number emissions to those from surface sources. Highly-precise estimates of the global particle number emission are given in the abstract as $(10.9 \pm 2.1) \times 10^{25}$ /year and $(6.06 \pm 1.18) \times 10^{14}$ /kg fuel. The geometric mean diameter of the particles is reported as 31.99 ± 0.8 nm.

In your review request, you asked me to assess whether the authors have responded adequately to Reviewer #3's comment as to whether the magnitude of aviation BC emissions with other global surface emissions are comparable. The authors reference a paper by Paasonen et al. (Atmos. Chem. Phys., 2016) as providing the number emissions of black carbon containing particles; however, the regions reported by that paper differ from those in the present paper. Presumably, the authors downloaded gridded particle number emissions netCDF files from

<http://www.iiasa.ac.at/web/home/research/researchPrograms/air/PN.html>, or were the mass-based baseline scenario emissions fields from <http://www.iiasa.ac.at/web/home/research/researchPrograms/air/ECLIPSEv5a.html> used for the comparisons in Section 3.8? The authors should be more explicit with what data were actually used for their comparisons and provide global emissions indices for comparison to the numbers reported in the abstract (Line 16). In addition, it does not appear that the model emissions data accounts for biomass burning emissions (e.g., open grassland and woodland fires), which are a huge source of global surface BC emissions, as comparisons are made only to “anthropogenic BC containing particles” (Line 530).

The editor should also be aware that the fundamental assumption in this paper that one can extrapolate ground test aircraft emissions data to all phases of flight despite significant differences in environmental and engine operating conditions is a very strong assumption that is neither well established nor validated in the present work (given a paucity of data points at taxi, approach, take off, climb out, climb, cruise, and descent conditions). Ground test data is very useful for exploring engine- and fuel-dependent emissions under the same set of controlled conditions and is also publicly available for some (mostly older) engines. Attempts to extrapolate ground measurements for these few engines to cruise conditions in the literature (e.g., Stettler et al., 2013; Peck et al., 2013; Abrahamson et al., 2016) have shown limited-to-mixed success in representing BC mass emissions. Given experimental sources of uncertainty, number emissions are even trickier to constrain than capturing integrated mass-based measurements.

A recent experimental paper quantified non-volatile particle number emissions indices for a single engine and single fuel as having variability about the mean EI of 26-39% (Moore et al., 2017), while the present study reports a mean BC particle number EI of 20% across a global dataset consisting of thousands of flights and presumably many different engine types and fuel compositions (Line 16). This range of variability/uncertainty seems unrealistically small. Similarly, the mean geometric mean diameter of 31.99 ± 0.8 nm seems overly precise given the dynamic range across engines and thrust conditions shown in Figure 1. Could one dispense with the complicated, 10-parameter empirical model and just assume a lognormal size distribution with geo. mean of 32 nm and geo. std. of 1.85 to convert mass to number and achieve similar results to those reported here? I think that this question gets at the heart of what the previous reviewers were getting at when they asked if the model is conceptually over fit; I do not understand how or why the statistical analysis presented in the rebuttal justifies the multi-parameter model.

In summary, I think that it is difficult to quantify with certainty the accuracy of the global estimates presented in this work or to characterize the present study as a significant advance in our understanding of the field. I disagree with the authors' statement on Line 370 that the mass-to-number conversion method was “validated” for relevant flight conditions in this manuscript. While

the present study is interesting and might be suitable for publication in a disciplinary journal, I am not sure that the present manuscript meets the editorial criteria for Nature Communications.

References:

Abrahamson et al., 2016 -- <https://pubs.acs.org/doi/abs/10.1021/acs.est.6b03749>

Moore et al., 2017 -- <https://www.nature.com/articles/nature21420.pdf>

Peck et al., 2013 -- <https://www.tandfonline.com/doi/abs/10.1080/10962247.2012.751467>

Stettler et al., 2013 -- <https://pubs.acs.org/doi/abs/10.1021/es401356v>

Response to Referee #4:

The authors appreciate the valuable comments by the reviewer. The comments help us to improve the manuscript. A point-by-point response is as follows.

Annotation:

- (1) The comments of the reviewer are shown in *italic font*;
- (2) The explanations for the changes in the manuscript are shown in green with underlines;
- (3) The revised contents in the manuscript are shown in blue color;
- (4) The reference numbers in this response are based on the list at the end of this document, which are different from those in the manuscript.

- 1. *In your review request, you asked me to assess whether the authors have responded adequately to Reviewer #3's comment as to whether the magnitude of aviation BC emissions with other global surface emissions are comparable. The authors reference a paper by Paasonen et al. (Atmos. Chem. Phys., 2016) as providing the number emissions of black carbon containing particles; however, the regions reported by that paper differ from those in the present paper. Presumably, the authors downloaded gridded particle number emissions netCDF files from <http://www.iiasa.ac.at/web/home/research/researchPrograms/air/PN.html>, or were the mass-based baseline scenario emissions fields from <http://www.iiasa.ac.at/web/home/research/researchPrograms/air/ECLIPSEv5a.html> used for the comparisons in Section 3.8? The authors should be more explicit with what data were actually used for their comparisons and provide global emissions indices for comparison to the numbers reported in the abstract (Line 16). In addition, it does not appear that the model emissions data accounts for biomass burning emissions (e.g., open grassland and woodland fires), which are a huge source of global surface BC emissions, as comparisons are made only to "anthropogenic BC containing particles" (Line 530).*

[Response]: Thank you very much for the comments. We utilized the gridded particle number emissions in the NetCDF files, and assigned the emissions into the regions defined in Figure S21. The total anthropogenic BC particle number emission is about 8.6×10^{27} per year. Please see **Change 1**.

The data only contain anthropogenic BC emissions. In this study, we only investigate the extra burden of BC emission generated from human activities, which is possible to be mitigated. Most of the biomass burnings emissions are from naturally occurring fires, which cannot be easily controlled and out of the scope of the present study.

[Change 1]: In the revised manuscript, the explanations for data of surface anthropogenic emissions were added into Section 3.8:

The gridded particle number emissions in the NetCDF files (Paasonen et al. 2016) were utilized, and the emissions were assigned into the regions defined in Figure S24. The total anthropogenic BC particle number emission on the ground is about 8.6×10^{27} per year.

- 2. *The editor should also be aware that the fundamental assumption in this paper that one can extrapolate ground test aircraft emissions data to all phases of flight despite significant differences in*

environmental and engine operating conditions is a very strong assumption that is neither well established nor validated in the present work (given a paucity of data points at taxi, approach, take off, climb out, climb, cruise, and descent conditions). Ground test data is very useful for exploring engine- and fuel-dependent emissions under the same set of controlled conditions and is also publicly available for some (mostly older) engines. Attempts to extrapolate ground measurements for these few engines to cruise conditions in the literature (e.g., Stettler et al., 2013; Peck et al., 2013; Abrahamson et al., 2016) have shown limited-to-mixed success in representing BC mass emissions. Given experimental sources of uncertainty, number emissions are even trickier to constrain than capturing integrated mass-based measurements.

[Response]: Thank you very much for the comments. Paucity of particulate matter emission data from jet engines, especially for cruise conditions, is a long-standing problem, which may remain unsolved in the near future. During the last decades, some valuable PM emission data at cruise conditions were accumulated mainly from the measurement campaigns conducted by the German Aerospace Center (Deutsches Zentrum für Luft- und Raumfahrt, DLR) and the National Aeronautics and Space Administration (NASA), e.g. the SULFUR Experiments (Schumann et al. 2002) and Alternative Fuel Effects on Contrails and Cruise Emissions Study (ACCESS) (Moore et al. 2017). As shown in Figure 2, not only the cruise BC particle number emissions from CFM56-2-C1 engine measured during the NASA ACCESS experiments were utilized to compare with the estimations, but also the cruise emissions from Mk501, CFM56-3B1, CF6-80C2A2, JT3D-3B and CFM56-5C2 engines measured during SULFUR experiments. Our estimations were in good agreement with these available measurements.

In this study, we have integrated most of the currently available relevant knowledge to make the best estimations for the time being. With the upcoming ICAO standard for aircraft engines produced after 2020/01/01, more particle number emission data will become available through the ground certification tests. This emission inventory will be continuously updated in the future when new data and knowledge emerge. In the revised manuscript, we stress that the current inventory is the best estimations for the time being based on the currently available data, and it has to be updated in the future. Please see Change 2.

It is hypothesized that the correlations between the particle size distribution (GMD and GSD) and the engine combustor inlet temperature (T_3) in conjunction with the black carbon mass emission index ($EI_m(BC)$) are applicable for both the ground and the cruise conditions. The physical basis for the assumption is that the BC particle generation inside the engines is dependent on the combustion conditions, and these parameters (T_3 and $EI_m(BC)$) can sufficiently represent the combustion conditions inside the engine.

The T_3 temperatures were estimated based on the environmental conditions (e.g. ambient temperature and pressure at different altitudes), aircraft speed (Mach number) and engine operation status (e.g. fuel flow rate, overall pressure ratio) as shown in equations S.14 to S.17. The $EI_m(BC)$ data from European Environment Agency (EEA) also included the influences of the flight operating and environmental conditions using the Advanced Emission Model (AEM). The influences of the different environmental and engine operating conditions were included in these parameters.

In the revised manuscript, firstly an additional dataset was included to support the assumption. It is a dataset of the BC particle size distributions measured for a typical turbofan engine operated at different simulated altitudes (from sea-level-static to 15.2 km, about 0.8 Mach) in a test facility at Arnold

Engineering Development Center (AEDC) (Howard et al. 1996). The dataset showed that the size distribution of BC particles was strongly dependent on the engine inlet temperature T_3 , but not on the various simulated altitudes with significantly different environmental conditions inside the test facility. Details about the experiments can be found in Change 3.

Then, in order to support that the correlation can also be applied to different engine operating conditions, the estimations were compared with a new dataset (Boies et al. 2015) from a jet engine with a double annular combustor (CFM56-5B4-2P), which is a special engine, not widely used in the current fleets. The operating conditions were totally different from the engines with a single annular combustor utilized to develop the correlation. The proposed correlation successfully captured the unique changes of the GMD and the number emissions of the BC particles generated in this double annular engine, even though the method was developed based on the engines with a single annular combustor. The results demonstrate the potential abilities of the proposed method to estimate the BC particle size distribution and the number emission for significantly different engine designs and operating conditions. Details about the experiments can be found in Change 4.

The reviewer brought up the uncertainties of the BC mass emissions which are the basis of our estimation of the particle number emission. In the revised manuscript, sensitivity tests were conducted to investigate the influence of BC mass emissions on the particle number emission and the size distribution. The sensitivity tests indicated that our estimations of the BC number emission and size distribution were robust, even when we applied substantial changes of the BC mass emission from 33% to 200% of the original EEA BC mass emissions. For details about the tests please see Change 5. The uncertainties of the estimations are not the same concept as the “variabilities” among different engines and flight conditions. We have provided detailed explanations in the response to Comment No. 3 below.

It should be also mentioned that, the dataset utilized to develop the correlations included 7 types of aircraft engines from the major manufacturers to cover a wide range of BC emissions: JT8D-219, PW2037, PW4168 (Pratt & Whitney Aircraft Group), CF6-80A2, CF6-80A2 (General Electric), CFM56-7B24/3, CFM56-2-C1 (CFM International), as described in Table S1. These engines have noticeably different designs and engine parameters, e.g. maximum fuel flow rate, maximum rated thrust and pressure ratio. As a result, the combustion conditions were already quite different in these engines, even though most of the measurements were conducted on the ground. The BC particle generation inside the engines is dependent on the combustion conditions. The proposed correlation can simultaneously provide satisfactory estimations for all of these engines, despite the substantial variabilities of the combustion conditions. The results indicate the potential applicability of the proposed method for different engine conditions.

[Change 2]: In the revised manuscript, Section 4, we stress that the current inventory is the best estimations for the time being based on the currently available data, and it has to be updated in the future:
Most of the currently available relevant data and knowledge were integrated to make the best estimations for the time being, but in view of the limited data, this emission inventory and the proposed methods have to be continuously updated in the future when new data and knowledge emerge.

[Change 3]: In the revised Supporting Information Section S2.2, we added a new part to investigate the dependence of BC particle size on \$T_3\$ at different simulated altitudes:
In the test experiments, the BC particle size distributions were measured for a typical turbofan engine

operated at different simulated altitudes (from sea-level-static to 15.2 km, about 0.8 Mach) in a test facility at Arnold Engineering Development Center (AEDC) (Howard et al. 1996). The dataset showed that the size distribution of BC particles was strongly dependent on the engine inlet temperature T_3 . The fitted relation between the particle size and T_3 is applicable for different simulated altitudes inside the test facility.

The detailed results are shown in Figure S2. The GMDs of the size distributions for soot particles were extracted and calculated by Durdina et al. (Durdina et al. 2017). A quadratic equation was utilized to fit the data at different simulated altitudes, which is similar to the GMD- T_3 relation in Figure S1. It is shown that all the data are in a good agreement with the fitted line, in spite of the different operating conditions and the altitudes.

It is notable that the sea-level-static condition (blue dots) and the 12.2 km 0.8 Mach condition (yellow dots) have the similar tested combustor inlet temperatures T_3 between 500 K and 750 K. The measured particle sizes of these two conditions were also quite comparable to each other, both with GMDs between 25 and 35 nm. The residuals between the measurements and the fitted relation were calculated for all the different altitudes as shown in the right panel of Figure S2. There is no significant correlation between the residuals and the altitudes.

Figure S2 Black carbon particle size at different simulated altitudes from sea level to 15.2 km: (a) Changes of the GMD of BC size distribution with T_3 ; (b) the residuals between the measurements and the fitted relation at different altitudes.

Table S3 ANOVA table for the residuals at different altitudes

Source	Sum of squares (SS)	Degree of freedom (df)	Mean squares (MS)	F-statistic	p value
Groups	5.60	4	1.40	0.5	0.73
Error	47.24	17	2.78		
Total	52.84	21			

The analysis of variance (ANOVA) test was conducted to investigate the differences among the residuals at different altitudes, which indicates that there is no statistically significant difference among them. The ANOVA table is shown in Table S3, with a p value of 0.73, which means there is a high risk (73%) to claim that the residuals are different. As a result, the experimental results indicate that the dependence of BC particle size on T_3 is applicable to different simulated altitudes.

Figure S3 Comparison between the data at different simulated altitudes and the data utilized to develop the GMD- T_3 correlation. The shaded area represents the 95% prediction interval.

In Figure S3, the data at different simulated altitudes were compared with the data utilized to develop the GMD- T_3 correlation. The results indicate that new dataset (colorful dots) agrees well with the utilized dataset (grey dots). All of the new data are within 95% prediction interval of the fitted correlation.

[Change 4]: In the revised Supporting Information, we added a new section (Section S2.9) to evaluate the proposed relation using the data from an engine with a double annular combustor:

S2.9 Independent validation using an engine with a double annular combustor

In order to investigate the performance of the proposed method for different engine designs, the estimations are compared with a new dataset (Boies et al. 2015) from a jet engine with a double annular combustor (CFM56-5B4-2P), which is a special engine, not widely used in the current fleets. There are one outer (pilot) annulus and one inner (main) annulus in this engine. For low thrusts, only the pilot annulus is fueled. For high thrusts, the main annulus is also triggered. As a result, the operating conditions are totally different from the engines with a single annular combustor utilized to develop our correlation.

Figure S7 a) shows the changes of the GMD with the thrust. It shows that the estimations (solid line) follow the measurements (red dots). All the measurements are within the 95% confidence prediction interval of the estimations. Before the start of the main annulus, the GMD increases with the thrust from about 20 nm to about 30 nm. The particle size greatly drops from 30 nm to about 20 nm after the start of the main annulus, because the utilization of the main annulus enhances the oxidation of the generated BC particles, leading to a much lower BC mass emission. As a result, the GMD becomes much smaller according to the proposed correlation Equation (6) in this study. There is a slight increase in GMD from about 25% to 100% thrust, due to the increase of T_3 . The correlation coefficient (r) between the estimated GMDs and the measurements is as high as 0.9 as shown in Figure S7 b).

The comparisons between the estimated BC particle number emissions and the measurements are shown in Figure S7 c) and d). The proposed relation captures the changes of $E_{I_n}(\text{BC})$ with the thrust reasonably well. For the low thrusts, the predictions are generally underestimated, but most of the measurements are still within a factor of two of the estimations. For high thrusts after the start of the main annulus, the estimations are in a better agreement with the measured data. Nearly all the data are within

the standard deviation of the predictions. The overall correlation coefficient (r) between the predictions and measurements is 0.94. The main statistical metrics are comparable with those from the engines with a single annular combustor as shown in Figure 2. The GSD data are not fully available for the double annular engine. Only the data for 6%, 17% and 31% of the maximum thrust are available, so we only compared the three data points with the data which were utilized to develop the GSD- $EI_m(BC)$ correlation and the fitted relation as shown in Figure S8. The new data (red dots) agree well with the utilized dataset (grey dots). All of the new data are within 95% prediction interval of the fitted correlation. At 31% thrust, the main annulus started, leading to lower BC mass emission. The measured GSD at 31% is smaller than that at 17%, which indicates a strong relation between GSD and $EI_m(BC)$.

Figure S7 Evaluation of the estimations for a double annular combustor (CFM56-5B4-2P): (a) the changes of the GMD with thrust level; (b) the direct comparison between the measured and estimated GMDs; (c) the changes of $EI_n(BC)$ with thrust level; (d) the direct comparison between the measured and estimated $EI_n(BC)$, the dashed lines are 1:2 (or 2:1) ratio lines and the dotted lines are 1:3 (or 3:1) ratio lines.

In general, the proposed method successfully captures the unique changes of GMD of the BC particles generated in this double annular engine, even though the method is developed based on the engines with a single annular combustor. The results demonstrate the necessities to include both the T_3 and $EI_m(BC)$ as the predictors for GMDs. They also show the potential abilities of the proposed method to estimate the BC particle size distribution and the number emission for significantly different engine designs and

operating conditions.

Figure S8 Comparison between the measured GSD for the engine with a double annular combustor (CFM56-5B4-2P) and the utilized data to develop the GSD- EI_m (BC) relation. The shaded area represents the 95% prediction interval.

[Change 5]: We added a new section S7.4 in the revised Supporting Information to investigate the influences of BC mass emissions on the BC particle number emission inventory:

S7.3 Sensitivity tests for the influences of BC mass emissions

The BC mass emission directly influences the estimation of particle number emission. Sensitivity tests were conducted to investigate the influence of BC mass emissions on the particle number emission and the size distribution. The estimations based on the EEA dataset were utilized as the baseline in the sensitivity tests. Besides the baseline, four different cases with substantial changes in BC mass emissions were included. It was assumed that the BC mass emissions were respectively 33% (1/3, 3.2 Gg), 50% (1/2, 4.8 Gg), 150% (14.3 Gg) and 200% (19.1 Gg) of the baseline emission (9.5 Gg). The range of these values is able to cover most of the BC mass emissions in the literature, as shown in Table 2.

Figure S21 shows the results of the sensitivity tests. The changes of particle number emissions (Figure S21 a) are much smaller than the corresponding changes of mass emissions, because higher EI_m (BC) induces larger and heavier particles, which alleviates the increase of BC particle number and makes the particle number emission relatively stable. It is similar when EI_m (BC) is lower with smaller and lighter particles. Figure S21 a) shows that, for the cases with 50% and 150% mass emissions, the resulted particle number emissions are respectively 7.7×10^{25} and 13.4×10^{25} , and the deviations from the baseline estimation (10.9×10^{25}) are about 1.5 and 1.2 times of the standard deviation of baseline, σ_{num} (2.1×10^{25}). Even with the largest changes (33% and 200%) in BC mass emissions, the variations of the estimated particle number emissions (6.0×10^{25} and 15.5×10^{25}) are still near two times of σ_{num} , which are indicated as the dashed lines. Generally, the number emission is more sensitive when the BC mass emission is lower.

Figure S21 Results of the sensitivity tests for the influences of BC mass emissions. The error bars are the estimated standard deviations of the MC simulated results (68% confidence interval), and the dashed lines are two times of the standard deviation (95% confidence interval).

The GMDs of all the emitted particles are 30.04 and 33.37 nm for the cases with 50% and 150% BC mass emissions. The changes are respectively -1.95 and 1.38 nm from the baseline (31.99nm), about -2.4 and 1.7 times of the estimated standard deviation σ_{gmd} (0.8 nm). For the 200% case, the change of GMD (2.37 nm) is still within $3\sigma_{gmd}$. For the 33% case, deviation from the baseline is stronger with the change of GMD -3.1 nm, beyond $3\sigma_{gmd}$. The GMD is more sensitive to the change of BC mass emission than the particle number emission.

The changes of GSDs are quite small. As shown in Figure S21 c), nearly all the changes of GSDs are still

within 2 times of the estimated GSD standard deviation σ_{gsd} (0.016), except the 33% case. For the 200% case, the GSD (1.87) is increased by 0.02 relative to the baseline (1.85) and the change is only about 1.25 times of σ_{gsd} . For the 50% case, the GSD is 1.82, with a difference of $-1.88\sigma_{gsd}$ from the baseline.

In summary, the sensitivity tests indicate that our estimations of the BC number emission and size distribution were robust, even when we applied substantial changes of the BC mass emission from 33% to 200% of the baseline. The sensitivity tests demonstrate that the BC particle number emission and size distribution are rather robust, considering the substantial changes of the BC mass emissions used in the tests. Therefore we consider the uncertainties are at reasonable based on the results.

- *3. A recent experimental paper quantified non-volatile particle number emissions indices for a single engine and single fuel as having variability about the mean EI of 26-39% (Moore et al., 2017), while the present study reports a mean BC particle number EI of 20% across a global dataset consisting of thousands of flights and presumably many different engine types and fuel compositions (Line 16). This range of variability/uncertainty seems unrealistically small. Similarly, the mean geometric mean diameter of 31.99 ± 0.8 nm seems overly precise given the dynamic range across engines and thrust conditions shown in Figure 1.*

[Response]: We apologize for that we did not provide clear explanations for the uncertainties in this study. The uncertainties of the estimations are not the same concept as the “variabilities” among different engines and flight conditions as shown in Figure S18 in Change 6. The variabilities are usually calculated as the standard deviations of the corresponding parameters from all the aircrafts.

In this study, the global GMD and GSD were obtained by fitting the distribution of the BC particles emitted from all the aircrafts during one year, and they were not the mean of the values of all the aircrafts. The uncertainties of EI_n s, GMDs and GSDs were not the standard deviation of values of all the aircrafts. Instead, we assessed the uncertainties of our estimations induced by the utilized uncertain parameters based on Monte Carlo (MC) Simulations, as shown in Figure S18. More details about the MC simulations and uncertainty estimation were added in Change 6.

The variability among different engine types and flight conditions is an inherent property, which cannot be reduced or eliminated, but the uncertainties can be reduced if we have better data or knowledge on the aviation BC particle emissions.

These uncertainties are only discussed for the overall statistical parameters, e.g. the global total particle number emission, the total emissions in different regions and the emissions from different flight phases. But these uncertainties of the overall parameters were calculated based on the uncertainties of individual engines through MC simulations. As shown in Figure 2 of the manuscript, our estimated uncertainties of $EI_n(BC)$ for individual engines are comparable with the measured values, indicated by the vertical and horizontal error bars. In fact, these measurements are the data measured by Moore et al. (Moore et al. 2017), mentioned by the reviewer.

The uncertainties of the global GMD and GSD appear to be small, but it should be noted that the uncertainties were obtained by fitting the distribution of the particles emitted from millions of flights. The statistical theory (law of large numbers) indicates that the mean of a large number of random variables from the same distribution is usually quite stable, and it will approach the actual mean of the distribution.

As a result, the distribution of the sum of all the emissions is relatively stable. We also provided sensitivity tests in Change 5 to show that our estimations are robust. The clarification for the uncertainties was added in the revised manuscript, please see Change 6.

We also calculated the variabilities as the standard deviations of the emissions from all the 27 million flights at the seven sub-phases (taxi, take-off, climb-out, approach, climb, cruise and descent). The results show that the estimated uncertainties are about 17% to 21% of the variabilities. Please see Change 7 for more details.

About the accuracy of the estimations, our estimations of fuel consumption, gaseous emissions and BC mass emission have a good agreement with the existing ones in the literature as shown in Table 2 of the manuscript. The fuel consumption is the basis for the emission inventory, and our estimation (180 million metric tons, Mt) is very close (within 4%) to the value reported by ICAO for the year of 2006, which is 187 Mt (Hupe et al. 2010).

As shown in Table 2, our estimation of BC mass emission is in line with other inventories in the literature. If we exclude the extreme values from AEIC, the standard deviation of the BC mass estimations is 2.55 Gg per year, which is about 27% of our estimation (9.5 Gg).

So far, the uncertainties have been calculated based on the parameters including estimated GMD, GSD and ϵ . In the sensitivity tests in Change 5, we included the influences of BC mass emissions to give a more comprehensive evaluation of the uncertainties. The sensitivity tests demonstrate that the BC particle number emission and size distribution are rather robust, despite the substantial changes of the BC mass emissions used in the tests. Therefore we consider the uncertainties are reasonable based on the results.

[Change 6]: In the revised manuscript, Section 3.5, added a new paragraph to discuss the variabilities of the emissions from all the 27 million flights:

The variabilities were also calculated as the standard deviations of the emissions from all the 27 million flights at the seven sub-phases (taxi, take-off, climb-out, approach, climb, cruise and descent). The variabilities of GMD and GSD are about 4.8 nm and 0.08, which are about five times of the estimated uncertainties. More details about the uncertainties and variabilities can be found in Section S7.3 in SI.

In the revised manuscript, we added a sentence to introduce the variabilities:

The variabilities of GMDs and GSDs for all the flights are about 4.8 nm and 0.08, respectively.

We add a new section S7.3 in the revised Supporting Information to explain the differences between the uncertainties and the variabilities in this study:

S7.3 Monte Carlo simulations and uncertainty estimations

By Monte Carlo (MC) Simulations, different total emissions were obtained based on an ensemble of the uncertain parameters (GMD, GSD and particle mass parameters of different engines) utilized to convert mass into number emissions, as shown in Figure S18. The global BC size distribution was obtained by 100 MC simulations. The number of MC runs is sufficient to reach steady state as shown in Figure S20.

In order to estimate the uncertainties, we first integrated the global distributions of all the aircraft BC emissions from millions of flights as shown in Figure S18. The emitted particles from each aircraft were

divided into 16 size bins as shown in Table S9. The global inventory with 16 size bins was calculated by integrating the size-resolved emissions from all the aircraft. Namely, we summed all the distributions for every aircraft over a full year. Finally, the uncertainties were calculated based on the MC results to quantify the reliability of these overall statistics (GMD, GSD and EI_n).

The uncertainties of the estimations are not the same concept as the “variabilities” among different engines and flight conditions as shown in Figure S18. The variabilities are usually calculated as the standard deviations of the corresponding parameters from all the aircraft. In this study, the global GMD and GSD were obtained by fitting the distribution of the BC particles emitted from all the aircraft during one year, and they were not the mean of the values of all the aircraft. The uncertainties of EI_n s, GMDs and GSDs were not the standard deviation of values of all the aircraft. Instead, we assessed the uncertainties of our estimations induced by the utilized uncertain parameters based on Monte Carlo (MC) Simulations, as shown in Figure S18. As a result, the uncertainties here do not represent the variabilities among different flights, but they indicate the influence of the uncertain parameters on the estimations.

Figure S18 Schematic diagram for the differences between the uncertainties of estimations and the variability among different emissions from various engines and flight conditions. Number N represents the amount of the civil flights in one year, about 27.34 million; number M represents of number MC runs, 100 in this study.

The mean and standard deviation from the MC runs are shown in Figure 4. The GMD and GSD of the global BC size distribution were obtained by fitting the particle numbers in the size bins into the lognormal distribution for each MC run, and the uncertainties of GMD and GSD were estimated from these MC runs.

[Change 7]: In Section S7.4 in the revised Supporting Information, we compared the variabilities and the uncertainties:

Table S7 Comparisons between variabilities among the 27 million flights at 7 sub-phases and the uncertainties of the estimated global emissions. The ratio is between uncertainties and variabilities.

	Particle number emission index $EI_n(BC)$, (#/kg-fuel)	Geometric mean diameter GMD, (nm)	Geometric standard deviation GSD
Variability	6.92×10^{14}	4.78	0.078
Uncertainty	1.18×10^{14}	0.8	0.016
Ratio	0.17	0.17	0.21

In Table S7, the variabilities were calculated as the standard deviations of the emissions from all the 27 million flights at the seven sub-phases (taxi, take-off, climb-out, approach, climb, cruise and descent). The uncertainties are about 17% to 21% of the variabilities.

- 4. *Could one dispense with the complicated, 10-parameter empirical model and just assume a lognormal size distribution with geo. mean of 32 nm and geo. std. of 1.85 to convert mass to number and achieve similar results to those reported here? I think that this question gets at the heart of what the previous reviewers were getting at when they asked if the model is conceptually over fit;*

[Response]: As explained above, the global GMD and GSD were obtained by directly fitting the global distribution of all the emitted BC particles, so they can be used to estimate the particle number emission based on the conversion method, if the particle mass parameters are available. The parameters for particle mass (C and ϵ) are thrust dependent. The total emission is a mixture from different thrust conditions. If we utilize the parameters for the particles with a GMD of about 35 nm for the conversion, the total BC number is estimated as 10.6×10^{25} , close to the result in the manuscript.

The parameters of the size distribution were obtained by bottom-up method based on the proposed correlations. Without these correlations, the global parameters cannot be obtained. They are only valid for the estimation of global emissions. For different regions and flight conditions (e.g. taking-off, cruise), the parameters are different as shown in Figure 4 and Figure 5.

For individual engines, our estimated uncertainties are comparable with the measurements as shown in Figure 1 and Figure 2.

- 5. *I do not understand how or why the statistical analysis presented in the rebuttal justifies the multi-parameter model.*

[Response]: The potential overfitting problem refers to whether the proposed GMD- T_3 & EI_m relation includes random noises of the data, which will deteriorate the quality of GMD estimations. The fitting performance can always get better with the increasing complexity of the fitted model. But when the complexity is too high, the fitted relation will capture the noises in the training data to the extent that it will fail to reasonably estimate new data. The bias-variance trade-off is a standard method to evaluate the overfitting problem.

Here we utilize a standard polynomial example to illustrate the over-fitting problem and how the bias-variance trade-off works, as shown in Figure R1. The true relation is quadratic. The poor fitted relation is linear. The over-fitted relation is with a polynomial order of 5. The scattered data were obtained by adding random noise to the true values. Only the training dataset was fitted, and the validation dataset was used to evaluate the fitted relation.

When the complexity is low, namely the linear relation, the fitted line (the green dot line) fails to represent the scattered data in the training dataset (red dots), which is usually referred to as high bias. The performance of the fitted relation is poor. Both the errors (RMSE) of training and validation dataset are high, as shown in Table R3.

The fitted quadratic relation (the back dashed line) is close to the true relation, and both the training and validation errors decrease. If we further increase the complexity of the model (the dot-dashed blue line), it

will be closer to the data in the training dataset (red dots), but it will deviate from the data in the validation dataset (yellow triangles). As result, the training error decreases, but the validation error increases again, which is usually referred to as high variance. The over-fitted relation cannot give reasonable predictions. A good fitted model should achieve the balance between bias and variance. In this study, we used the bias-variance trade-off to assess the fitted relations. In Figure S5, the results indicated that the bias and variance of our proposed model are both very low, so the model was the optimal model based on the current dataset.

Figure R1 Example for over-fitting problem and bias-variance trade-off evaluation

Table R1 Root Mean Square Error (RMSE)

	Training dataset (bias)	Validation dataset (variance)
Poor fitted relation	6.88	6.80
Well fitted relation	1.92	2.50
Over-fitted relation	1.70	7.71

- 6. I disagree with the authors' statement on Line 370 that the mass-to-number conversion method was "validated" for relevant flight conditions in this manuscript.

[Response]: The word "validated" has been deleted, please see Change 8.

[Change 8]: Deleted the word "validated" on Line 370

Through the conversion method, the developed BC mass emission inventory is converted into a number emission inventory.

References

- Boies, Adam M., Marc E. J. Stettler, Jacob J. Swanson, Tyler J. Johnson, Jason S. Olfert, Mark Johnson, Max L. Eggersdorfer, Theo Rindlisbacher, Jing Wang, Kevin Thomson, Greg Smallwood, Yura Sevcenco, David Walters, Paul I. Williams, Joel Corbin, Amewu A. Mensah, Jonathan Symonds, Ramin Dastanpour, and Steven N. Rogak. 2015. "Particle Emission Characteristics of a Gas Turbine with a Double Annular Combustor." *Aerosol Science and Technology* 49(9):842-55.
- Durdina, Lukas, Benjamin T. Brem, Ari Setyan, Frithjof Siegerist, Theo Rindlisbacher, and Jing Wang.

2017. "Assessment of Particle Pollution from Jetliners: from Smoke Visibility to Nanoparticle Counting." *Environmental Science & Technology* 51(6):3534-41.
- Howard, R., R. S. Hiers, P. D. Whitefield, D. E. Hagen, J. C. Wormhoudt, R. C. Miake-Lye, and R. Strange. 1996. "Experimental Characterization of Gas Turbine Emissions at Simulated Flight Altitude Conditions." Arnold AFS, TN United States: Sverdrup Technology, Inc., Arnold Engineering Development Center.
- Hupe, Jane, Blandine Ferrier, Theodore Thrasher, Ebad Jahangir, Tetsuya Tanaka, Helene Manzoni, Lorenzo Gavilli, Angelika Fuchs-Ledingham, and Vanessa Muraca. 2010. "ICAO Environmental Report 2010." in *CAO Environmental Report* Montréal, Québec, Canada: International Civil Aviation Organization.
- Moore, Richard H., Kenneth L. Thornhill, Bernadett Weinzierl, Daniel Sauer, Eugenio D'Ascoli, Jin Kim, Michael Lichtenstern, Monika Scheibe, Brian Beaton, Andreas J. Beyersdorf, John Barrick, Dan Bulzan, Chelsea A. Corr, Ewan Crosbie, Tina Jurkat, Robert Martin, Dean Riddick, Michael Shook, Gregory Slover, Christiane Voigt, Robert White, Edward Winstead, Richard Yasky, Luke D. Ziemba, Anthony Brown, Hans Schlager, and Bruce E. Anderson. 2017. "Biofuel blending reduces particle emissions from aircraft engines at cruise conditions." *Nature* 543(7645):411-15.
- Paasonen, P., K. Kupiainen, Z. Klimont, A. Visschedijk, H. A. C. Denier van der Gon, and M. Amann. 2016. "Continental anthropogenic primary particle number emissions." *Atmos. Chem. Phys.* 16(11):6823-40.
- Schumann, U., F. Arnold, R. Busen, J. Curtius, B. Kärcher, A. Kiendler, A. Petzold, H. Schlager, F. Schröder, and K. H. Wohlfrom. 2002. "Influence of fuel sulfur on the composition of aircraft exhaust plumes: The experiments SULFUR 1-7." *Journal of Geophysical Research: Atmospheres* 107(D15):AAC 2-1-AAC 2-27.

Reviewers' comments:

Reviewer #4 (Remarks to the Author):

The authors have put a lot of effort into addressing my previous comments, which is appreciated. Despite this, some specific concerns remain and are detailed below. I also think that it's important to not overstate what the authors have actually done by, for example, making strong statements like those on Lines 571-572 implying that a size-resolved number emissions inventory has been developed for the first time. This work developed a parameterisation of soot particle mode size in terms of engine T3 and EI_BCM by fitting published ground data, and this parameterisation is then used with an existing mass-based global emission inventory to convert the mass basis to a number basis, assuming the particles consist of a single, lognormal size distribution. The present work relies heavily on the pre-existing mass inventory for most of the heavy lifting. Prior number emissions inventories are also referenced for comparison (e.g., AERO2k developed by EUROCONTROL), so this is also not the first number emissions inventory. This study does place important order-of-magnitude-type constraints on the number of particles emitted by aircraft regionally and worldwide and is valuable for that reason; however, it's not clear to me how this paper might change/influence thinking in the field (a Nature Comm. review criterion). Figure 3 in the present work is visually pretty much the same as Figures 4 and S12-S14 in Stettler et al. (2013), except that the present Figure 3 is in number units instead of mass units. The underlying global mass emissions here appear to be quite different than prior work (e.g., Stettler et al.), while the number emissions are 2.5x that of AERO2k (Line 387). Who is right? The low number emission in AERO2k is not "caused" by the low mass emission; presumably both number and mass emissions are both lower for some common causal reason. It would be helpful to remember in this manuscript that many inventories try to predict emissions from aircraft, engine, and environmental conditions; meanwhile this study predicts emissions from environmental conditions and emissions themselves. Consequently, the method already has a leg up on the competition because it is starting with a slightly different form of the answer it seeks to constrain (in this case using a priori knowledge of BC mass emissions to predict BC number emissions).

Specific comments:

1) In the previous review, I asked the authors to place their emissions factors in the context of terrestrial BC emissions, including the significant sources of biomass burning. The authors response is that since biomass burning emissions are from "naturally occurring fires" they are "out of scope of the present study". I think that this answer is evasive and non-responsive. I believe (as did the prior reviewer) that it is within the scope of the present study to contextualize the reported aviation emissions by comparison to the (expected) much larger surface emissions source. The fundamental

question is do aviation emissions matter? Are they a significant source of BC mass or number, or is aviation a negligible source? Since this is a global study as evidenced by the paper title, the global emissions seem like the appropriate metric on which to evaluate importance.

In a comprehensive review, Bond et al. (2016) report total BC emissions for the industrial and pre-1750's era of 6100 and 1400 Gg/yr, respectively. The total for all sources is 7500 Gg/yr. The mass emissions from aviation reported in the present study are 9.5 Gg/yr, which is 0.13% of all BC emissions and 0.16% of anthropogenic emissions reported by Bond et al. on a mass basis.

Bond et al. also report typical mass size distributions for biomass burning and industrial BC emissions that have mode diameters near 200 nm. To first order, we might assume a monodisperse aerosol with this mode size and unit density to translate the Bond et al. mass emissions to number emissions. From Table 6 of Bond et al, then we have:

Mass (Gg/yr.)	Number (#/yr.)
---------------	----------------

All Sources:

Energy	4770	9.1e27
--------	------	--------

Open Burning	2760	5.3e27
--------------	------	--------

Total All	7530	1.4e28
-----------	------	--------

Pre-Industrial:

Energy	390	7.4e26
--------	-----	--------

Open Burning	1020	1.9e27
--------------	------	--------

Total Pre-industrial	1410	2.7e27
------	--------

Industrial:

Energy	4380	8.4e27
--------	------	--------

Open Burning	1740	3.3e27
--------------	------	--------

Total Industrial	6120	1.2e28
------------------	------	--------

From this, admittedly coarse, estimation method, it is worthwhile to note that the energy-related industrial number emissions of $8.4e27$ are in reasonable agreement with the $8.6e27$ number reported on Line 534 of the present manuscript. Now, accounting for the additional emissions from open burning, the contribution of aviation number emissions to anthropogenic emissions is $1.09e26/1.2e28 = 0.9\%$. The contribution of aviation number emissions to present day BC emissions is $1.09e26/1.4e28 = 0.8\%$.

2) The paragraph transition on Lines 72-74 is awkward. The new paragraph seems out of place here.

3) On Lines 87-88, the new number emissions inventory is mentioned for the first time. It would be good to have a link here to where it can be downloaded.

4) On Line 131, it is suggested that the mass EI is estimated from Equation 3, which is not right. In this study, EI is known from an existing global emission inventory.

5) Where is the plot mentioned on Lines 176-177?

6) On Line 425-426, it is stated that "Our results demonstrate that the global aviation BC particle emission approximately follows a lognormal distribution as shown in Figure 4a and b...". This is circular reasoning as the lognormal size distribution is imposed from Equation 4. Of course this is not evident to the reader from the manuscript, but rather is buried on Pg. 20 of a 42-page supporting information document.

7) The numbers reported in the paragraph at Lines 455-463 range pretty widely. Are all of these geometric number mean diameters? Similarly, are these only for the BC particles or do they include volatile particles (e.g., sulphuric acid and organics)? I suspect that the 11.8 nm GMD reported on Line 458 is actually capturing sulphur+organics (rather than BC-only) GMD. Similarly, the GMD of 60 nm on Line 460 is assumed by Ref. 65 (without validation), but is not the nominal value of 38 nm actually reported by the underlying reference (Page 11 of Barrett et al., 2010). Moore et al. (2017) report bimodal size distributions for aircraft take-off plumes that they measured for different aircraft types. The larger of the two modes likely is dominated by the contribution of the soot particles, while the smaller mode is composed of the volatile particles. How do the aircraft-specific, take-off size distributions in this study compare to those reported by that group?

8) Figure S7 is plotted in terms of thrust, while most other similar Figures S2 and S3 are shown in terms of T3. Please show these data in terms of T3 as well (since that is the authors' chosen

independent variable in the parameterisation) so that they can be compared to each other. What is "thrust" and how was it determined?

9) The fundamental concern that I have with this paper's conclusions is that if the mass EI is uncertain / poorly constrained, then the derived number EI will be too (even if the derived number is relatively stable due to offsetting errors in mass EI and mean size). The authors note that their estimates are based on a global mass emission of 9.5 Gg/yr, while AERO2k is based on 3.9 Gg/yr. Stettler et al., (2013) suggest that BC emissions are likely to be closer to 16.9 Gg/yr via the new FOX method, while also comparing their results to a literature range of 2.0-6.3 Gg/yr. The added Figure S21 goes a long way toward alleviating these concerns by showing the insensitivity of the predicted number emissions over this range of literature variability. I note that this sort of variability is not captured within the error bars of Figure S21, but it is within the dashed lines. It would probably be good to emphasize the conclusions from this figure more.

10) Finally, I think it's worth mentioning in this manuscript that, unlike long-lived trace gases such as CO₂, particle number and mass are not conserved and the lifetime of these particles in the atmosphere is relatively short (particle number decreases on the order of some minutes due to coagulation and on the order of a few weeks due to deposition to the ground).

References cited:

Barrett et al. (2010) "Guidance on the use of AEDT gridded aircraft emissions in atmospheric models", A technical note submitted to the US Federal Aviation Administration.

Bond et al. (2018) "Bounding the role of black carbon in the climate system: A scientific assessment", *J. Geophys. Res.*, 118, 5380-5552.

Moore et al. (2017) "Take-off engine particle emission indices for in-service aircraft at Los Angeles International Airport", *Sci. Data*, 4, 170198.

Stettler et al. (2013) "Global Civil Aviation Black Carbon Emissions", *Environ. Sci. Technol.*, 47, 10397-10404.

Response to Referee #4:

The authors appreciate the valuable comments by the reviewer. The comments help us to improve the manuscript. A point-by-point response is as follows.

Annotation:

- (1) The comments of the reviewer are shown in *italic font*;
- (2) The explanations for the changes in the manuscript are shown in red with underlines;
- (3) The revised contents in the manuscript are shown in blue color;

- *The authors have put a lot of effort into addressing my previous comments, which is appreciated. Despite this, some specific concerns remain and are detailed below. I also think that it's important to not overstate what the authors have actually done by, for example, making strong statements like those on Lines 571-572 implying that a size-resolved number emissions inventory has been developed for the first time. This work developed a parameterisation of soot particle mode size in terms of engine T3 and EI_BCM by fitting published ground data, and this parameterisation is then used with an existing mass-based global emission inventory to convert the mass basis to a number basis, assuming the particles consist of a single, lognormal size distribution. The present work relies heavily on the pre-existing mass inventory for most of the heavy lifting. Prior number emissions inventories are also referenced for comparison (e.g., AERO2k developed by EUROCONTROL), so this is also not the first number emissions inventory. This study does place important order-of-magnitude-type constraints on the number of particles emitted by aircraft regionally and worldwide and is valuable for that reason; however, it's not clear to me how this paper might change/influence thinking in the field (a Nature Comm. review criterion). Figure 3 in the present work is visually pretty much the same as Figures 4 and S12-S14 in Stettler et al. (2013), except that the present Figure 3 is in number units instead of mass units. The underlying global mass emissions here appear to be quite different than prior work (e.g., Stettler et al.), while the number emissions are 2.5x that of AERO2k (Line 387). Who is right? The low number emission in AERO2k is not "caused" by the low mass emission; presumably both number and mass emissions are both lower for some common causal reason. It would be helpful to remember in this manuscript that many inventories try to predict emissions from aircraft, engine, and environmental conditions; meanwhile this study predicts emissions from environmental conditions and emissions themselves. Consequently, the method already has a leg up on the competition because it is starting with a slightly different form of the answer it seeks to constrain (in this case using a priori knowledge of BC mass emissions to predict BC number emissions).*

[Response]: Thank you very much for the comments. In the revised manuscript, we utilized neutral expressions to avoid overstating the importance of the work. We wrote “a size-resolved BC particle number emission inventory” instead of “first number size-resolved BC particle number emission inventory”. More discussions were added for the dependence on the BC mass emission. Please see the response to Comment No.9.

We also stressed that the current inventory is the estimations with our best effort for the time being based on the currently available data, and it has to be updated in the future in the Implications section:

“Most of the currently available relevant data and knowledge were integrated to make the estimations with our best effort for the time being, but considering the limited data, this emission inventory and the

proposed methods have to be continually updated in the future when new data and knowledge are obtained.”

We included discussion on the short lifetime of BC particles and the necessity to further study the transformation of the particles in the environment. Please see the response to Comment No.10.

In this study, we have integrated the currently available relevant data and knowledges, e.g. the recent standardized BC particle emission measurements, which were not available for the AERO2k development, so the fundamental data utilized in this study should be more reliable than those in AERO2k inventory.

Figure 4 has some similarities with the BC mass emission in Stettler et al. (2013), because the flight routes are the same in the two studies, which both utilized the flight data in Aviation Emission Inventory Code (AEIC). However, the spatial distributions are not the same, because the BC mass emissions utilized in this study are different from those calculated by the adopted FOX method in Stettler et al. (2013), and the BC number emissions are not linearly correlated with the BC mass emission.

- *1. In the previous review, I asked the authors to place their emissions factors in the context of terrestrial BC emissions, including the significant sources of biomass burning. The authors response is that since biomass burning emissions are from "naturally occurring fires" they are "out of scope of the present study". I think that this answer is evasive and non-responsive. I believe (as did the prior reviewer) that it is within the scope of the present study to contextualize the reported aviation emissions by comparison to the (expected) much larger surface emissions source. The fundamental question is do aviation emissions matter? Are they a significant source of BC mass or number, or is aviation a negligible source? Since this is a global study as evidenced by the paper title, the global emissions seem like the appropriate metric on which to evaluate importance.*

In a comprehensive review, Bond et al. (2016) report total BC emissions for the industrial and pre-1750's era of 6100 and 1400 Gg/yr, respectively. The total for all sources is 7500 Gg/yr. The mass emissions from aviation reported in the present study are 9.5 Gg/yr, which is 0.13% of all BC emissions and 0.16% of anthropogenic emissions reported by Bond et al. on a mass basis.

Bond et al. also report typical mass size distributions for biomass burning and industrial BC emissions that have mode diameters near 200 nm. To first order, we might assume a monodisperse aerosol with this mode size and unit density to translate the Bond et al. mass emissions to number emissions. From Table 6 of Bond et al, then we have:

Mass (Gg/yr.)	Number (#/yr.)	
All Sources:		
Energy	4770	9.1e27
Open Burning	2760	5.3e27
Total All	7530	1.4e28
 Pre-Industrial:		
Energy	390	7.4e26
Open Burning	1020	1.9e27
Total Pre-industrial	1410	2.7e27

Industrial:

Energy	4380	8.4e27
Open Burning	1740	3.3e27
Total Industrial	6120	1.2e28

From this, admittedly coarse, estimation method, it is worthwhile to note that the energy-related industrial number emissions of 8.4e27 are in reasonable agreement with the 8.6e27 number reported on Line 534 of the present manuscript. Now, accounting for the additional emissions from open burning, the contribution of aviation number emissions to anthropogenic emissions is $1.09e26/1.2e28 = 0.9\%$. The contribution of aviation number emissions to present day BC emissions is $1.09e26/1.4e28 = 0.8\%$.

[Response]: Thank you very much for bringing the results of open burning from Bond et al. (2013) to our attention. We agree with the reviewer. In the revised manuscript, the BC particle mass and number emissions from open burning were included. In order to avoid misunderstanding, in the revised manuscript, we clarified that the anthropogenic emissions of the ECLIPSE inventory do not include open burning emissions (grassland, woodland and forest fires). Please see Change 1. We used “open burning” instead of “biomass burning” because the ECLIPSE inventory already includes the residential biomass burning emissions.

The BC number emission from open burning was roughly estimated based on the BC mass emission data reported by Bond et al. (2013) with an approach similar to the one suggested by the reviewer. Instead of the reviewer’s monodisperse method, we assumed lognormal distributions for the BC particle sizes. There might be a miscalculation in the rough estimation in the reviewer’s comment. If we utilize the 200nm particle diameter and unit effective density (10^3 kg m^{-3}), the energy related BC mass emission (4770 Gg) is equivalent to 1.14×10^{27} particles, not 9.1×10^{27} .

$$Num = \frac{Mass}{m_p} = \frac{Mass}{\rho \cdot V_p} = \frac{4770 \times 10^6 [kg]}{10^3 [kg \cdot m^{-3}] \times \left(\frac{1}{6} \cdot \pi \cdot (200 \times 10^{-9} [m])^3 \right)} = 1.14 \times 10^{27}. \quad (S.1)$$

In the revised manuscript, the parameters for the number-size lognormal distributions of urban emissions and open burning emissions were obtained from the data in Bond et al. (2013) as shown in Figure S25. The GMDs were 75.2nm and 140.6nm respectively for the urban and open burning emissions, which were considerably smaller than the mode size (200 nm) of the mass-size distribution. The methods to estimate the BC particle number emissions were described in the revised manuscript, as shown in Change 2. The BC particle number emission from open burning was estimated to be 1.1×10^{27} , which was included in the comparison with the aviation emissions as shown in Change 3. In the revised manuscript, the open burning BC mass emission (Bond et al. 2013) was added to the ECLIPSE Version 5 (V5) anthropogenic BC mass emission inventory (Klimont et al. 2017) utilized in this study. Please see Change 4.

The previous versions of ECLIPSE inventories were also utilized as references for the energy-related emissions by Bond et al. (2013). The BC emission in ECLIPSE V5 is higher than that reported in Bond et al. (2013), which is mainly due to the inclusion of emissions from kerosene wick lamps, gas flaring, and the re-estimation of the emissions from China using the regional coal statistics (Klimont et al. 2017).

Finally our calculation shows that the aviation BC mass emissions are about 0.13% of the anthropogenic emissions according to the ECLIPSE V5 BC mass emission inventory (Klimont et al. 2017), and about 0.1% of all the emissions including the open burning emissions reported by Bond et al. (2013).

[Change 1]: In the revised manuscript, we clarified that the anthropogenic emissions do not include open burning emissions in Section 3.8:

“The above anthropogenic emissions exclude the open burning emissions (grassland, woodland and forest fires).”

[Change 2]: In the revised manuscript, a new section S8.2 was added in the supporting information to explain the method to roughly estimate the BC particle number emission from open burning:

“S8.2 Estimation of BC particle number emission from open burning

The BC particle number emission from open burning is not directly available in the literature. In a comprehensive review, Bond et al. (Bond et al. 2013) reported that the BC mass emission of open burning was about 2760 Gg per year. In this study, we roughly estimate the BC particle number emission by converting the mass emission into number emission, assuming a lognormal distribution of the BC particles and utilizing an effective density of 10^3 kg m^{-3} . The parameters for the lognormal distributions of urban emissions and open burning emissions were obtained from the data in Bond et al. (Bond et al. 2013) as shown in Figure S25. The GMDs were 75.2nm and 140.6nm respectively for the urban and open burning emissions. The BC particles from open burning are normally larger than those from urban emissions, which mainly originate from energy-related applications which have higher combustion efficiencies.

Figure S25 Size distributions of the BC particles in urban emissions and open burning, fitted by Bond et al. (2013).

The mass to number conversion follows the similar methods for the aviation emissions. Assuming that the particle is spherical, we can express the particle mass m_p as

$$m_p = \rho \cdot \frac{1}{6} \pi \cdot d^3, \tag{S.2}$$

where ρ is the effective density of the BC particle, which is assumed to be 10^3 kg m^{-3} ; d is the volume equivalent diameter, which follows the lognormal distribution as shown in Figure S25. The total BC mass *Mass* can be calculated as

$$Mass = \int_0^{\infty} m_p \cdot Num \cdot n(d_m) dd_m = \rho \cdot \frac{1}{6} \pi \cdot Num \int_0^{\infty} d^3 \cdot n(d_m) dd_m, \quad (S.3)$$

where Num is the total number of the BC particles. According to the properties of lognormal distribution (Equation (S.24)) described in Section S4.2, the total BC mass is calculated as

$$Mass = \rho \cdot \frac{1}{6} \pi \cdot Num \cdot GMD^3 \cdot \exp\left(\frac{3^2 \cdot (\ln(GSD))^2}{2}\right). \quad (S.4)$$

The GMD and GSD are the parameters of the lognormal distributions shown in Figure S25. The total BC particle number emission can be estimated as

$$Num = \frac{Mass}{\rho \cdot \frac{1}{6} \pi \cdot GMD^3 \cdot \exp\left(\frac{3^2 \cdot (\ln(GSD))^2}{2}\right)}. \quad (S.5)$$

The estimated BC particle number emission of open burning per year is about 1.1×10^{27} as shown in Table S11.

In order to evaluate the validity of this rough estimation method, we used this method to calculate the BC number emission based on the total BC mass emission (7264 Gg per year) in ECLIPSE V5 BC inventory (Klimont et al. 2017), and the result was 9.1×10^{27} per year which was close to the estimation (8.6×10^{27} per year) by Paasonen et al. (Paasonen et al. 2016) using the same mass inventory.

In this study, we adopted the global anthropogenic BC mass emission inventory ECLIPSE V5 (Klimont et al. 2017) to compare with the aviation emissions. The ECLIPSE inventory was utilized as a reference for energy-related emissions in Bond et al. 2013, but the value in Version 5 was higher than that reported in Bond et al. (2013) (See table S11), which was mainly due to the inclusion of emissions from kerosene wick lamps, gas flaring, and re-estimation of the emissions from China using the regional coal statistics (Klimont et al. 2017).

Table S11 Comparison between the BC emissions from ECLIPSE V5 (Klimont et al. 2017; Paasonen et al. 2016) and Bond et al. (2013)(Bond et al. 2013)

	ECLIPSE V5 (Klimont et al. 2017; Paasonen et al. 2016)	Bond et al. (2013)(Bond et al. 2013)
Anthropogenic BC mass (Gg)	7264	4770
Anthropogenic BC number (#)	8.6×10^{27}	6.0×10^{27}
Open burning BC mass (Gg)	\	2760
Open burning BC number (#)	\	1.1×10^{27}

”

[Change 3]: In the revised manuscript, the rough estimation of BC particle number emission from open burning was added in Paragraph 1, Section 3.8:

“The BC particle number emission from open burning was estimated to be 1.1×10^{27} per year based on the mass emission (2760 Gg per year) reported by Bond et al. (Bond et al. 2013), and the details of this estimation can be found in Section S8.2 of SI.”

The open burning emission was also included in the comparison with the aviation emission in Paragraph 2, Section 3.8:

“At the global scale, the BC number emissions from air traffic are equivalent to about 3.6% of the surface transportation emissions, about 1.3% of all the anthropogenic BC containing particles emitted on the

ground and about 1.1% of all the emissions including the open burning contribution.”

Change 4: In the revised manuscript, the BC mass emission from open burning reported by Bond et al. (2013) was included in the discussion in Section 3.8:

“However, the contribution of aviation to the BC mass emission is much lower, only equivalent to about 0.6% of the surface transportation (Klimont et al. 2017), 0.13% of all the surface anthropogenic emission (Klimont et al. 2017) and 0.1% of the total emissions including the open burning contribution, which is about 2760 Gg per year (Bond et al. 2013).”

- 2. The paragraph transition on Lines 72-74 is awkward. The new paragraph seems out of place here.

[Response]: Thank you very much for the comments. We moved this paragraph discussing the measurement system loss to Section 2.3 Dataset for model training. Please see **Change 5**.

Change 5: In the revised manuscript, we moved the paragraph about the measurement system loss to Section 2.3 Dataset for model training.

- 3. On Lines 87-88, the new number emissions inventory is mentioned for the first time. It would be good to have a link here to where it can be downloaded.

[Response]: We included the inventory as supplementary data. Please see **Change 6**. The link for download of the full dataset of Monte Carlo simulations will be provided in the future.

Change 6: In the last paragraph of the introduction, we explained that the inventory was included as supplementary data of this manuscript.

“In this study, a size-resolved BC particle number emission inventory (included as supplementary data) was developed for the global civil aviation based on the recent measurements.”

- 4. On Line 131, it is suggested that the mass EI is estimated from Equation 3, which is not right. In this study, EI_m is known from an existing global emission inventory.

[Response]: We agree with the reviewer. Equation 3 was only used to derive the correlation between EI_m and EI_n , and it was not used to estimate the mass emission index. We changed this sentence in the revised manuscript, please see **Change 7**.

Change 7: In the revised manuscript, we changed the introduction for Equation 3 in Section 2.1:

“The mass emission index EI_m is expressed as follows”

- 5. Where is the plot mentioned on Lines 176-177?

[Response]: The plot is Figure S4 in the Supporting Information. In the revised manuscript, we added the reference to this figure, please see **Change 8**.

Change 8: In the revised manuscript, we added the reference to Figure S4 for the GSD- T_3 and GSD- EI_m relations:

“The plot between GSD and T_3 is highly scattered, but there is a good dependence of GSD on EI_m (BC) (Figure S4)”.

- 6. On Line 425-426, it is stated that “Our results demonstrate that the global aviation BC particle

emission approximately follows a lognormal distribution as shown in Figure 4a and b...". This is circular reasoning as the lognormal size distribution is imposed from Equation 4. Of course this is not evident to the reader from the manuscript, but rather is buried on Pg. 20 of a 42-page supporting information document.

[Response]: In Equation 4 or Pg. 20 in the supporting information, we only assumed that the particles emitted from a single stage of a flight are lognormal, but the distribution in Figure 4 is the sum of the distributions from all aircrafts over a full year, which is not necessarily lognormal. Figure R1 shows that the sum of two lognormal distributions (LN_1 and LN_2) is no longer lognormal. As a result, no circular reasoning was applied here.

Figure R1 Demonstration for the sum of two lognormal distributions. LN_1 is lognormal distribution with a GMD of 48nm and a GSD of 1.8; LN_2 is lognormal distribution with a GMD of 22nm and a GSD of 1.4; the sum of LN_1 and LN_2 is shown as the dotted line (LN_1+LN_2).

- 7. The numbers reported in the paragraph at Lines 455-463 range pretty widely. Are all of these geometric number mean diameters? Similarly, are these only for the BC particles or do they include volatile particles (e.g., sulphuric acid and organics)? I suspect that the 11.8 nm GMD reported on Line 458 is actually capturing sulphur+organics (rather than BC-only) GMD. Similarly, the GMD of 60 nm on Line 460 is assumed by Ref. 65 (without validation), but is not the nominal value of 38 nm actually reported by the underlying reference (Page 11 of Barrett et al., 2010). Moore et al. (2017) report bimodal size distributions for aircraft take-off plumes that they measured for different aircraft types. The larger of the two modes likely is dominated by the contribution of the soot particles, while the smaller mode is composed of the volatile particles. How do the aircraft-specific, take-off size distributions in this study compare to those reported by that group?

[Response]: The numbers at Lines 455-463 are all geometric mean diameters for BC particles, which were reported on Page 11 of Barrett et al. (Barrett et al. 2010), as mentioned by the reviewer. The wide range was caused by the estimation method utilized in the report (Barrett et al. 2010). They utilized the combinations of mass emission indices (M_{NV}) 0.01~0.07 g/kg-fuel (nominal 0.03 g/kg-fuel) and the number emission indices (N_{NV}) 1×10^{14} ~ 60×10^{14} (nominal 4×10^{14}) to estimate the particle size (9 types of combinations in total):

Table of GMD estimations for non-volatile particles from the AEDT inventory (Barrett et al. 2010)

D_{NV} (nm)	Low N_{NV}	Nominal N_{NV}	High N_{NV}
Low M_{NV}	41	26	11
Nominal M_{NV}	60	38	15
High M_{NV}	79	50	20

The smallest GMD (11nm) was obtained when the high number emission and the low mass emission were utilized. The highest GMD (79nm) was obtained when the low number emission and the high mass emission were utilized.

The 11.8 nm GMD on Line 458 was only for the BC particles, excluding sulphuric and organics particles, in IPCC report 1999 (J.E.Penner et al. 1999) to assess the radiative forcing from black carbon aerosols. The related descriptions can be found in Section 6.4.2 “Direct Radiative Forcing from Black Carbon Aerosols” in Chapter 6 “Potential Climate Change from Aviation” of IPCC report 1999 (J.E.Penner et al. 1999). The main statements are quoted as follows:

“We reexamined results by inserting BC aerosol in a layer between 8 and 13 km in the GFDLR30 GCM using the method of Haywood and Ramaswamy (1998). A log-normal distribution with a geometric mean radius of 0.0118 μ m and a standard deviation of 2.0 was assumed.”

The GMD of 60 nm utilized by Zhou et al. (Zhou and Penner 2014) is corresponding to the estimation using low number emission index and nominal mass emission index, Row 2, Column 1 in the above table. The main statements can be found in Section 2.3 “Aircraft Soot as Heterogeneous IN”. (Zhou and Penner 2014), and were quoted as follows:

“We used a lognormal size distribution with a geometric standard deviation of 1.6 and a geometric diameter of 60nm to calculate the initial aircraft soot number [Barrett et al., 2010].”

Zhou et al. (Zhou and Penner 2014) adopted this value (60nm) for their assessment. Actually in our manuscript, we already included the nominal GMD (38nm), which agrees reasonably with our estimations: “In the AEDT inventory, the GMD was estimated to range from 11 to 79 nm, and 38 nm for the nominal mass and number emission scenarios (Barrett et al. 2010), which agrees reasonably with our estimations.”

The GMDs reported by Moore et al. (Moore et al. 2017) ranging from 24.7nm to 83.1nm with a 61.1nm GMD for all the samples, are generally larger than the data in our study, which has GMDs ranging from 14.7nm to 41nm shown as the training data in Table S1. The training data in this study were measured near the engine exit plane (usually closer than 1m), however the data reported by Moore et al. (Moore et al. 2017) were measured at a location more than 400m from the engine, where the emitted particles might have already mixed with the ambient particles (e.g. the particles from road traffic emissions) and grown to larger particles due to coagulation and condensation.

In this study, we only utilized standardized measurements near the engine exit plane, because the aerosol dynamics in the long distance measurement depends on a lot of external factors, including the meteorological conditions (e.g. humidity, temperature and atmospheric stability), the dilution of the plume and the ambient pollution (e.g. ambient particle concentration), which lead to much higher uncertainties. Aerosol dynamics is an important part for further assessment of the environmental and climate impacts of aviation emissions, so in the last paragraph of this manuscript, we stressed the necessity of micro-physics model for the aerosol dynamics to assess the transformation of the particles in the environment:

“The developed inventory contains the primary BC particles, and further studies are needed to understand the transformation of the particles in the environment (Wong et al. 2014). Sub-grid particle-cloud interaction models are also required to assess the climate impacts (Kärcher and Voigt 2017).”

- 8. Figure S7 is plotted in terms of thrust, while most other similar Figures S2 and S3 are shown in terms of T_3 . Please show these data in terms of T_3 as well (since that is the authors' chosen independent variable in the parameterisation) so that they can be compared to each other. What is "thrust" and how was it determined?

[Response]: We agree with the reviewer. In the revised manuscript, we added the plots with T_3 . Please see Change 9. “Thrust” is the power of the engine, namely the mechanical force to move the aircraft forward. During the standard measurements, the engine thrust levels were controlled according to the engine combustor inlet temperature (T_3) for which the corresponding thrust levels are known at sea level in the international standard atmosphere (15 °C, 1013.25 hPa)(Durdina et al. 2017). The GMD depends on both T_3 and EI_m especially for this unique double annular combustor engine, which has a discontinuity in the EI_m near 25% of the maximum power. The EI_m is much higher than single annular combustor engines before the discontinuity point, and becomes much smaller after the point. As a result, the GMD- T_3 or GMD- EI_m correlation is not able to capture the discontinuous behavior as shown in Figure S7(e) and (f). The GMD- $T_3&EI_m$ has much better performance when the influences of both T_3 and EI_m are included, shown as the black solid lines in Figure S7(e) and (f).

[Change 9]: In the revised manuscript, we added the plots with T_3 .

Figure S7 Evaluation of the estimations for a double annular combustor (CFM56-5B4-2P): (a) the changes of the GMD with thrust level; (b) the direct comparison between the measured and estimated GMDs; (c) the changes of $EI_n(BC)$ with thrust level; (d) the direct comparison between the measured and estimated $EI_n(BC)$, the dashed lines are 1:2 (or 2:1) ratio lines and the dotted lines are 1:3 (or 3:1) ratio lines; (e) the changes of the GMD with T_3 and the color indicates the BC mass emission indices EI_m ; (f) the changes of GMD with EI_m and the color indicates the combustor inlet temperature T_3 .

- 9. The fundamental concern that I have with this paper's conclusions is that if the mass EI is uncertain / poorly constrained, then the derived number EI will be too (even if the derived number is relatively stable due to offsetting errors in mass EI and mean size). The authors note that their estimates are based on a global mass emission of 9.5 Gg/yr, while AERO2k is based on 3.9 Gg/yr. Stettler et al., (2013) suggest that BC emissions are likely to be closer to 16.9 Gg/yr via the new FOX method, while also comparing their results to a literature range of 2.0-6.3 Gg/yr. The added Figure S21 goes a long way toward alleviating these concerns by showing the insensitivity of the predicted number emissions over this range of literature variability. I note that this sort of variability is not captured within the error bars of Figure S21, but it is within the dashed lines. It would probably be good to emphasize the conclusions from this figure more.

[Response]: Thank you very much for the comments. We added more discussions for the sensitivity tests in the revised manuscript, please see Change 10.

[Change 10]: In the revised manuscript, a new paragraph was added to discuss the sensitivity tests and

the variability of the estimated BC number emissions in Section 3.4 Particle number emission

“The uncertainty of the estimated BC particle number emissions depends on that of the BC mass emission. The sensitivity tests were conducted to evaluate the influences of the BC mass emissions, and the results indicated that when the BC mass emissions were 33% (3.2Gg) and 200% (19.1Gg) of the baseline emission (9.5 Gg), the variations of the estimated particle number emissions (0.6×10^{26} and 1.55×10^{26}) were near the 95% confidence interval ($0.68 \sim 1.5 \times 10^{26}$ per year), shown as the dashed lines in Figure S21. Therefore our number estimation was well constrained in the presence of significant mass emission uncertainty. More details about the sensitivity tests are introduced in Section S7.4 of SI.”

- 10. Finally, I think it's worth mentioning in this manuscript that, unlike long-lived trace gases such as CO₂, particle number and mass are not conserved and the lifetime of these particles in the atmosphere is relatively short (particle number decreases on the order of some minutes due to coagulation and on the order of a few weeks due to deposition to the ground).

[Response]: We agree with the reviewer. In the Implications section, we mentioned the short lifetime of BC particles and the necessity to further study on the transformation of the particles in the environment. Please see Change 11.

[Change 11]: The short lifetime of BC particles was added in the Implications section:

“Different from long-lived trace gases such as CO₂, airborne BC particle number and mass are not conserved with relatively short lifetime due to coagulation and deposition, so further studies are needed to understand the transformation of the particles in the environment (Wong et al. 2014).”

References

- Barrett, Steven, Michael Prather, Joyce Penner, Henry Selkirk, Sathya, Balasubramanian, Andreas Doppelheuer, Gregg Fleming, Mohan Gupta, Rangasayi Halthore, James Hileman, Mark Jacobson, Stephen Kuhn, Stephen Lukachko, Rick Miake-Lye, Andreas Petzold, Christopher Roof, Martin Schaefer, Ulrich Schumann, Ian Waitz, and Roger Wayson. 2010. "Guidance on the use of AEDT gridded aircraft emissions in atmospheric models." in *A technical note submitted to the US Federal Aviation Administration* Washington, DC: Federal Aviation Administration.
- Bond, T. C., S. J. Doherty, D. W. Fahey, P. M. Forster, T. Berntsen, B. J. DeAngelo, M. G. Flanner, S. Ghan, B. Kaercher, D. Koch, S. Kinne, Y. Kondo, P. K. Quinn, M. C. Sarofim, M. G. Schultz, M. Schulz, C. Venkataraman, H. Zhang, S. Zhang, N. Bellouin, S. K. Guttikunda, P. K. Hopke, M. Z. Jacobson, J. W. Kaiser, Z. Klimont, U. Lohmann, J. P. Schwarz, D. Shindell, T. Storelvmo, S. G. Warren, and C. S. Zender. 2013. "Bounding the role of black carbon in the climate system: A scientific assessment." *Journal of Geophysical Research-Atmospheres* 118(11):5380-552.
- Durdina, Lukas, Benjamin T. Brem, Ari Setyan, Frithjof Siegerist, Theo Rindlisbacher, and Jing Wang. 2017. "Assessment of Particle Pollution from Jetliners: from Smoke Visibility to Nanoparticle Counting." *Environmental Science & Technology*.
- J.E.Penner, D.H.Lister, D.J.Griggs, D.J.Dokken, and M.McFarland. 1999. "Aviation and the Global Atmosphere." Prepared in collaboration with the Scientific Assessment Panel to the Montreal Protocol on Substances that Deplete the Ozone Layer.
- Kärcher, B., and C. Voigt. 2017. "Susceptibility of contrail ice crystal numbers to aircraft soot particle emissions." *Geophysical Research Letters* 44(15):8037-46.
- Klimont, Z., K. Kupiainen, C. Heyes, P. Purohit, J. Cofala, P. Rafaj, J. Borken-Kleefeld, and W. Schöpp. 2017. "Global anthropogenic emissions of particulate matter including black carbon." *Atmos. Chem. Phys.* 17(14):8681-723.

- Moore, Richard H., Michael A. Shook, Luke D. Ziemba, Joshua P. DiGangi, Edward L. Winstead, Bastian Rauch, Tina Jurkat, Kenneth L. Thornhill, Ewan C. Crosbie, Claire Robinson, Taylor J. Shingler, and Bruce E. Anderson. 2017. "Take-off engine particle emission indices for in-service aircraft at Los Angeles International Airport." *Scientific Data* 4:170198.
- Paasonen, P., K. Kupiainen, Z. Klimont, A. Visschedijk, H. A. C. Denier van der Gon, and M. Amann. 2016. "Continental anthropogenic primary particle number emissions." *Atmos. Chem. Phys.* 16(11):6823-40.
- Wong, Hsi-Wu, Mina Jun, Jay Peck, Ian A. Waitz, and Richard C. Miake-Lye. 2014. "Detailed Microphysical Modeling of the Formation of Organic and Sulfuric Acid Coatings on Aircraft Emitted Soot Particles in the Near Field." *Aerosol Science and Technology* 48(9):981-95.
- Zhou, Cheng, and Joyce E. Penner. 2014. "Aircraft soot indirect effect on large-scale cirrus clouds: Is the indirect forcing by aircraft soot positive or negative?" *Journal of Geophysical Research: Atmospheres* 119(19):11,303-11,20.

REVIEWERS' COMMENTS:

Reviewer #4 (Remarks to the Author):

The authors have done a nice job responding to most of my comments.

Regarding Comment #6: Of course the sum of multiple lognormal modes is not necessarily lognormal - this is especially true if only two modes are considered, they are clearly separated in size space, and they are fairly equally weighted as in the idealized, illustrative example presented in the authors' response (Pg. 7 of response). However, this illustrative example is very different from the curves shown in Figure 4, where 1) the resolution of the size bins is much coarser (so small shoulders that indicate deviation from lognormality as in the illustrative example might not be visible), and 2) the dominant constituents of the overall curves (i.e., approach/climb-out in LTO and cruise and climb in CCD) largely overlap in size space (which is clearly not the case in the illustrative example). I think that the statement now on Lines 435-437 should be removed -- even if the reasoning isn't circular, the coarseness of the size distribution prevents such a conclusion from being drawn with certainty.

Regarding Comment #10: The proposed sentence change on Line 609-610 captures the lifetime issue, but I very much like the authors' discussion on Pg. 8 of their response that also captures the transformation issue that reads as follows: "In this study, we only utilized standardized measurements near the engine exit plane, because the aerosol dynamics in the long distance measurement depends on a lot of external factors, including meteorological conditions (e.g., humidity, temperature, and atmospheric stability), the dilution of the plume, and ambient pollution (e.g., ambient particle concentration)." I think that this text should be added to Line 610 as well in order to add specificity to the statement motivating the need to understand the transformation of particles in the environment. It also conveys to the reader that the tailpipe emissions inventory presented here is an upper limit since particles will be transformed in the atmosphere. As the authors indicate, their emissions results at 1 m behind the engine plane will be very different than an emission inventory based on results at even a short distance of 400 m behind the engine.

Finally, it's worth pointing out that while aviation may contribute BC number equivalent to 11% of road transport emission in North America (and different contributions elsewhere), most of these particles are emitted high in the upper troposphere, while the road transport emissions are emitted at the surface. Thus, if one is interested in the impacts of aviation on surface air quality, public health, or aerosol - warm cloud interactions, the contribution from aviation is likely to be much, much smaller than is implied by this 11% number. If one is interested in the direct climatic impact or the formation of contrail cirrus clouds, the role of aviation potentially becomes more important.

Response to Referee #4:

The authors appreciate the valuable comments by the reviewer. The comments help us to improve the manuscript. A point-by-point response is as follows.

Annotation:

- (1) The comments of the reviewer are shown in *italic font*;
- (2) The explanations for the changes in the manuscript are shown in red with underlines;
- (3) The revised contents in the manuscript are shown in blue color;

- *The authors have done a nice job responding to most of my comments. Regarding Comment #6: Of course the sum of multiple lognormal modes is not necessarily lognormal - this is especially true if only two modes are considered, they are clearly separated in size space, and they are fairly equally weighted as in the idealized, illustrative example presented in the authors' response (Pg. 7 of response). However, this illustrative example is very different from the curves shown in Figure 4, where 1) the resolution of the size bins is much coarser (so small shoulders that indicate deviation from lognormality as in the illustrative example might not be visible), and 2) the dominant constituents of the overall curves (i.e., approach/climb-out in LTO and cruise and climb in CCD) largely overlap in size space (which is clearly not the case in the illustrative example). I think that the statement now on Lines 435-437 should be removed -- even if the reasoning isn't circular, the coarseness of the size distribution prevents such a conclusion from being drawn with certainty.*

[Response]: Thank you very much for the suggestion. We agree with the reviewer. The statement was removed in the revised manuscript. Please see Change 1.

Change 1]: In the revised manuscript, we removed the following statement in the 2nd paragraph of Section "Particle size distribution":

"Our results demonstrate that the global aviation BC particle emission approximately follows a lognormal distribution as shown in Figure 4 (a) and (b), even though different size distribution peaks and shapes exist for individual flight phases. The similar result was observed for the integrated size distribution of the continental road emissions⁶⁶."

- *Regarding Comment #10: The proposed sentence change on Line 609-610 captures the lifetime issue, but I very much like the authors' discussion on Pg. 8 of their response that also captures the transformation issue that reads as follows: "In this study, we only utilized standardized measurements near the engine exit plane, because the aerosol dynamics in the long distance measurement depends on a lot of external factors, including meteorological conditions (e.g., humidity, temperature, and atmospheric stability), the dilution of the plume, and ambient pollution (e.g., ambient particle concentration)." I think that this text should be added to Line 610 as well in order to add specificity to the statement motivating the need to understand the transformation of particles in the environment. It also conveys to the reader that the tailpipe emissions inventory presented here is an upper limit since particles will be transformed in the atmosphere. As the authors indicate, their emissions results at 1 m behind the engine plane will be very different than an emission inventory based on results at even a short distance of 400 m behind the engine.*

[Response]: We agree with the reviewer. The text was added in the revised manuscript. Please see

Change 2.

[Change 2]: In the revised manuscript, we added the following text in the last paragraph of Section "Discussion":

"In this study, we only utilized standardized measurements near the engine exit plane, because the aerosol dynamics in the long distance measurement depends on a lot of external factors, including meteorological conditions (e.g., humidity, temperature, and atmospheric stability), the dilution of the plume, and ambient pollution (e.g., ambient particle concentration)."

- *Finally, it's worth pointing out that while aviation may contribute BC number equivalent to 11% of road transport emission in North America (and different contributions elsewhere), most of these particles are emitted high in the upper troposphere, while the road transport emissions are emitted at the surface. Thus, if one is interested in the impacts of aviation on surface air quality, public health, or aerosol - warm cloud interactions, the contribution from aviation is likely to be much, much smaller than is implied by this 11% number. If one is interested in the direct climatic impact or the formation of contrail cirrus clouds, the role of aviation potentially becomes more important.*

[Response]: We agree with the reviewer. The statement suggested by the reviewer was added in the revised manuscript. Please see Change 3.

[Change 3]: In the revised manuscript, we added the following text in the last paragraph of Section "Comparison to the surface emissions":

"Most of these particles from aviation are emitted in the upper troposphere and lowermost stratosphere, while the road transport emissions are emitted at the surface, so the contribution of aviation emissions to the surface BC concentration should be much smaller than that implied by these ratios."